# Discrete Diffusion in Large Language and Multimodal Models: A Survey

## Abstract

In this work, we provide a systematic survey of Discrete Diffusion Language Models (dLLMs) and Discrete Diffusion Multimodal Language Models (dMLLMs). Unlike autoregressive (AR) models, dLLMs and dMLLMs adopt a multi-token, parallel decoding paradigm using full attention and a denoising-based generation strategy. This paradigm naturally enables parallel generation and fine-grained output control. These capabilities are previously difficult to achieve with AR models. A growing number of industrial-scale proprietary d(M)LLMs, as well as a large number of open-source academic d(M)LLMs, have demonstrated performance comparable to their autoregressive counterparts, while achieving up to *10×* acceleration in inference speed. These developments position discrete diffusion models as a promising alternative to intelligence based on the traditional autoregressive approach. In this work, we present a comprehensive overview of the research in the dLLM and dMLLM domains. We trace the historical development of dLLMs and dMLLMs, formalize the underlying mathematical frameworks, list commonly-used modeling methods, and categorize representative models. We further analyze key techniques for training, inference, quantization. We also discuss the trustworthy issues and summarize emerging applications across language, vision-language, and biological domains. We conclude by discussing future directions for research and deployment.

## 1 Introduction

In recent years, Large Language Models (LLMs) and Multimodal Large Language Models (MLLMs) have demonstrated remarkable advances, exhibiting capabilities that increasingly resemble, or even surpass, human-level performance in domains traditionally associated with intelligence. Modern LLMs and MLLMs achieve superior scores on standard benchmarks designed for general knowledge, comprehension, and reasoning, suggesting that these systems are no longer merely text completion engines but competent general-purpose agents.

To date, the predominant paradigm for both LLMs and MLLMs has been autoregressive (AR) (OpenAI , 2024; OpenAI, 2024; DeepSeek-AI, 2025; Gemini Team , 2024; Gemini Team, 2025). Despite their successes, autoregressive (AR) models face intrinsic limitations. Their left-to-right decoding hinders parallel inference, reducing efficiency. They also struggle with precise structural control (e.g., length or format), making natural language inefficient for fine-grained orchestration of tools and agentic tasks. Moreover, causal attention forces one-pass static perception of inputs, limiting iterative refinement of token representations.

Discrete Diffusion Large Language Models (dLLMs) and Discrete Diffusion Multimodal Large Language Models (dMLLMs) (Nie et al., 2025b; Ye et al., 2025b; Yu et al., 2025; You et al., 2025; Li et al., 2025e; Yang et al., 2025b) have recently emerged as a promising direction. In tasks such as code generation (DeepMind, 2025; Inception Labs, 2025), planning (Ye et al., 2025b), and Sudoku (Ye et al., 2025b), dLLMs have been shown to achieve better performance than autoregressive models. For example, Gemini Diffusion DeepMind (2025), released in May 2025, achieves a latency advantage but trails the February 2025 Gemini 2.5 AR model by roughly three percent on average; Mercury Coder 2 Ermon (2026), released in February 2026, shows an average two percent improvement over the August 2025 GPT-5 mini baseline on code tasks. Moreover,

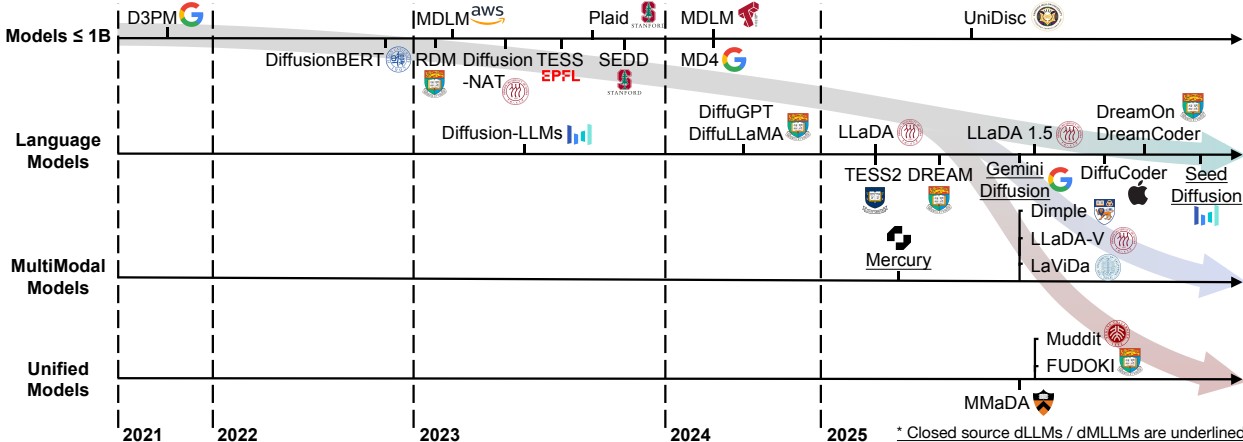

Figure 1: A timeline of existing dLLMs and dMLLMs in recent years. The timeline is established mainly according to the release date (e.g., the submission date to arXiv) of the technical paper for a model. The affiliation marked in the figure is based on the first affiliation listed in each paper.

Prabhudesai et al. (2025) demonstrates that in a data-constrained setting, increasing the number of training FLOPs for dLLMs enables them to outperform AR.

**Revised Part**

In contrast to autoregressive (AR) generation, discrete diffusion models formulate generation as an iterative denoising process over discrete token sequences. This paradigm removes the strict left-to-right constraint and enables parallel, structurally controllable generation through bidirectional attention. The key properties of discrete diffusion are as follows:

- **Parallel Decoding:** Unlike AR models, which generate tokens sequentially, discrete diffusion models produce multiple tokens simultaneously at each denoising step.

- **Alternative Control Mechanism:** Discrete diffusion frames generation as a denoising or infilling process rather than unconstrained left-to-right decoding. This formulation enables fine-grained control over output attributes such as length, format, and reasoning structure by conditioning on predefined templates.

- **Dynamic Contextual Representation:** Through bidirectional attention, discrete diffusion models continuously update their representations of both visual and linguistic context during generation. In contrast, autoregressive transformers compute key–value (KV) representations incrementally, and once tokens are generated, their representations remain fixed. Diffusion language models instead allow token representations to be refined based on both preceding and subsequent context.

Early efforts in discrete diffusion established the foundational mathematical formulations of discrete diffusion, introducing a token corruption scheme specifically designed for categorical data (Austin et al., 2021; Hoogeboom et al., 2021). These models demonstrated the feasibility of diffusion-based generation on various types of discrete data, including natural language (Austin et al., 2021; Zheng et al., 2023), images (Austin et al., 2021), and biological sequences such as DNA (Sahoo et al., 2024). In this early stage, experiments were limited to models with around or fewer than 1 billion parameters. Through simplifications and reparameterizations (Zheng et al., 2023; Sahoo et al., 2024; Shi et al., 2024), along with practical engineering efforts, the absorbing-state discrete diffusion formulation has gradually become the predominant mathematical framework adopted by open-source models and are termed as the masked diffusion model.

With the masked diffusion formulation, recent advances have significantly improved the scalability and effectiveness of discrete diffusion models (Gong et al., 2024; Nie et al., 2025a). A major breakthrough on the

industrial front came with the presentation of discrete diffusion-based large language models by Inception Labs and Google, namely *Mercury* (Inception Labs, 2025) and *Gemini Diffusion* (DeepMind, 2025). These models report comparable performance on code and mathematics benchmarks with their AR counterpart, while also achieving *10×* speedups in decoding, with about 1000 tokens per second.

In parallel, the research community has developed and open-sourced an increasing number of discrete diffusion-based language and multimodal models. The development began with dLLM models trained on large-scale text corpora, such as *LLaDA* (Nie et al., 2025b) and *Dream* (Ye et al., 2025b). Later, using the public available dLLM as the backbones, dMLLMs, such as *Dimple* (Yu et al., 2025), *LaViDa* (Li et al., 2025e), and *LLaDA-V* (You et al., 2025), are developed through multimodal alignment, instruction tuning, preference learning, and then reasoning enhancement.

To provide a comprehensive framework for understanding discrete diffusion large language models (dLLMs) and discrete diffusion multimodal large language models (dMLLMs), this survey systematically explores recent advances in modeling, training, generation and applications of discrete diffusion techniques.

In the rest of this paper, Sec. 2 presents the mathematical foundations of discrete diffusion models. Sec. 3 lists several modeling techniques for the discrete diffusion task. These techniques build upon mathematical formulations, primarily aiming to enhance model flexibility or introduce additional capabilities. Sec. 4 surveys representative discrete diffusion language models across varying scales. This includes early-stage models, scaled dLLMs, dMLLMs, and unified models. Sec. 5 discusses the key training strategies used in dLLMs and dMLLMs. Sec. 6 lists various inference techniques used in dLLMs and dMLLMs. Sec. 7.1 discusses the quantization techniques of dLLMs. Sec. 7.2 discusses trustworthy issues in dLLMs. Sec. 8 reviews the broad range of applications powered by discrete diffusion models. Finally, Appendix Sec.F summarize potential directions for future research. The organization of this survey is illustrated in Fig. 2.

Revised Part

## 1.1 Compared with Other Surveys

Earlier surveys, such as Li et al. (2025f); Yi et al. (2024); Zhang et al. (2026b); de Groot et al. (2025); Tseng et al. (2025), provide overviews of diffusion-based generative models. However, several of these works rely on relatively outdated literature and therefore do not reflect the most recent developments in the field.

For instance, Yi et al. (2024) divides text generation into conditional, unconstrained, and multi-mode categories, and their literature coverage primarily extends up to late 2023. de Groot et al. (2025) survey discrete diffusion models across natural language and genomic sequences, focusing on foundational formulations and adaptations of pretrained models. Their chronological overview includes only seven dLLM papers up to mid-2025, and does not cover billion-scale diffusion language models or unified language–vision systems. Tseng et al. (2025) propose a taxonomy of diffusion-based large language models based on sampling strategies, guidance types, noise schedules, and temporal conditioning. While informative, their survey covers work only up to early 2025, and includes only very limited discussion of subsequent developments (e.g., a single 2025 work).

As a result, these surveys primarily cover early-stage diffusion models in either continuous or discrete settings, and do not capture the rapid evolution of mask-based diffusion language models (dLLMs) and multimodal diffusion systems emerging after mid-2025. Notably, during this period, dLLMs have scaled to the billion-parameter regime, and both training and inference paradigms have undergone substantial changes compared to earlier small-scale models.

Our survey differs substantively from these works in both scope and organisation. We provide a comprehensive overview of discrete diffusion language and multimodal models through early 2026, cataloguing proprietary and open-source models exceeding one billion parameters—including LLaDA 1.5 Zhu et al. (2025a), Dream Ye et al. (2025b) and DreamOn Wu et al. (2025d), Seed Diffusion Song et al. (2025b), Fudoki Wang et al. (2025c), MMaDA Yang et al. (2025b), Muddit Shi et al. (2025), and Mercury Inception Labs (2025)—and covering multimodal systems such as Dimple Yu et al. (2025), LaViDa Li

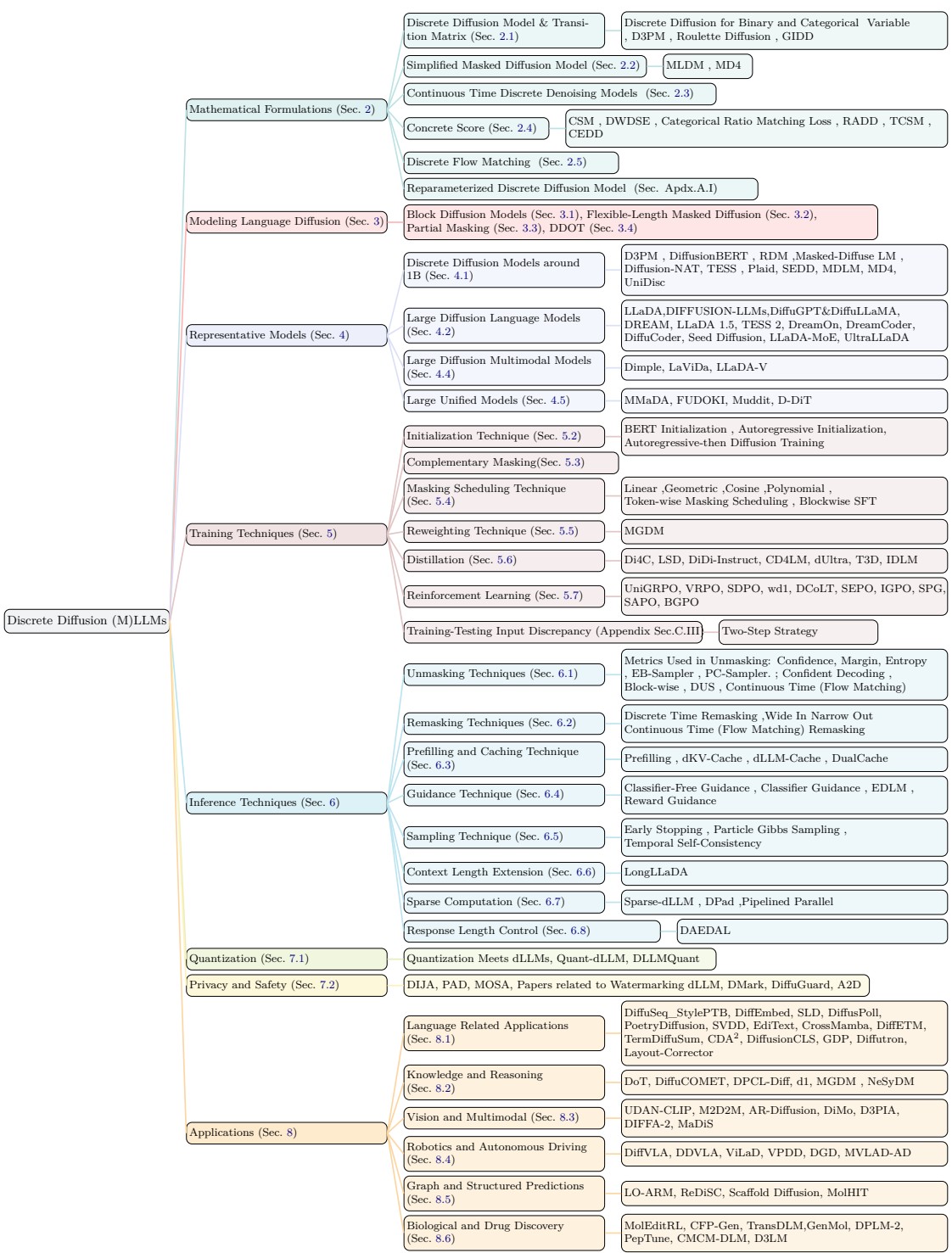

Figure 2: Overview of our survey. We begin by introducing the mathematical foundations (Sec. 2) and modeling methods (Sec. 3) of discrete diffusion language models . Next, we present a high-level overview of representative base models (Sec. 4), followed by discussions on training strategies (Sec. 5), inference techniques (Sec. 6) and quantization (Sec. 7.1). Furthermore, we discuss the privacy and safety studies of discrete diffusion language models (Sec. 7.2). In addition, we also review a wide range of applications (Sec. 8) that adopt discrete diffusion language models as their core model.

et al. (2025e), and LLaDA-V You et al. (2025). In contrast to earlier surveys that predate large-scale dLLMs, we synthesise recent industrial systems and emerging research architectures, enabling readers to understand the current state of the field. We also provide a rigorous mathematical exposition of discrete diffusion, tracing the development of masked and continuous-time formulations, and systematically analysing the evolution of forward and reverse processes across successive model generations.

Our organisation provides finer granularity than the high-level taxonomies in Li et al. (2025f). Li et al. (2025f) categorise training strategies into pre-training and post-training, and group inference techniques into parallel decoding, unmasking/remasking, guidance, and efficiency improvements. While these categories capture broad trends, they do not fully reflect the detailed design space. In contrast, we extend this taxonomy by explicitly distinguishing initialization strategies, complementary masking schedules, importance reweighting mechanisms, loss formulations, and reinforcement learning adaptations. For inference, we further cover adaptive sampling, heuristic and planner-based unmasking strategies, step-reduction techniques, KV-cache optimisation, dynamic length control, and hybrid autoregressive–diffusion decoding. This fine-grained organisation enables a more precise understanding of methodological differences and open research challenges.

Finally, Zhang et al. (2026b) survey parallel text generation and categorise methods into autoregressive (AR) and non-autoregressive paradigms, with diffusion models appearing only as a subsection (Section 4.2) within the non-AR category. Their primary focus lies on AR-based acceleration techniques such as speculative decoding and edit-based refinement, while diffusion-based approaches receive comparatively limited coverage. In contrast, our survey is fully dedicated to diffusion language models. We provide comprehensive discussions on training, inference, multimodal extensions, safety, and quantization, and construct a detailed timeline tracking the evolution from sub-billion to multi-billion parameter diffusion models. This holistic and up-to-date coverage, combined with in-depth theoretical analysis, positions our survey as a complementary and substantially extended reference to prior work.

## 2 Mathematical Formulations

In this section, we discuss the mathematical formulation of the discrete diffusion.

### 2.1 Discrete Diffusion Model and Transition Matrix

Diffusion models with discrete state spaces were initially introduced in Sohl-Dickstein et al. (2015) for binary variables, and later extended to categorical variables in Hoogeboom et al. (2021). Based on the previous works, Discrete Denoising Diffusion Probabilistic Models (D3PMs) (Austin et al., 2021) provides a general and flexible framework.

Let $x_0 \sim q(x_0)$ denote the data distribution over sequences composed of $K$ categorical values. The D3PM framework defines a forward Markov process $q(x_{1:T} \mid x_0)$ that gradually corrupts the data into noise, and a parameterized reverse process $p_\theta(x_{0:T})$ that learns to denoise:

$$q(x_{1:T} \mid x_0) = \prod_{t=1}^{T} q(x_t \mid x_{t-1}), \tag{1}$$

$$p_\theta(x_{0:T}) = p(x_T) \prod_{t=1}^{T} p_\theta(x_{t-1} \mid x_t). \tag{2}$$

Each $x_t$ is a discrete random variable and $q(x_t \mid x_{t-1})$ is defined via a time-dependent transition matrix $Q_t$, with categorical transition probabilities:

$$q(x_t \mid x_{t-1}) = \mathrm{Cat}(x_t; p = x_{t-1}Q_t), \tag{3}$$

where $x_{t-1}$ is a one-hot vector and $Q_t \in \mathbb{R}^{K \times K}$ is a row-stochastic matrix. The marginal distribution $q(x_t \mid x_0)$ and the posterior $q(x_{t-1} \mid x_t, x_0)$ are:

$$q(x_t \mid x_0) = \text{Cat}(x_t; p = x_0 Q_{1:t}), \tag{4}$$

$$Q_{1:t} = Q_1 Q_2 \ldots Q_t, \tag{5}$$

$$q(x_{t-1} \mid x_t, x_0) = \text{Cat}\left(x_{t-1}; p = \frac{x_t Q_t^\top \circ x_0 Q_{1:t-1}}{x_0 Q_{1:t} x_t^\top}\right). \tag{6}$$

D3PM framework supports various types of transition matrices $Q_t$, each inducing a different corruption behavior. Here, we present the two most commonly used transition matrices: uniform and absorbing. Additional types, including hybrid, discretized Gaussian, band-diagonal, and embedding-based, are described in Appendix Section A.I.

- **Uniform**: $Q_t = (1 - \beta_t)I + \frac{\beta_t}{K}\mathbf{1}\mathbf{1}^\top$ yields a uniform stationary distribution. The uniform transition matrix looks like

$$Q_t^{\text{uniform}} = \begin{bmatrix} 1 - \frac{K-1}{K}\beta_t & \frac{\beta_t}{K} & \cdots & \frac{\beta_t}{K} \\ \frac{\beta_t}{K} & 1 - \frac{K-1}{K}\beta_t & \cdots & \frac{\beta_t}{K} \\ \vdots & \vdots & \ddots & \vdots \\ \frac{\beta_t}{K} & \frac{\beta_t}{K} & \cdots & 1 - \frac{K-1}{K}\beta_t \end{bmatrix}. \tag{7}$$

- **Absorbing**: $Q_t = (1 - \beta_t)I + \beta_t \mathbf{1}e_m^\top$, where $e_m$ is a vector with a one on the absorbing state and zeros elsewhere. Tokens either remain unchanged or are mapped to a special [MASK] token with probability $\beta_t$. The absorbing transition matrix looks like

$$Q_t^{\text{absorb}} = \begin{bmatrix} 1 - \beta_t & 0 & \cdots & 0 & \beta_t \\ 0 & 1 - \beta_t & \cdots & 0 & \beta_t \\ \vdots & \vdots & \ddots & \vdots & \vdots \\ 0 & 0 & \cdots & 1 - \beta_t & \beta_t \\ 0 & 0 & \cdots & 0 & 1 \end{bmatrix}. \tag{8}$$

Following the $x_0$-parameterization, the model predicts $p_\theta(x_{t-1} \mid x_t)$ using:

$$p_\theta(x_{t-1} \mid x_t) \propto \sum_{\tilde{x}_0} q(x_{t-1}, x_t \mid \tilde{x}_0)\tilde{p}_\theta(\tilde{x}_0 \mid x_t), \tag{9}$$

where $\tilde{p}_\theta(\tilde{x}_0 \mid x_t)$ is a network predicting logits over $x_0$. This parameterization ensures the reverse distribution respects the sparsity pattern of $Q_t$.

The loss function combines the variational lower bound $\mathcal{L}_{\text{vb}}$ with an auxiliary cross-entropy denoising term $\mathcal{L}_{\text{ce}}$:

$$\mathcal{L}_\lambda = \mathcal{L}_{\text{vb}} + \lambda \mathcal{L}_{\text{ce}}, \tag{10}$$

$$\mathcal{L}_{\text{vb}} = \mathbb{E}_{q(x_0)}\left[\underbrace{\text{KL}(q(x_T \mid x_0) \| p(x_T))}_{\mathcal{L}_T} + \underbrace{\sum_{t=2}^{T} \mathbb{E}_{q(x_t|x_0)}\left[\text{KL}(q(x_{t-1} \mid x_t, x_0) \| p_\theta(x_{t-1} \mid x_t))\right]}_{\mathcal{L}_{t-1}}\right.$$

$$\left. - \underbrace{\mathbb{E}_{q(x_1|x_0)}\left[\log p_\theta(x_0 \mid x_1)\right]}_{\mathcal{L}_0}\right], \tag{11}$$

$$\mathcal{L}_{\text{ce}} = \mathbb{E}_{q(x_0)}\mathbb{E}_{q(x_t|x_0)}[-\log \tilde{p}_\theta(x_0 \mid x_t)]. \tag{12}$$

In $\mathcal{L}_{\text{vb}}$,

- $\mathcal{L}_T$ is the KL divergence between the forward terminal distribution $q(x_T \mid x_0)$ and the prior $p(x_T)$,

- $\mathcal{L}_{t-1}$ is the KL divergence between the forward posterior $q(x_{t-1} \mid x_t, x_0)$ and the learned reverse model $p_\theta(x_{t-1} \mid x_t)$ at each intermediate step,

- $\mathcal{L}_0$ is the cross-entropy loss for reconstructing $x_0$ from $x_1$ using the reverse model.

Such decomposition enables the model to be trained efficiently by sampling time steps $t$ uniformly and estimating each term using stochastic gradient descent. The forward posterior $q(x_{t-1} \mid x_t, x_0)$ has a closed-form expression under categorical diffusion, and the model is typically parameterized to predict $p_\theta(x_0 \mid x_t)$, from which $p_\theta(x_{t-1} \mid x_t)$ is derived analytically. The auxiliary $\mathcal{L}_{ce}$ is added to encourage accurate prediction of the original data $x_0$ from corrupted samples.

Besides a flexible representation of discrete diffusion, D3PM also unifies various paradigms such as BERT, autoregressive models, and masked language models within the discrete diffusion framework.

## 2.2 Simplified Masked Diffusion Model

A widely adopted class of discrete diffusion models is based on the absorbing state and is often referred to as the *Masked Diffusion Model*. Both Shi et al. (2024) and Sahoo et al. (2024) introduced simplifications to the diffusion process and the corresponding training objective for masked diffusion, yielding improved performance and computational efficiency. For the simplification of general discrete diffusion, we discuss Reparameterized Discrete Diffusion Models (RDMs) (Zheng et al., 2023) in Appendix Sec.A.II.

In masked diffusion, the forward process progressively replaces input tokens with a special `[MASK]` token. Once a token is masked, it remains in that state throughout the remainder of the process, making the `[MASK]` token an absorbing state. At each time step $t$, the forward transition for a token $x$ is defined as:

$$q(x_t \mid x_0) = \text{Cat}(x_t; \alpha_t x_0 + (1 - \alpha_t)m), \tag{13}$$

where $m$ is the one-hot vector corresponding to the `[MASK]` token, and $\alpha_t \in [0, 1]$ is the monotonically decreasing schedule such that $\alpha_0 \approx 1$ and $\alpha_T = 0$. Here, $t = 0$ corresponds to clean tokens, while $t = T$ corresponds to complete noise.

The reverse process aims to denoise the masked sequence by substituting `[MASK]` tokens with predicted tokens. Importantly, unmasked tokens are carried out unchanged throughout the denoising process. The reverse posterior at a previous time $s$ conditioned on $x_t$ and $x_0$ is given by:

$$q(x_s \mid x_t, x_0) = \begin{cases} \text{Cat}(x_s; x_t), & \text{if } x_t \neq m, \\ \text{Cat}\left(x_s; \dfrac{(1 - \alpha_s)m + (\alpha_s - \alpha_t)x_0}{1 - \alpha_t}\right), & \text{if } x_t = m. \end{cases} \tag{14}$$

This formulation reflects two key properties of the masking process: (1) If the current token $x_t$ is not masked, the posterior is deterministic: $x_s = x_t$. (2) If the token is masked, then the posterior is a linear interpolation between the mask vector $m$ and the clean token $x_0$, scaled by the noise schedule parameters $\alpha_s$ and $\alpha_t$.

Let $f_\theta(x_t)$ be the neural network output predicting the original token $x_0$ from the noisy input $x_t$. The above reverse transition is rewritten as:

$$p_\theta(x_s \mid x_t) = \begin{cases} \text{Cat}(x_s; x_t), & \text{if } x_t \neq m, \\ \text{Cat}\left(x_s; \frac{(1 - \alpha_s)m + (\alpha_s - \alpha_t)f_\theta(x_t)}{1 - \alpha_t}\right), & \text{if } x_t = m. \end{cases} \tag{15}$$

The variational lower bound for discrete diffusion can be simplified using the above formulation, leading to a final loss in the form of

$$\mathcal{L} = \sum_{t=2}^{T} \mathbb{E}_{x_0, x_{1:T}} \left[ -\frac{\alpha_{t-1} - \alpha_t}{1 - \alpha_t} \sum_{n=1}^{N} \delta_m(x_{t,n}) x_{0,n} \log[f_\theta(x_t)]_n \right], \tag{16}$$

where $\delta_m(x_{t,n})$ denotes the indicator function, $\delta_m(x_{t,n}) = 1$, if the n-th token of $x_t$ is a masked token, otherwise, $\delta_m(x_{t,n}) = 0$.

By defining the discrete time series as $0, \frac{1}{T}, \ldots, 1 - \frac{1}{T}, 1$ and letting $T \to \infty$, both works Sahoo et al. (2024); Shi et al. (2024) extend the above loss Eq. (16) to the continuous-time setting:

$$\mathcal{L} = \int_0^1 \mathbb{E}_{x_{0:1}} \Big[ \frac{\alpha_t'}{1 - \alpha_t} \sum_{n=1}^N \delta_m(x_{t,n}) x_{0,n} \log[f_\theta(x_t)]_n \Big] dt. \tag{17}$$

This loss corresponds to a reweighted cross-entropy loss evaluated only over masked tokens. Such loss formulation is significantly simpler than the original variational bound and has become the standard training objective for subsequent large discrete diffusion models.

## 2.3 Continuous Time Discrete Denoising Models

D3PM operates in discrete time, *i.e.*, with time steps $t = 0, 1, 2, \ldots, T$. Campbell et al. (2022) describes a continuous-time framework for discrete denoising models, formulated as a Continuous-Time Markov Chain (CTMC), where $t \in [0, T]$. This approach generalizes discrete diffusion models by allowing transitions at arbitrary time points. The infinitesimal transition probabilities are given by:

$$q_{t|t-\Delta t}(x' \mid x) = \delta_{x,x'} + R_t(x, x')\Delta t + o(\Delta t). \tag{18}$$

This process converges to a tractable reference distribution as $t \to T$. The time-reversed generative process is another CTMC with reverse rate matrix $\hat{R}_t$, expressed as:

$$\hat{R}_t(x, x') = R_t(x', x) \sum_{x_0} \frac{q_{t|0}(x' \mid x_0)}{q_{t|0}(x \mid x_0)} p_\theta(x_0 \mid x), \tag{19}$$

where $p_\theta(x_0 \mid x)$ is a learnable denoising model approximating $q_{0|t}(x_0 \mid x)$.

The training of $p_\theta(x_0 \mid x)$ is guided by a continuous-time version of the variational lower bound. Let $Z_t(x) = \sum_{x' \neq x} R_t(x, x')$ be the total outgoing rate and $r_t(x' \mid x) = R_t(x, x')/Z_t(x)$ the normalized jump probability. The continuous-time variational lower bound is:

$$\mathcal{L}_{\mathrm{vb}}(\theta) = T \, \mathbb{E}_{t \sim \mathcal{U}(0,T)} \mathbb{E}_{x \sim q_t} \mathbb{E}_{x' \sim r_t(\cdot|x)} \Big[ \sum_{x'' \neq x} \hat{R}_t^\theta(x, x'') - Z_t(x) \log \hat{R}_t^\theta(x', x) \Big] + C, \tag{20}$$

where $C$ is constant with respect to $\theta$. This objective can be efficiently optimized using stochastic gradient descent by sampling $(x, x')$ pairs according to the forward process.

During inference, however, the exact simulation of the reverse CTMC can be computationally prohibitive. Instead, the tau-leaping algorithm Gillespie (2001) approximates the reverse process by applying multiple transitions within a time interval $\tau$ simultaneously. For a current state $x_t$, the number of transitions to each $x'$ during $[t - \tau, t]$ is modeled as:

$$P_{x'} \sim \mathrm{Poisson}(\tau \cdot \hat{R}_t^\theta(x_t, x')). \tag{21}$$

The next state $x_{t-\tau}$ is obtained by aggregating the sampled transitions. This method supports parallel decoding by allowing simultaneous updates across multiple dimensions.

To further improve sample fidelity, predictor-corrector steps are used. After a tau-leaping step, corrector transitions with rate matrix $R_c = R_t + \hat{R}_t^\theta$ are applied to refine the sample distribution toward the target marginal $q_t(x)$. This approach is analogous to Langevin correctors in continuous diffusion models.

## 2.4 Concrete Score

In the continuous-time discrete diffusion framework, as formulated in previous Campbell et al. (2022), the reverse process can also be analytically expressed in terms of the forward transition rate matrix and a function known as the concrete score (Meng et al., 2022). This construction enables training via score matching, analogous to score-based models in continuous diffusion model.

Let $R_t(x, x')$ be the forward transition rate matrix of a continuous-time Markov chain (CTMC) over a discrete state space $\mathcal{X}$. The reverse-time transition rate $\hat{R}_t(x, x')$ can be formulated as:

$$\hat{R}_t(x, x') = \begin{cases} \dfrac{p_t(x')}{p_t(x)} R_t(x', x), & x' \neq x, \\ -\sum_{k \neq x} \hat{R}_t(x, k), & x' = x. \end{cases} \tag{22}$$

Here, the scalar ratio $\frac{p_t(x')}{p_t(x)}$ is referred to as the concrete score. It quantifies the relative likelihood of two discrete states at time $t$ and modulates the reverse transition rate accordingly. Thus, instead of learning the full reverse transition distribution, one can train a model $s_\theta(x, t)$ to estimate the concrete score:

$$s_\theta(x, t) \approx \left[ \frac{p_t(x')}{p_t(x)} \right]_{x' \in \mathcal{X}}. \tag{23}$$

In the Appendix Sec.A.III, we discuss the commonly-used training loss under concrete score formulation, the connection of concrete score with traditional cross-entropy loss and the time independency simplification of concrete score.

## 2.5 Discrete Flow Matching (DFM)

Build upon the continuous-time Markov chain (CTMC) framework in Continuous Time Discrete Denoising Models (Campbell et al., 2022), Discrete Flow Matching (DFM) (Gat et al., 2024) extends the Flow Matching paradigm to categorical sequence data. The model defines a probability path $p_t$ interpolating between a source distribution $p$ (e.g., all-mask sequences) and a target distribution $q$ (e.g., the data distribution), such that $p_0 = p$ and $p_1 = q$. We note that this direction of time is the opposite of the notation used in Sec. A.2.

Given a coupling distribution $\pi(x_0, x_1)$ between source and target sequences, the marginal probability at time $t$ is defined as:

$$p_t(x) = \sum_{x_0, x_1 \in D} p_t(x \mid x_0, x_1) \pi(x_0, x_1), \tag{24}$$

where the conditional path $p_t(x \mid x_0, x_1)$ factorizes over positions:

$$p_t(x \mid x_0, x_1) = \prod_{i=1}^{N} p_t(x^i \mid x_0, x_1), \tag{25}$$

with token-level conditional paths defined as a convex combination of basis distributions:

$$p_t(x^i \mid x_0, x_1) = \sum_{j=1}^{m} \kappa_{t,j}^i w_j(x^i \mid x_0, x_1), \tag{26}$$

where $\kappa_{t,j}^i \geq 0$, $\sum_j \kappa_{t,j}^i = 1$ form a scheduler controlling the path dynamics.

The generative process is defined via probability velocity fields $\{u_t^i\}_{i=1}^N$ guiding transitions between states. The update rule for sampling is:

$$x_{t+h}^i \sim \delta_{x_t^i}(\cdot) + h u_t^i(\cdot, x_t), \tag{27}$$

where $u_t$ is said to generate $p_t$ if the process satisfies:

$$p_{t+h}(x) = p_t(x) - h \operatorname{div}_x(p_t u_t) + o(h), \tag{28}$$

with the discrete divergence operator:

$$\operatorname{div}_x(v) = \sum_{z \in D} \left[ v(z, x) - v(x, z) \right]. \tag{29}$$

For the convex interpolation path:

$$p_t(x^i \mid x_0, x_1) = (1 - \kappa_t)\delta_{x_0}(x^i) + \kappa_t \delta_{x_1}(x^i), \tag{30}$$

the corresponding generating velocity takes the closed form:

$$u_t^i(x^i, z) = \frac{\dot{\kappa}_t}{1 - \kappa_t} \left[ p_{1|t}(x^i \mid z) - \delta_z(x^i) \right], \tag{31}$$

where $p_{1|t}(x^i \mid z)$ is the probability denoiser: the conditional probability of the target token $x_1^i$ given the current state $z$.

To estimate the posteriors required in the generative process, the model minimizes the cross-entropy loss:

$$\mathcal{L}(\theta) = -\sum_{i=1}^{N} \mathbb{E}_{t,(x_0,x_1),x_t} \left[ \log p_{1|t}(x_1^i \mid x_t; \theta) \right], \tag{32}$$

where $x_t \sim p_t(\cdot \mid x_0, x_1)$.

Revised Part

### 2.6 Mathematical Formulation of dMLLMs and Unified Models

Discrete Diffusion Multimodal Large Language Model (dMLLM) extend dLLMs by conditioning on a continuous vision representation. Let $x = (x_1, \ldots, x_L)$ denote a sequence of discrete tokens (words) and let $v$ denote the image input. A vision encoder $E_{\mathcal{V}}$ produces continuous features $h_v = E_{\mathcal{V}}(v)$. The forward corruption process of a dMLLM is the same continuous-time Markov chain used in dLLMs: at time $t \in [0, T]$, each discrete token $x_i$ is replaced by a mask token with rate $\lambda(t)$; the image features $h_v$ remain fixed because the image is not discretized. The reverse process learns a conditional score function $s_\theta(\cdot)$ that predicts the distribution of the original tokens given their corrupted version and the vision features: $q_\theta(x, |, \tilde{x}_t, h_v)$. The training objective is the same negative evidence lower bound used in dLLMs, except that the model must marginalize over the corruption process conditional on $h_v$. Because the vision encoder output is continuous and fixed across all denoising steps, image tokens are not masked and are not sampled; only the discrete tokens are corrupted and denoised. The training of discrete diffusion multimodal language models (dMLLMs) generally follows a multi-phase strategy that combines a pre-trained visual backbone, vision-language alignment, and post-training. First, a strong vision encoder is adopted—often pre-trained on large-scale image datasets—to produce continuous visual embeddings. These embeddings remain fixed throughout the diffusion process and serve as context for language generation. Second, an alignment phase is performed to connect visual and linguistic modalities. This phase typically uses an autoregressive objective with a causal attention mask to ensure that the language model can attend to the visual embeddings and generate coherent text conditioned on images. Third, post-training refines the model using supervised fine-tuning (SFT) on instruction-following datasets and, when applicable, reinforcement learning (RL) based preference optimization.

Unified models seek to denoise text and images with a single discrete diffusion or flow process. Images are quantized into discrete tokens via a VQ-VAE or VQ-GAN; each token $y_i$ represents either a word from the text vocabulary or a code from the image codebook. The forward corruption is a continuous-time Markov chain that independently replaces each token by a mask token at rate $\lambda(t)$, regardless of modality. The reverse process is a masked token predictor that infers the original discrete tokens for both text and image modalities: $q_\theta(y, |, \tilde{y} * t)$. Because both modalities are discrete, there is no continuous vision encoder; all tokens, including image codes, are masked and denoised. For example, Muddit formulates a unified CTMC over a finite alphabet and defines the training objective as a continuous-time negative ELBO shared across modalities. MMaDA generalizes this by combining a discrete diffusion loss $L_{\text{diff-disc}}$ for masked token prediction with a continuous diffusion loss $L_{\text{diff-cont}}$ for continuous latent variables (when included), and FUDOKI extends the framework using discrete flow matching instead of diffusion, introducing metric-induced probability paths and kinetic optimal velocities. Unified models therefore treat images as sequences of discrete tokens, mask both text and image tokens, and denoise them jointly;

the continuous features of the original images do not appear in the model but are represented through discrete codebook indices.

In dMLLMs, images are encoded into continuous embeddings by a vision encoder and are not discretized or masked; only the text tokens participate in the discrete diffusion process. The model conditions on image features through cross-attention and requires a predetermined output length. In unified models, images are quantized into discrete tokens via VQ-VAE or VQ-GAN, and the same diffusion or flow process corrupts and denoises both text and image tokens; there is no continuous vision encoder during sampling. Unified models can therefore generate images and text within a single transformer by masking and denoising a mixed sequence of discrete codes. These distinctions will be clearly articulated in the revised manuscript to address the reviewer's questions about tokenization, masking and integration of continuous image features.

## 3 Modeling Language Diffusion

Beyond the mathematical formulation introduced in the previous section, in this section, we present several techniques used when modeling the language diffusion task, such as specialized neural network designs, and extra sub-tasks. These techniques are introduced to enhance the controllability and flexibility.

### 3.1 Block Diffusion Models

**Revised Part**

Block diffusion models address the computational inefficiency of applying diffusion-based denoising over the entire sequence with full bidirectional attention. In standard discrete diffusion language models, all tokens are jointly refined, which leads to expensive global attention computation and makes standard KV-cache reuse difficult. Block diffusion decomposes the sequence into multiple blocks: tokens within each block are generated through diffusion-based denoising, while previous blocks serve as autoregressive context. This design improves inference efficiency by reducing the effective denoising region and also partially relaxes the fixed-length limition in generation, since generation can proceed block by block rather than requiring the whole output sequence to be specified in advance.

Block Diffusion models (BD3-LMs) (Arriola et al., 2025a) provide a hybrid framework that interpolates between autoregressive language models and fully parallel diffusion models. Instead of denoising all tokens simultaneously, BD3-LMs segment the sequence into blocks and perform discrete denoising diffusion within each block, while conditioning autoregressively on all preceding blocks. The mathematical formualtion of BD3-LMs are provided in Appendix Sec.B.I.

### 3.2 Flexible-Length Masked Diffusion

**Revised Part**

Flexible-length masked diffusion is designed to overcome the fixed-length generation constraint of standard masked diffusion models. Conventional masked diffusion typically initializes generation with a sequence of mask tokens of predetermined length, implying that the target output length must be specified before decoding. This assumption limits its applicability to open-ended generation, editing, and insertion-based generation. Flexible-length masked diffusion introduces additional sequence-editing states or operations, such as insertion and deletion, so that the model can dynamically expand or shrink the generation canvas during denoising. As a result, the model can generate variable-length outputs without relying on a fixed preset sequence length.

Flexible-Length Masked Diffusion (FlexMDM) (Kim et al., 2025a) extends masked diffusion models to handle variable-length sequences by introducing a third token state, called empty, in addition to the usual masked and ground truth states. This allows the model not only to recover missing tokens (unmasking) but also to insert new tokens into the sequence. During inference, the process alternates between two steps: first, the model predicts how many new mask tokens should be inserted before each existing token, effectively expanding the sequence length; then, it replaces all mask tokens with actual content tokens. By repeating this insertion–unmasking cycle, FlexMDM can gradually generate longer sequences, enabling it to produce

variable-length outputs while maintaining the flexibility of generating tokens in any order. In Appendix Sec. B.II, we provide the mathematical formulation of FlexMDM along with a more detailed description of its generation process.

### 3.3 Partial Masking

Revised Part
Partial masking addresses the inefficient use of intermediate denoising states in standard discrete diffusion. In conventional masked diffusion, a token may remain masked for several denoising steps, and intermediate model predictions before the final unmasking step are not explicitly preserved in the generated sequence. This creates a mismatch between the multi-step computation used during decoding and the discrete token-level update that only occurs when a token is finally unmasked. Partial masking mitigates this issue by decomposing each original token $x_i$ into multiple sub-tokens $(y_{i,1}, \ldots, y_{i,\ell})$. Instead of decoding the whole token in a single update, the model can progressively reveal sub-token information across multiple denoising steps. This enables a finer-grained denoising trajectory and allows intermediate computation to contribute more directly to token reconstruction.

In masked diffusion model, each token has only two states. Partial Masking (Chao et al., 2025) introduces an additional *intermediate state* by decomposing each token into a sequence of sub-tokens.

Each original token $x_0^i \in \mathcal{X}$, where $\mathcal{X} = \{0, \ldots, C-1\}$, is mapped into a sub-token sequence $y_0^i = [y_0^{i,1}, \ldots, y_0^{i,\ell}] \in \mathcal{Y}^\ell$ through an invertible base-$b$ encoding function $f$: $f : \mathcal{X} \to \mathcal{Y}^\ell$, $y_0^i = f(x_0^i)$, where $\mathcal{Y} = \{0, \ldots, b-1\}$ and $b = \lceil \sqrt[\ell]{C} \rceil$. The inverse mapping $f^{-1}$ reconstructs tokens from sub-tokens, ensuring lossless transformation: $x_0^i = f^{-1}(y_0^i)$. Consequently, the forward pass, backward pass, training, and inference of diffusion with partial masking all occur at the sub-token level. Here, we use subscript $t = 0$ to represent clean tokens, consistent with the original paperChao et al. (2025); however, this is the opposite of the notation used in Sec. A.2.

### 3.4 Diffusion with Optimal Transport Position Coupling

Revised Part
Diffusion with optimal transport position coupling addresses the lack of positional flexibility in standard discrete diffusion models. Conventional discrete diffusion assumes a fixed sequence canvas, where token positions are predetermined throughout the denoising process. This assumption is restrictive for tasks such as infilling, continuation, and text editing, where newly generated tokens may require position reallocation and existing position IDs may need to be adjusted after insertion or restructuring. This method jointly denoises token values and token positions by coupling discrete token generation with optimal-transport-based position refinement. Formally, generation is modeled as the joint evolution of content variables $x_t$ and position variables $z_t$, where token denoising updates $x_t$ and position refinement updates $z_t$. This joint formulation allows the model to modify both what tokens are generated and where they should be placed, thereby supporting more flexible infilling and variable-position generation.

Standard discrete diffusion models rely on fixed token positions during generation, which prevents them from handling flexible-length or flexible-position text infilling. Discrete Diffusion with Optimal Transport Position Coupling (DDOT) (Zhang et al., 2025) jointly denoises both token values and token positions, enabling dynamic sequence restructuring while preserving relative ordering. During sampling, DDOT alternates between token denoising and position refinement:

1. Predict token distributions $s_\theta(x_t, t)$ and replace masks accordingly.

2. Predict position velocities $v_\theta(z_t, t)$ and update the position variable $z_t$ with Euler steps.

In Appendix Section B.III, we present additional mathematical formulations of DDOT.

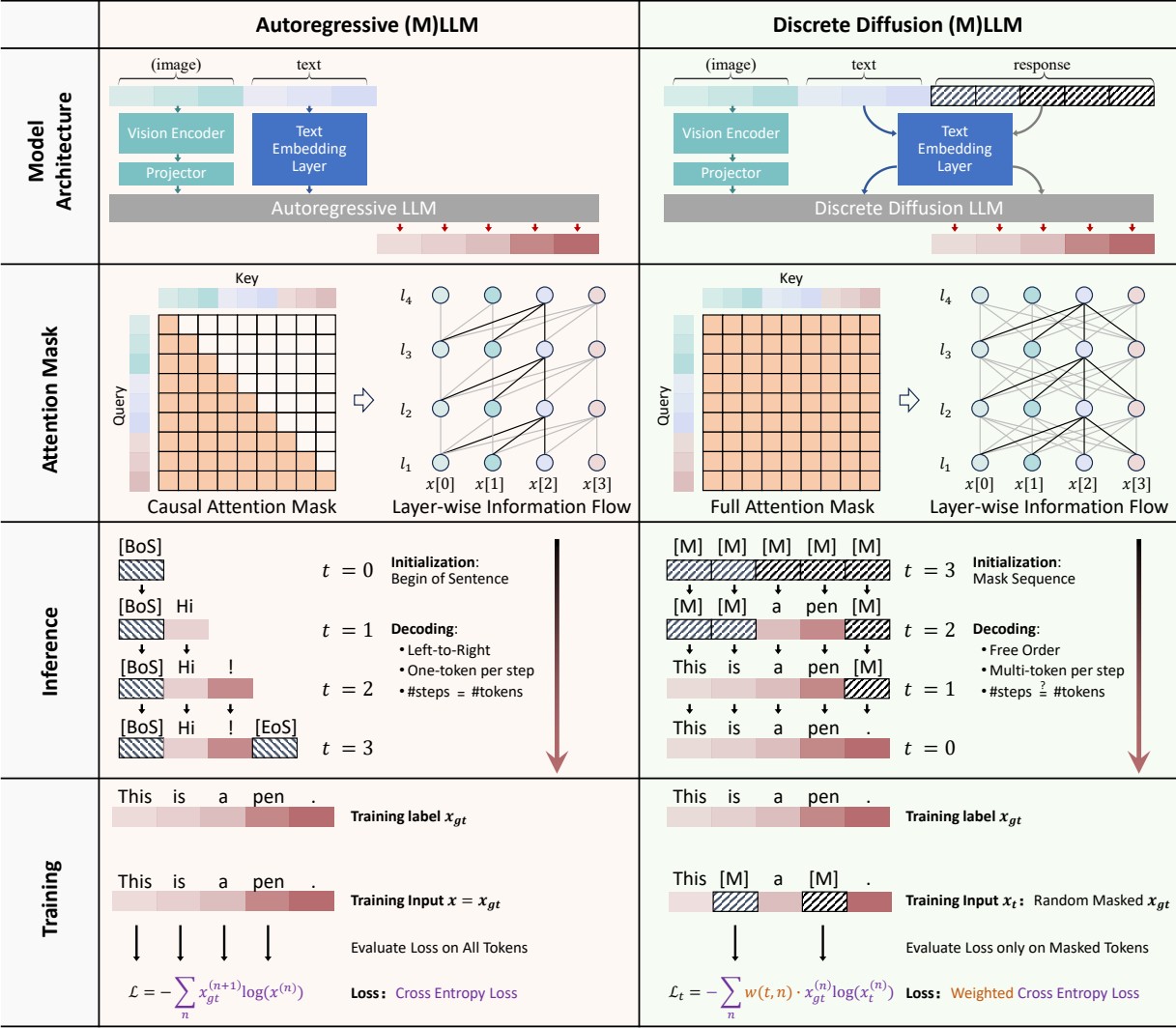

Figure 3: This figure compares autoregressive models and discrete diffusion models from four perspectives. First, regarding model architecture, both models share the same network structure; the key difference lies in their generation mechanisms. In addition to the LLM, both MLLM and dMLLM require an additional vision encoder. In terms of the attention mask, the autoregressive model uses a causal attention mask, whereas the discrete diffusion model adopts a full (bidirectional) attention mask. During inference, the autoregressive model starts from a BoS token and generates tokens one by one from left to right. In contrast, the discrete diffusion model begins with a sequence of mask tokens and denoises all tokens in parallel. At each step, a subset of tokens is selected and replaced with non-mask tokens, continuing until no mask tokens remain. For training, the autoregressive model directly takes the input sequence and applies next-token prediction loss. The discrete diffusion model first randomly masks the input tokens and then computes a weighted cross-entropy loss over the masked positions.

# 4    Representative Models

In this section, we provide a high-level overview of the representative works. In the following sections, we give detailed discussions on the training paradigms and inference-time decoding strategies of the models scaled to sizes comparable to LLMs. An evolutionary diagram of representative dLLM and dMLLM models is shown in Fig. 1.

## 4.1 Discrete Diffusion Models around 1B

Sohl-Dickstein et al. (2015) first introduces a diffusion process over binary variables. This idea is generalized to categorical variables by Hoogeboom et al. (2021), who demonstrates its effectiveness on image generation. Building on these foundations, D3PM (Austin et al., 2021) proposes a more flexible family of noising schedules that extends discrete diffusion to a broader class of discrete spaces (see Sec. 2.1). DiffusionBERT (He et al., 2022) explores training BERT (Devlin et al., 2019) to reverse a discrete diffusion process with an absorbing state, introducing a token-aware noise schedule for the forward pass and methods to embed time-step information into BERT.

Zheng et al. (2023) simplifies the formulation of discrete diffusion process in D3PM and introduces a new model family named Reparameterized Discrete Diffusion Models (RDMs). It reformulate the backward process of discrete diffusion in D3PM (Austin et al., 2021) into a two-stage sampling procedure and yield a greatly simplified training objective and enable more flexible decoding algorithms. MLDM (Sahoo et al., 2024) and MD4 (Shi et al., 2024) further simplified the discrete diffusion model specifically for cases with an absorbing state, also referred to as masked diffusion models. Masked-Diffuse LM (MDLM) (Chen et al., 2023) propose to leverage the inherit linguistic features of texts to encourage the model to recover the text following an easy-first-generation nature, and directly predict the discrete token with cross-entropy loss to stabilize the intermediate diffusion steps. Diffusion-NAT (Zhou et al., 2023) uses the pretrained BART (Lewis et al., 2019) as the language backbone and unifies the inference process of pretrained language models and the denoising process of discrete diffusion models in a non-autoregressive manner, thus the BART model plays the role of the parameterized denoiser in discrete diffusion models. Furthermore, TESS (Mahabadi et al., 2023) leverages a new form of self-conditioning and applies diffusion on the logit simplex instead of the learned embedding space. Plaid (Gulrajani & Hashimoto, 2023) takes the first step towards closing the likelihood gap between autoregressive and diffusion-based language models through SEDD (Lou et al., 2023) generalize the idea of score matching into the discrete spaces by proposing a novel loss named score entropy, which can be integrated seamlessly to build discrete diffusion models and can significantly boost model performance.

For unified models, UniDisc (Swerdlow et al., 2025) is a prior work on unified discrete diffusion model for text and image modalities. Conceptually, UniDisc treats an image and a caption as two token sequences (from discrete codebooks) and denoises them together with a decoder-only Transformer and bidirectional attention.

## 4.2 Large Diffusion Language Models

### 4.2.1 DIFFUSION-LLMs

DIFFUSION-LLMs (Ye et al., 2023) is the first work we identified that scales discrete diffusion language models to 3B and 10B parameters, showing that their performance improves consistently with model size. Its training involves two stages. First, the model is pretrained with a masked LLM objective (similar to BERT (Devlin et al., 2019)) to acquire world knowledge. Next, the pretrained masked LLM is reprogrammed into a dLLM using the RDM loss and training strategy on either specific downstream task datasets or instruction-following datasets. Experiments demonstrate that scaled-up discrete diffusion models, like autoregressive models, possess zero-shot generation, in-context learning, and even reasoning capabilities. Moreover, DIFFUSION-LLMs show that dLLMs can outperform autoregressive models on reasoning tasks requiring implicit planning, such as Path-Finding on Path-Star Graphs (Bachmann & Nagarajan, 2024).

### 4.2.2 LLaDA Series

The LLaDA series represents the pioneering line of discrete diffusion-based alternatives to autoregressive LLMs. LLaDA (Nie et al., 2025b), the first work in this line of research, is a discrete diffusion large language model. It follows the standard masked diffusion model framework, employing a Transformer with bidirectional attention. Its training objective is the variational likelihood bound (ELBO) (Song et al., 2020), rather than the exact log-likelihood. While LLaDA demonstrates strong performance after supervised finetuning, aligning a dLLM with human preferences (akin to RLHF (Kaufmann et al., 2023) for AR models)

remains challenging. LLaDA 1.5 (Zhu et al., 2025a) specifically addresses this by introducing Variance-Reduced Preference Optimization (VRPO) for dLLMs. The main challenge addressed by LLaDA 1.5 is the high variance that arises when estimating log-probabilities with ELBO during reinforcement learning. Based on both theoretical and empirical analysis, LLaDA 1.5 proposes three solutions: increasing the number of Monte Carlo samples, using optimal allocation of samples (many timesteps but one mask per step), and applying antithetic sampling in preference comparisons to reduce variance.

> **Revised Part**
> LLaDA-MoE Zhu et al. (2025b) introduces a masked diffusion language model with a sparse Mixture-of-Experts (MoE) architecture. The model maintains a total capacity of 7 billion parameters but activates only 1.4 billion parameters during inference. Its training pipeline comprises two large pretraining stages on a mixed text corpus (10 trillion tokens each), followed by two annealing stages that refine the model on 500 billion tokens each and increase the rotary positional embedding base and context length from 4k to 8k tokens; finally, supervised fine-tuning on high-quality prompt–answer pairs produces the instruct model. This sparse MoE design allows LLaDA-MoE to surpass dense 8billion-parameter diffusion models such as LLaDA and Dream across knowledge, reasoning, mathematics and coding benchmarks while using far fewer active parameters.

### 4.2.3 DiffuGPT & DiffuLLaMA

DiffuGPT and DiffuLLaMA (Gong et al., 2024) propose converting a pretrained autoregressive Transformer (such as GPT-2 (Radford et al., 2019) or LLaMA (Touvron et al., 2023)) into a dLLM, thereby avoiding the cost of training large models entirely from scratch. Crucially, (Gong et al., 2024) establish theoretical connections between autoregressive next-token prediction and the diffusion denoising objective, enabling alignment between the two paradigms during adaptation. Leveraging the autoregressive model's knowledge as initialization, the diffusion model can be scaled up with significantly less data (under 200 billion tokens, compared to trillions for training from scratch). Experiments span models from 127M to 7B parameters, and the resulting series achieves performance comparable to autoregressive LLMs. Notably, due to their bidirectional nature, DiffuGPT and DiffuLLaMA can perform infilling natively without prompt engineering or token reordering, a challenge for autoregressive models. During inference, they also support a trade-off between generation speed and quality by adjusting the number of diffusion iterations, often requiring fewer refinement steps to achieve fluent outputs than other dLLMs.

### 4.2.4 DiffuCoder

DiffuCoder (Gong et al., 2025) is a 7B-parameter dLLM on a large code corpus, the authors reveal how dLLMs dynamically shift between autoregressive-like and parallel generation patterns. To stabilize preference optimization under diffusion training, they propose coupled-GRPO, a reinforcement learning algorithm that reduces variance via complementary mask sampling, yielding measurable improvements in code generation benchmarks.

### 4.2.5 DREAM Series

DREAM 7B (Ye et al., 2025b) is one of the most powerful open-source dLLMs to date. DREAM 7B achieves performance on par with, or exceeding, autoregressive models of similar size (e.g. it matches LLaMA3 8B (Grattafiori et al., 2024) and Qwen 2.5-7B (Qwen et al., 2025) on many benchmarks). A key to DREAM's success is an optimized training recipe distilled from extensive experiments at smaller scales. Ye et al. (2025b) carefully explores the design choices on a 1B model and identify two especially valuable components: (1) AR weight initialization as in (Gong et al., 2024) and (2) context adaptive noise scheduling as in (Ye et al., 2025a). The AR initialization of DREAM is chosen to be Qwen2.5 (Qwen et al., 2025).

The Dream series has several follow-up works. Dream-Coder 7B (Xie et al., 2025a) introduces a diffusion-based code model, adapted from Qwen2.5-Coder (Hui et al., 2024) and trained on 322B tokens. It supports multiple decoding styles, including sketch-first generation, left-to-right decoding, and interleaved reasoning. DreamOn (Wu et al., 2025d) resolves the fixed-canvas limitation of diffusion decoding by introducing special

tokens (<|expand|>, <|delete|>) for variable-length generation. During inference, a masked token can be predicted as one of these operations: the <|expand|> token splits the current mask into two masks, while the <|delete|> token removes the current mask. This innovation enables true flexible infilling and significantly improves performance on benchmarks such as HumanEval-Infilling.

### 4.2.6 TESS 2

TESS 2 (Tae et al., 2025) is another dLLM that is not only large-scale but also instruction-following and general-purpose. The training recipe for TESS 2 is a culmination of ideas from prior works, such as AR initialization (Gong et al., 2024) and reward guidance (Mahabadi et al., 2023). TESS 2 starts by adapting a powerful AR base model via continued pretraining on the diffusion objective, and then applies thorough instruction-tuning to that adapted model. TESS 2 finds that both the adaptation procedure and the choice of base model are crucial for a good dLLM. For reward guidance, TESS 2 shows that the choice of reward model exhibits some robustness.

### 4.2.7 Seed Diffusion

Seed Diffusion (Preview) (Song et al., 2025b) is a powerful discrete diffusion model primarily designed for code generation, achieving an inference speed of 2,146 tokens per second on H20 GPUs. Seed Diffusion employs two types of data noising: 80% masked language modeling and 20% random deletion, insertion, or substitution, with edit distance used to measure differences between sequences. The training loss combines a mask prediction term and an overall reconstruction term. To enable better sequential generation, Seed Diffusion leverages a pretrained diffusion model to generate partial trajectory data, which is then filtered by overall log-likelihood to construct a refined trajectory dataset. To further reduce decoding steps, it introduces a reinforcement learning paradigm that maximizes the edit distance between consecutive steps, encouraging the model to decode as many tokens as possible per step while maintaining correctness.

**Revised Part**

### 4.3 UltraLLaDA

UltraLLaDA He et al. (2025a) studies how to extend the context window of diffusion LLMs without retraining from scratch. The authors introduce a "diffusion-aware NTK" extension of rotary positional embeddings and evaluate different long-context masking strategies. Post-training the LLaDA-8B base model with these modifications produces UltraLLaDA, a diffusion LLM capable of handling up to 128k tokens. The paper reports that this model significantly outperforms training-free baselines on long-context benchmarks, demonstrating stable perplexity and high task accuracy across extended context lengths. In addition to the modified embeddings, the authors investigate adaptive attention masking and end-of-document concatenation to reduce cross-document interference. Their experiments show that UltraLLaDA consistently outperforms LongLLaDA and the original LLaDA base model on long-context retrieval and language modeling tasks, highlighting the effectiveness of lightweight post-training for context extension.

### 4.4 Large Diffusion Multimodal Models

### 4.4.1 Dimple

Dimple (Yu et al., 2025) is one of the first Discrete Diffusion Multimodal Large Language Models (dMLLMs). Its base architecture (vision encoder + transformer LLM) resembles existing vision-language models (e.g. Qwen-VL (Bai et al., 2023b), LLaVA (Liu et al., 2023b;a; 2024)). One of the Dimple's key innovations is its two-stage hybrid training. In Stage 1, with the weights of Dream-7B (Ye et al., 2025b) as an initialization, it is fine-tuned autoregressively on vision-language instruction data (for alignment and instruction following). In Stage 2, it is then further fine-tuned with a discrete diffusion objective. This hybrid approach is devised because pure diffusion training is found to be unstable (leading to length bias and performance drop). By warming up with autoregressive training first, Dimple-7B achieves stable training and eventually surpasses the fully-autoregressive models.

During inference, Dimple introduces a *confident decoding* strategy for efficiency: the model dynamically chooses how many tokens to fill in at each step based on model confidence (see Sec. 6.1). Empirically, this reduces the number of iterations to about $\frac{\text{response length}}{3}$. Dimple also re-implements an autoregressive *prefilling* trick: by filling in some tokens from the existing context, it speeds up inference by about $1.5\times$–$7\times$ with minimal impact on quality. Under the same training budget and dataset as LLaVA-NEXT (Liu et al., 2024), Dimple-7B achieves higher aggregate scores on multimodal benchmarks than LLaVA-NEXT-7B. This result shows that with a proper hybrid training recipe, a discrete dMLLM can match or exceed strong autoregressive baselines on vision-language tasks.

### 4.4.2 LaViDa

LaViDa (Li et al., 2025e) is among the first models to extend discrete diffusion into the multimodal large language model (dMLLM) setting. It consists of a vision encoder (e.g. SigLIP-400M (Zhai et al., 2023)) and a discrete diffusion Transformer. Its language model is a standard discrete dLLM (either LLaDA-8B or Dream-7B). LaViDa's key innovation is complementary masking in training: for each training sample, two distinct mask patterns are created so that each token is masked in one of the two versions. This ensures that even short or rare answer tokens (e.g. object names in vision tasks) contribute to the loss and all tokens are learned efficiently, improving alignment between the visual encoder and the language model. During inference, LaViDa employs a special Prefix-DLM (Li et al., 2025e) attention mask so that the encoded image and prompt tokens can be cached and reused. The model also uses a timestep-shifting schedule to improve sample quality.

### 4.4.3 LLaDA-V

LLaDA-V (You et al., 2025) is one of the pioneering efforts in developing Discrete Diffusion Multimodal Large Language Models (dMLLMs). Built on LLaDA (Nie et al., 2025b), LLaDA-V undergoes three training phases. In the first stage, language–image alignment, the MLP projector is trained to align visual features with LLaDA's word embeddings. The second stage, visual instruction tuning, fine-tunes the model on large-scale multimodal instruction data to build instruction-following abilities. Finally, the third stage, multimodal reasoning enhancement, focuses on strengthening reasoning capabilities by training on reasoning-focused multimodal QA data with detailed reasoning chains, and further balancing direct answering and explicit reasoning through mixed training with "no_think" and "think" tags.

## 4.5 Large Unified Model

### 4.5.1 MMaDA

MMaDA (Yang et al., 2025b) employs a unified diffusion architecture with a shared probabilistic formulation across image and text modalities. It uses a single diffusion-based transformer for all data types (text, images, etc.), rather than separate encoders for each modality. During training, MMaDA is fine-tuned with a *mixed long chain-of-thought* strategy. Reasoning steps from both text and vision tasks are converted into a unified CoT format so that the model learns aligned reasoning across modalities. For example, the rationale for answering a visual question is interleaved into the textual input. This CoT alignment provides a form of cold-start for the final reinforcement learning (RL) stage, allowing complex multi-step reasoning from the outset. Finally, MMaDA proposes a unified policy-gradient-based RL algorithm named *UniGRPO*. By incorporating diversified reward modeling, UniGRPO unifies the post-training across both reasoning and generation tasks, improving the performance.

### 4.5.2 FUDOKI

FUDOKI (Wang et al., 2025c) is a unified multimodal model built on discrete flow matching (Shaul et al., 2024). It uses a metric-induced probability path with kinetic-optimal velocities (Shaul et al., 2024), which significantly improves over simple masking by enabling continuous self-correction during generation. For efficiency, FUDOKI is initialized from a pretrained AR-based multimodal LLM (Janus-1.5B (Wu et al., 2025a)) and then transferred to the discrete flow matching framework. For input modalities, text is tokenized

Table 1: Comparison of dLLM Models on Benchmarks

| Model | Params(B) | ARC-C | AlignBench | Arena-Hard | GPQA | GSM8K | Hellaswag | HumanEval | IFEval | MBPP | MMLU | MMLU-pro | MTBench | Math | PIQA | WinoG |
|---|---|---|---|---|---|---|---|---|---|---|---|---|---|---|---|---|
| LLaDA | 8 | 88.5 | 5.4 | 10 | 33.3 | 69.4 | 75.6 | 49.4 | 62.2 | 41 | 65.5 | 37 | 7.2 | 31.9 | | |
| LLaDA 1.5 | 8 | | 5.9 | 14.3 | 36.9 | 83.3 | | 52.4 | 66.2 | 42.8 | | | 7.3 | 42.6 | | |
| DiffuLLaMA | 7 | | | | | 63.1 | 58.7 | | | | | | | | 63.3 | 56.4 |
| DiffuCoder | 7 | | | | | | | 73.2 | | 78.6 | | | | | | |
| Dream | 7 | 59.8 | | | 36.6 | 77.2 | 73.3 | 57.9 | | | 56.2 | 59.5 | | 39.6 | 75.8 | 74.5 |
| Dream_Instruct | 7 | | | | 33 | 81 | | 55.5 | 62.5 | 58.8 | 67 | 43.3 | | 39.2 | | |

Table 2: Comparison of dMLLM Models on Benchmarks

| Model | Params(B) | AI2D | ChartQA | DocVQA | InfoVQA | MMBench_en_test | MME_cog | MME_perc | MMMU | MathVerse | MathVision | MathVista | ScienceQA | TextVQA | VQAv2 |
|---|---|---|---|---|---|---|---|---|---|---|---|---|---|---|---|
| Dimple | 7 | 74.4 | 63.4 | | | 75.6 | 432 | 1514 | 45.2 | | | 42.3 | 77.1 | 61.6 | |
| LaViDa-L | 8 | 70 | 64.6 | 59 | 34.2 | 70.5 | 341 | 1366 | 43.3 | 27.2 | 20.4 | 44.8 | 80.2 | 56.3 | 72.2 |
| LaViDa-D | 7 | 69 | 61 | 56.1 | 36.2 | 73.8 | 378 | 1463 | 42.6 | 24.1 | 19.4 | 42.1 | 81.4 | 57.1 | 75.2 |
| LLaDA-V | 8 | | | | | 82.9 | 491 | 1507 | 48.6 | 28.5 | | 59.7 | | | |

normally, while images are handled by separate pipelines: a semantic encoder (SigLIP (Zhai et al., 2023)) extracts features for image understanding, and a pixel encoder/decoder (Sun et al., 2024) converts images to/from discrete image tokens for generation. At the output stage, FUDOKI has two output heads—one predicting text tokens and one predicting image tokens—and selects the appropriate head depending on the target modality during inference.

### 4.5.3   Muddit

Muddit (Shi et al., 2025) is another unified model that uses purely discrete diffusion to handle text and images under one framework. The architecture of Muddit comprises a single multimodal diffusion transformer (MM-DiT) (Shi et al., 2025), plus encoders/decoders for each modality. The MM-DiT follows a dual-/single-stream design (similar to FLUX (Labs, 2024)) and is initialized from the pretrained Meissonic (Bai et al., 2024). Inputs are quantized into a shared token space: images are encoded by a pretrained VQ-VAE (Van Den Oord et al., 2017) into discrete codebook indices, and text is encoded by a CLIP text encoder (Radford et al., 2021). During training and inference, the MM-DiT predicts masked tokens in this joint space. A linear head maps the predicted tokens to actual text tokens for text output, while the VQ-VAE decoder reconstructs pixels from image tokens.

Revised Part

### 4.6   D-DiT

Dual diffusion Transformer (D-DiT) Li et al. (2025h) for unified image generation and understanding introduces a dual-branch diffusion model that unifies continuous image diffusion and discrete masked text diffusion under a single transformer. The model employs a cross-modal maximum-likelihood framework that jointly trains the conditional likelihoods of images and text with a single loss, enabling end-to-end tasks such as text-to-image generation, image captioning and visual question answering. Experiments report improvements in image quality and multimodal tasks compared with baseline models, while noting that performance can degrade on complex scenes and that the model still requires a preset sequence length.

### 4.7   Benchmark Evaluation

Tab. 1 and 2 compare the performance of current discrete diffusion language models and multimodal models across different benchmarks. The data are collected from the original papers with differing training setups, corpora, tokenizers, decoding budgets and post-training recipes, and therefore cannot be used for direct apples-to-apples comparisons. These results provide indicative evidence of current progress rather than definitive conclusions.

# 5 Training Techniques

In this section, we summarize the techniques employed during the training of diffusion language models (dLLMs) and diffusion multimodal language models (dMLLMs).

## 5.1 Challenges

First, we summarize several challenges encountered in the training of discrete diffusion models. These challenges stem from low corpus utilization and high variance due to stochastic masking.

### 5.1.1 Low Corpus Utilization

Unlike autoregressive training, where each token in the answer sequence contributes to the learning signal, discrete diffusion training applies supervision only to a randomly selected subset of tokens at each time step. Given an input sequence $x$ of length $L = L_{\mathrm{prompt}} + L_{\mathrm{answer}}$, diffusion training samples a timestep $t \in [1, T]$ and computes loss only over the masked tokens at that timestep. This leads to sparse supervision across training samples, resulting in inefficient utilization of the corpus.

### 5.1.2 Random Sampling of Time Index

In diffusion training, the time index $t$ is randomly sampled for each training instance. As a result, only a single generation step is supervised per sample, while the decoding process at inference time typically involves multiple time steps. This mismatch introduces a coverage gap between training and inference: although decoding requires refinement over many steps, training provides gradient signals for only one of those steps per instance.

## 5.2 Initialization Techniques

To address the inefficiencies and instabilities in training dLLMs and dMLLMs, several works adopt advanced initialization strategies that convert the full diffusion training process into a fine-tuning task. This approach accelerates convergence and enhances final model performance.

Because the diffusion generation process can be interpreted as a multi-step masked language modeling (MLM) procedure. He et al. (2022) initializes diffusion models from pretrained BERT. Moreover, Gong et al. (2024); Ye et al. (2025b) explored direct adaptation from autoregressive language models by aligning the training objectives of the two paradigms. A key technique enabling this transition is the *shift operation*. In standard diffusion training, the model predicts the original token $x_0$ from its corrupted version $x_t$ at each timestep. However, this formulation differs from AR training, where each hidden state $h_i$ is trained to predict the next token $x_{i+1}$ in a left-to-right fashion. To bridge this gap, Gong et al. (2024); Ye et al. (2025b) propose shifting the output logits of the diffusion model by one position, such that the model's prediction at position $i$ corresponds to token $x_{i+1}$. This allows the diffusion model to be initialized with pretrained autoregressive models.

Another approach similar to initialization is Autoregressive-then-Diffusion Training, making the Diffusion Training as post-training of autoregressive training. Dimple (Yu et al., 2025) uses an *autoregressive-then-diffusion* training approach, demonstrating notable performance improvements for DMLLM. The Dimple training pipeline is divided into two distinct phases:

- **Phase I: Autoregressive Training.** In the first phase, Dimple is treated as an autoregressive model using the causal attention mask and next-token prediction loss.

- **Phase II: Diffusion Fine-tuning.** After autoregressive training, Dimple is treated as an diffusion model using the full attention masks and timestep-dependent masked language modeling losses.

### 5.3 Complementary Masking Technique

To improve the utilization of the corpus, in Li et al. (2025e), to ensure that all tokens participate in training, complementary masking constructs two complementary masked versions of each input sequence: $X_t$ and $X_t^C$. $X_t$ and $X_t^C$ have non-overlapping masked spans. For example, consider the sentence:

"The answer is dog."

One masked variant might be:

"The [M] [M] dog."

and its complement:

"[M] answer is [M]."

This setup ensures that all tokens are eventually masked and optimized over the course of training.

### 5.4 Masking Scheduling Technique

*Masking scheduling* governs the corruption process in the forward diffusion formulation. Specifically, the schedule defines the corruption level $\alpha_t$ at each timestep $t$, thereby determining the proportion of tokens masked during training. An effective schedule balances learning stability and generation quality by controlling the signal-to-noise ratio across timesteps.

#### 5.4.1 Uniform Masking Scheduling

Masking scheduling can either apply the same scheduling function to all tokens, referred to as uniform masking scheduling, or assign different scheduling functions to individual tokens, referred to as token-wise masking scheduling. We first introduce two commonly used uniform masking scheduling methods.

Given a timestep $t \in [0, 1]$, the forward process defines the corruption as:

$$q(x_t \mid x_0) = \alpha_t x_0 + (1 - \alpha_t)m, \tag{33}$$

where $m$ is the one-hot [MASK] token. The loss at each step is reweighted according to:

$$w_t = \frac{\alpha_t'}{1 - \alpha_t}, \tag{34}$$

where $\alpha_t' = \frac{d\alpha_t}{dt}$ is the derivative of $\alpha_t$ with respect to time. Linear Schedule and Cosine Schedule are two commonly-used scheduling strategies are as follows. Their corresponding schedule functions are plotted in Fig. 4.

- **Linear Schedule (Austin et al., 2021):**

$$\alpha_t = 1 - t, \quad w_t = -\frac{1}{t}. \tag{35}$$

- **Cosine Schedule (Chang et al., 2022):**

$$\alpha_t = 1 - \cos\left(\frac{\pi}{2}(1 - t)\right), \quad w_t = -\frac{\pi}{2}\tan\left(\frac{\pi}{2}(1 - t)\right). \tag{36}$$

The theoretical analyses of the optimal masking scheduling function remain scarce. One pioneering work (Zhang, 2025) theoretically proves the optimality of cosine scheduling under the Fisher–Rao geometry. In Appendix Sec.C.I, we include another two uniform scheduling functions.

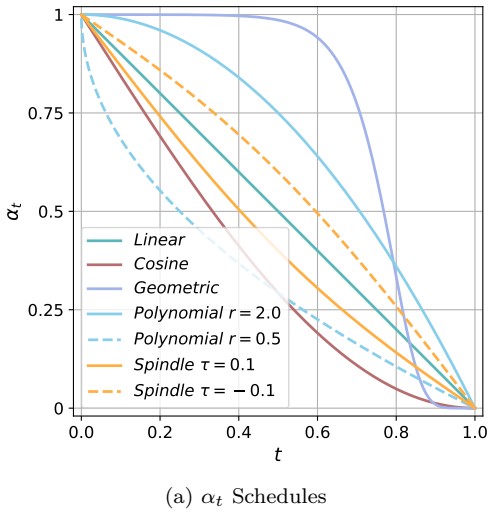
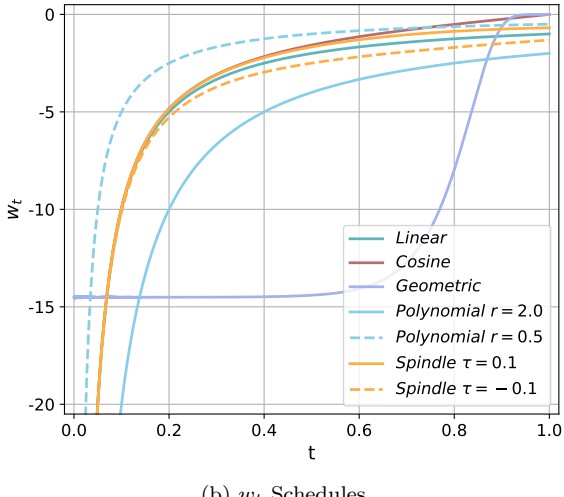

(a) $\alpha_t$ Schedules  (b) $w_t$ Schedules

Figure 4: Different schedules for $\alpha_t$ and $w_t$. To unify notation, we also transformed the spindle schedule from the discrete-time format used in the original paper to a continuous-time format. The revised formulation is as follows: $\alpha_t = 1 - t - \tau \sin(\pi t)$ and $w_t = -\frac{1 + \tau \pi \cos(\pi t)}{t + \tau \sin(\pi t)}$, where $\tau$ corresponds to the original $\lambda \tilde{H}$.

### 5.4.2 Token-wise Masking Scheduling

Uniform masking scheduling uses the same scheduling for all tokens, which ignores the inherent variation in informativeness among different tokens. He et al. (2022) introduces a *token-wise masking schedule*, which defines a *Spindle-shaped Noise Schedule*, where the entropy-aware corruption pattern resembles a spindle curve: more informative tokens are masked earlier and less informative ones later. This design ensures that low-entropy, easier-to-predict tokens are decoded first, leading to more coherent and stable sequence generation. The math of the Spindle-shaped Noise Schedule is discussed in Appendix Sec.C.I.

### 5.4.3 Block-wise Masking Scheduling

For reasoning tasks, the prevailing approach adopts a blockwise semi-autoregressive decoding strategy. However, previous random masking across the full response, does not reflect this procedure. Let the response be partitioned into $M$ contiguous blocks $b^{(1)}, \ldots, b^{(M)}$ of size $B$. For a given active block $a$, Blockwise SFT (Sun et al., 2025a) defines:

- *Prefix* $I_{\text{prefix}}^{(a)}$: tokens before block $a$, kept clean and fixed.

- *Active block* $I^{(a)}$: tokens of block $a$, subject to stochastic masking.

- *Suffix* $I_{\text{suffix}}^{(a)}$: tokens after block $a$, fully hidden.

In other words, the masking rule is:

$$m_i = \begin{cases} 0, & i \in I_{\text{prefix}}^{(a)} \quad \text{(clean prefix)}, \\ \text{Bernoulli}(\pi), & i \in I^{(a)} \quad \text{(masked active block)}, \\ 1, & i \in I_{\text{suffix}}^{(a)} \quad \text{(fully hidden suffix)}, \end{cases}$$

with $\pi \sim \text{Uniform}(10^{-3}, 1)$. Loss and gradient updates are computed only for the active block.

## 5.5 Reweighting Technique

Multi-Granularity Diffusion Modeling (MGDM) (Ye et al., 2025a) introduces an additional token-level reweighting factor $v(x_{t,n})$ in the loss function, yielding:

$$\mathcal{L}_{\text{MGDM}} = \sum_{n=1}^{N} \sum_{t=1}^{T} w(t) \cdot v(x_{t,n}) \cdot \ell(x_0, x_t, n; \theta), \tag{37}$$

where $\ell(x_0, x_t, n; \theta)$ is the CE loss on the n-th token, and the adaptive token-level weight is defined as:

$$v(x_{t,n}) = \alpha(1 - \exp(-\ell(x_0, x_t, n; \theta)))^{\beta}, \tag{38}$$

with hyperparameters $\alpha > 0$, $\beta > 0$. This reweighting assigns larger weights to harder tokens (i.e., those with higher loss), thereby prioritizing difficult subgoals during training and accelerating convergence.

Revised Part

### 5.6 Distillation

To enable efficient few-step or even single-step generation while preserving performance, Di4C (Hayakawa et al., 2024) introduces a distillation strategy that compresses a multi-step discrete diffusion language model (dLLM) into a reduced-step counterpart. It employs two loss functions: *distillation loss* and *consistency loss*, grounded in distributional matching and multi-path coherence. The distillation loss transfers probabilistic knowledge from a teacher model performing multi-step denoising to a student model designed for fewer-step generation, while the consistency loss enforces stable behavior across predictions from different intermediate noise levels. Detailed formulations of these losses are provided in Appendix Sec. C.II.

Learnable Sampler Distillation aligns the intermediate score trajectory of a few-step student sampler with that of a high-quality teacher and extends to non-uniform time schedules (LSD+) to further improve sampling quality Fu et al. (2025a). Ultra-Fast Language Generation via Discrete Diffusion Divergence Instruct minimizes an integral KL divergence between student and teacher while matching intermediate states and applying grouped reward normalization; it achieves up to $64\times$ speedup with negligible entropy loss Zheng et al. (2026). Distillation of Discrete Diffusion by Exact Conditional Distribution Matching decomposes the reverse conditional distribution and matches conditional components to train one-step and few-step students Gao & Sun (2025). C4DLM introduces discrete-space consistency distillation and confidence-adaptive decoding to produce trajectory-invariant students and allocate computation based on token confidence, yielding a $5\times$ speedup on GSM8K and an average $3\times$ speedup across tasks Liang et al. (2026). dUltra formulates distillation as an on-policy reinforcement learning problem and jointly trains the diffusion LLM with an unmasking planner to achieve improved accuracy–efficiency trade-offs Chen et al. (2026). T3D employs trajectory self-distillation with direct discriminative optimization to narrow the gap between few-step and full-step decoding Zhang et al. (2026d). Discrete Moment Matching Distillation extends moment matching to discrete spaces, maintaining high quality and diversity while occasionally enabling the student to outperform the teacher Hoogeboom et al. (2026). IDLM generalizes inverse distillation to discrete diffusion models, establishes the uniqueness of the optimization, and introduces gradient-stable relaxations; experiments demonstrate a 4–64$\times$ reduction in inference steps without degrading perplexity Li et al. (2026a).

### 5.7 Reinforcement Learning

Reinforcement learning has been extensively applied to discrete diffusion language models (Yang et al., 2025b; Zhu et al., 2025a; Huang et al., 2025b; Tang et al., 2025b; Han et al., 2025; He et al., 2025b; Wang et al., 2025d). Reinforcement learning for diffusion language models largely inherits the paradigm from autoregressive models; however, it also presents several unique challenges, such as the estimation of likelihood. The following provides a brief summary of several reinforcement learning techniques, with detailed descriptions presented in Appendix Sec. C.IV.

Revised Part

Diffusion-based GRPO (UniGRPO) extends clipped policy optimization by integrating structured nois­ing and KL-regularized surrogate rewards (Yang et al., 2025b). Variance-Reduced Preference Opti­mization (VRPO) improves stability by replacing intractable log-likelihoods in DPO with ELBO es­timates and applying advanced variance reduction techniques (Zhu et al., 2025a). To address reward propagation across trajectories, Stepwise Decomposition Preference Optimization (SDPO) reformulates alignment into tractable per-step KL-regularized objectives (Han et al., 2025). Weighted Policy Op­timization (wd1) recasts the objective as a weighted likelihood maximization, where weights derived from centered rewards ensure better sample efficiency (Tang et al., 2025b). Finally, Diffusion Chain of Lateral Thought (DCoLT) introduces a reinforcement-learned Unmask Policy Module that adap­tively controls the token unmasking order during generation (Huang et al., 2025b). Together, these methods highlight complementary directions for improving stability, efficiency, and controllability in diffusion-based reinforcement learning. Score Entropy Policy Optimization (SEPO) Zekri & Boullé (2025) is a policy-gradient algorithm designed specifically for discrete diffusion models. It introduces a clipped-ratio loss and self-normalized importance sampling to handle non-differentiable rewards while maintaining low variance, enabling fine-tuning of diffusion LLMs for conditional or unconditional gen­eration tasks with improved performance. Inpainting-Guided Policy Optimization (IGPO) Zhao et al. (2025b) addresses the unstable gradients encountered when training diffusion LLMs with reinforcement learning. It guides exploration by inserting partial ground-truth reasoning traces during sampling and uses entropy-based filtering to preserve self-generated traces, thereby restoring meaningful gradients and substantially improving performance on math reasoning benchmarks. Sandwiched policy gradient (SPG) Wang et al. (2026) for masked diffusion language models tackles the difficulty of applying rein­forcement learning to diffusion LLMs by jointly optimizing tractable lower and upper bounds on the intractable log-likelihood. SPG maximizes a lower bound on the likelihood for high-reward sequences and minimizes an upper bound for low-reward sequences, forming a "sandwiched" objective that yields more robust policy gradients. A block-wise masking strategy is used to estimate these bounds via Monte Carlo, reducing gradient variance and improving optimization stability. Step-Aware Policy Optimiza­tion (SAPO) Xie et al. (2026) introduces a process-based reward that encourages incremental progress through the denoising steps. By aligning the agent's reward with a latent reasoning hierarchy, SAPO guides the model to learn structured reasoning paths and demonstrates strong performance on reasoning benchmarks. Boundary-Guided Policy Optimization (BGPO) Lin et al. (2025b) tackles the high mem­ory overhead of reinforcement learning in diffusion models. It maximizes a linear lower bound on the evidence lower bound objective, which is equivalent to the original objective for on-policy training. This allows gradient accumulation over large Monte Carlo sample sizes while maintaining constant memory usage, enabling more accurate likelihood approximation and leading to improved results on math, code and planning tasks.

# 6 Inference Techniques

In this section, we summarize the techniques involved in the inference phase of dLLMs and dMLLMs. These techniques affect both performance and efficiency, typically requiring a trade-off between the two. Ideally, the goal is to improve decoding efficiency without compromising performance.

## 6.1 Unmasking Techniques

In dLLMs and dMLLMs, the model predicts all the response tokens at each step. However, only a subset of the masked tokens are selected to be unmasked at every iteration, while the remainder remain masked. The main challenges are deciding which and how many tokens to unmask per iteration. Figure 5 provides a detailed illustration of the unmasking strategies. In this section, we discuss each category in detail and describe the specific unmasking strategies proposed in each work.

### 6.1.1 Discrete-Time Unmasking

Random Unmasking

The simplest strategy is to randomly select $s_t$ masked tokens to unmask at step $t$. The value of $s_t$ can be fixed across steps or controlled by a scheduling function as discussed in the training techniques, such as cosine scheduling (You et al., 2025).

Metric-Based Unmasking

Rather than relying on random selection, metric-based strategies assign a metric value to each token prediction and select tokens to be unmasked based on the metric value.

Let $p^{(i)} \in \mathbb{R}^K$ be the predicted probability distribution over the vocabulary for the $i$-th token, where $K$ is the vocabulary size. The following are some commonly used metrics.

- **Maximum Probability (Confidence)** (Ye et al., 2025b; Kim et al., 2025b):

$$c^{(i)} = \max(p^{(i)}), \tag{39}$$

  indicating the model's certainty about the most likely token. (Wu et al., 2025b) provides a theoretical analysis of the equivalence between the parallel decoding using confidence and sequential decoding using confidence.

- **Margin** (Ye et al., 2025b; Kim et al., 2025b):

$$c^{(i)} = p^{(i)}_{\text{top1}} - p^{(i)}_{\text{top2}}, \tag{40}$$

  where $p^{(i)}_{\text{top1}}$ and $p^{(i)}_{\text{top2}}$ are the first and second highest probabilities, respectively. This measures how dominant the top prediction is.

- **Negative Entropy** (Ye et al., 2025b; Kim et al., 2025b):

$$c^{(i)} = \sum_{k=1}^{K} p^{(i)}_k \log(p_i + \epsilon), \tag{41}$$

  with a small constant $\epsilon$ for numerical stability. This captures the peakedness of the distribution.

- In **EB-Sampler** (Ben-Hamu et al., 2025), instead of performing filtering at the token level, the metric is defined over a set of tokens, aiming to directly decide on unmasking a subset $U$ of tokens.

$$c^{(U)} = \sum_{l \in U} H(p^{(l)}) - \max_{l \in U} H(p^{(l)}) \tag{42}$$

  with $H(p^{(l)})$ is the entropy of $p^{(l)}$.

- In **PC-Sampler.** (Huang et al., 2025a), the metric is defined as

$$c^{(i)} = w^{(i)} \cdot p^{(i)}_{x^{(i)}} \cdot [-\log p_{\mathcal{D}}(x^{(i)})], \quad w^{(i)} = e^{-\lambda \cdot i}, \tag{43}$$

  where $x^{(i)}$ is the predicted token at position $i$. In this metric, $w^{(i)}$ controls the degree of left-to-right decoding by applying an exponentially decaying weight based on token positions. The parameter $\lambda$ serves as a hyperparameter. $p^{(i)}_{x^{(i)}}$ denotes the model's predicted probability for the current token, reflecting its confidence. $p_{\mathcal{D}}$ represents the token frequency distribution estimated from a publicly available corpus $p_{\mathcal{D}}$, used to downweight trivial words such as "the" and "a".

Selection Policies

After obtaining the metric value for each token, the diffusion model performs selection based on different policies.

- **Top-$s_t$ Strategy (Austin et al., 2021):** Select the $s_t$ tokens with the highest confidence scores for unmasking. The value of $s_t$ follows the same scheduling principles as in random unmasking.

- **Confident Decoding:** As introduced in Dimple (Yu et al., 2025), this strategy dynamically selects the number of tokens to unmask based on a fixed confidence threshold $\gamma \in (0, 1)$. The motivation of this approach is that decoding should adapt to the semantic structure of the text: some steps may allow many tokens to be confidently predicted, while others may necessitate more caution. Thus, the number of decoded token should be adaptively adjusted at each step. At step $t$, the set of positions to decode is defined as:

$$\mathcal{I}_t = \{i \mid c_t^{(i)} \geq \gamma\}, \tag{44}$$

  where $c_t^{(i)}$ is the confidence score at position $i$. If $\mathcal{I}_t$ is non-empty, all tokens in $\mathcal{I}_t$ are unmasked. Otherwise, only the token with the highest confidence is selected. This approach enables:

  - decoding multiple tokens in parallel when the model is confident, improving efficiency;
  - avoiding low-confidence predictions, preserving generation quality.

- **Block-wise Unmasking:** Block-wise semi-autoregressive decoding strategy (Nie et al., 2025b) divides the full response into multiple blocks, similar to block diffusion. During each forward pass, predictions are generated for all blocks simultaneously. However, the unmasking of tokens follows a left-to-right, block-by-block order. Tokens in the next block are only allowed to be unmasked once all tokens in the previous block have been unmasked.

- **Dilated Unmasking Schedule (DUS)** (Luxembourg et al., 2025): DUS considers the relative distances between tokens during decoding and requires the distances among decoded tokens to decrease gradually from large to small. Specifically, let $\mathcal{U}_t$ denote the selected tokens at time $t$, DUS selects equally spaced tokens as following:

$$\mathcal{U}_0 = \varnothing, \tag{45}$$
$$\mathcal{P}_t = \{k \mid (k - 1) \bmod s_t = 0\}, \tag{46}$$
$$\mathcal{U}_t = \mathcal{U}_{t-1} \cup \mathcal{P}_t, \tag{47}$$

  where $s_t$ is a dilation coefficient that decreases over time.

## 6.2 Remasking Techniques

For masked discrete diffusion model, once a token is unmasked, it remains unchanged in subsequent steps. This static behavior limits the model's capacity to revise or refine earlier predictions. To address this, the remasking technique reintroduces masked tokens at previously unmasked positions, enabling iterative refinement of generated outputs.

### 6.2.1 Remasking in General Masked Diffusion Models.

Wang et al. (2025b) formulates the reversal diffusion process with remasking as

$$q_\sigma(\mathbf{z}_s \mid \mathbf{z}_t, \mathbf{x}) = \begin{cases} \mathrm{Cat}(\mathbf{z}_s; (1 - \sigma_t)\mathbf{x} + \sigma_t \mathbf{m}), & \mathbf{z}_t \neq \mathbf{m}, \\ \mathrm{Cat}\left(\mathbf{z}_s; \frac{\alpha_s - (1 - \sigma_t)\alpha_t}{1 - \alpha_t}\mathbf{x} + \frac{1 - \alpha_s - \sigma_t \alpha_t}{1 - \alpha_t}\mathbf{m}\right), & \mathbf{z}_t = \mathbf{m}, \end{cases} \tag{48}$$

where $\sigma_t$ is used to control the ratio of remasked tokens. When $\mathbf{z}_t \neq \mathbf{m}$, the token has already been decoded. The model samples the next token $\mathbf{z}_s$ from a distribution that mixes the input $\mathbf{x}$ and the mask token $\mathbf{m}$, controlled by $\sigma_t$. This enables remasking by reintroducing uncertainty into already decoded tokens. When $\mathbf{z}_t = \mathbf{m}$, the token is still masked. The sampling distribution is a weighted combination of $\mathbf{x}$ and $\mathbf{m}$, adjusted by both $\alpha_t$ and $\sigma_t$, allowing flexible control over how much information from the input or the mask dominates.

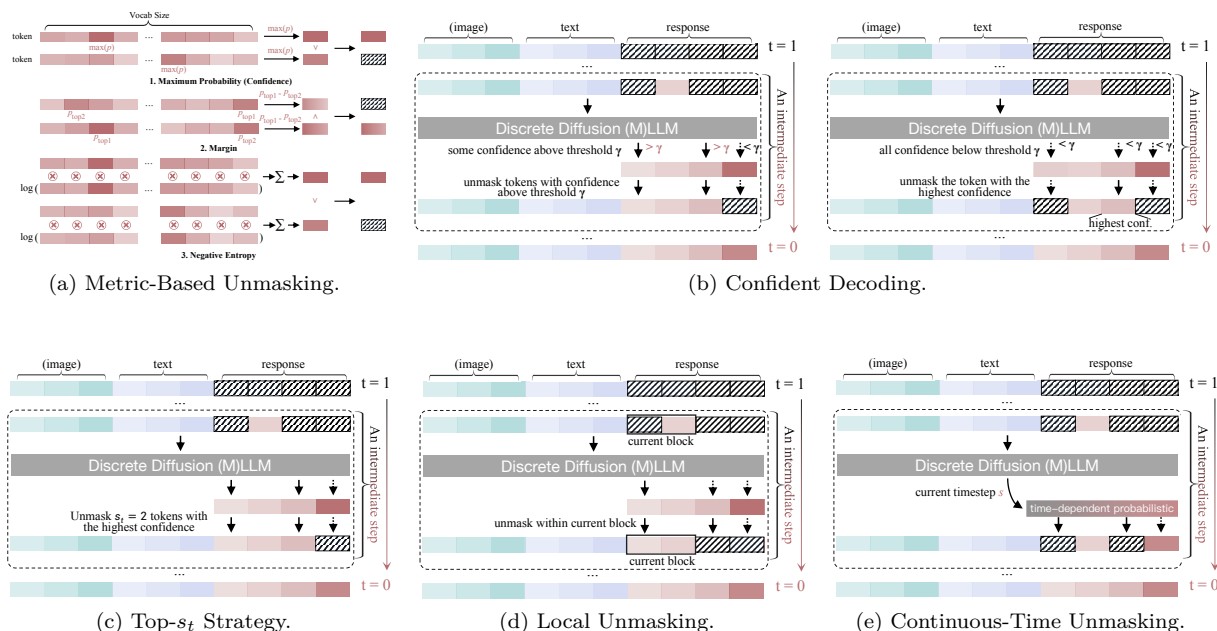

Figure 5: **Unmasking strategies.** We divide the unmasking strategies used in dLLMs and dMLLMs into two categories: Discrete-Time Unmasking (a,b,c,d) and Continuous-Time Unmasking (e). In discrete-time unmasking, besides random unmasking, there are other two unmasking strategies: Metric-Based Unmasking (Maximum Probability (Confidence), Margin and Negative Entropy, see (a)) and Selection Policies (Top-$s_t$ Strategy (see (c)), Confident Decoding (see (b)), and Local Unmasking (see (d))).

### 6.2.2 Wide-In Narrow-Out

Under the block-wise decoding setting, Hong et al. (2025) introduces an alternative verification-based re-masking approach. Specifically, A appends an additional set of shadow tokens after the current response, with the number of shadow tokens equal to the current block size and each shadow token corresponding to one token in the block. By adjusting the attention mask, the original token sequence is prevented from attending to the shadow tokens, while each shadow token can attend to all tokens except its corresponding token in the current block. Consequently, the shadow tokens verify the decoded tokens without interfering with the decoding process of the original response sequence. If some decoded tokens are found to have low confidence, they are replaced with mask tokens.

### 6.3 Prefilling and Caching Techniques

Prefilling and Key-Value Cache (KV-Cache) are standard inference acceleration techniques widely adopted in autoregressive language models. Intuitively, Prefilling and KV-Cache avoids redundant computation by storing the key and value representations from previous decoding steps, enabling the model to reuse them instead of recalculating at each new time step. In autoregressive models, the use of causal attention masks ensures that caching is theoretically lossless. In contrast, dLLMs and dMLLMs employ full (bidirectional) attention mechanisms, wherein each token can attend to all other positions. As a result, even tokens that have already been decoded and unmasked may have their key and value vectors influenced by updates to other tokens during subsequent diffusion iterations. Thus, caching in dLLM and dMLLM are not theoretically lossless.

For dLLMs, dKV-Cache (Ma et al., 2025) and dLLM-Cache (Liu et al., 2025b) develops the naive KV-Cache techniques. Their observations are that, with small update intervals (*e.g.*, 2 to 8), caching in dLLMs leads to minimal performance degradation and achieves ˜10x speed-up. For dMLLMs, Dimple (Yu et al., 2025)

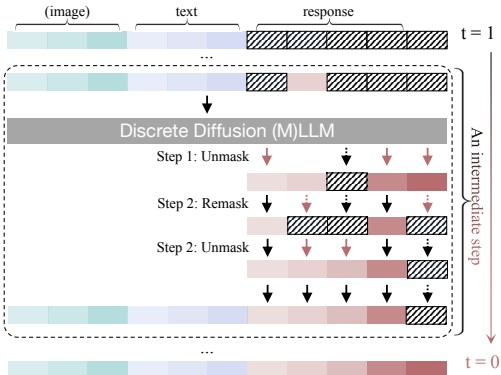

Figure 6: **Remasking** in General Discrete Diffusion Models.

and LaViDa (Li et al., 2025e) empirically verify that the use of prefilling incurs negligible performance degradation on the majority of vision-language benchmarks and provides a 2× to 7× speed-up.

Following are some representative designs for KV-Cache.

- **dKV-Cache** (Ma et al., 2025). The core idea of dKV-Cache is to cache the key-value pairs when the tokens are unmasked and reuse the cached key-value pairs with an interval hyperparameter to periodically update the cached key-value pairs.

- **dLLM-Cache** (Liu et al., 2025b). In addition to key-value (KV) pairs, dLLM-Cache stores attention (AttnOut) and feed-forward (FFNOut) outputs, reusing them in later steps. It applies different update intervals for prompt and response segments, with the prompt cache updated far less frequently. For responses, beyond periodic full updates, an Adaptive Partial Update strategy selectively refreshes tokens: cosine similarity between current and cached values identifies significant changes, and after each forward pass, a subset of tokens is proportionally updated.

- **DualCache** (Wu et al., 2025b). DualCache adopts a block-wise caching strategy, decoding text block by block. The block currently being decoded is not cached, while the surrounding blocks are cached.

## 6.4 Guidance

In dLLMs and dMLLMs, post-processing on the predicted logits or sampling probabilities is commonly referred to as *guidance*, following terminology from image diffusion models. Guidance methods are used to influence the generation process toward desired characteristics, such as improved diversity or controllability. Fig. 7 provides an illustration of several guidance techniques. The detailed mathematical formulations are included in Appendix Sec. D.II. *Classifier-free guidance* adjusts conditional predictions with unconditional ones to mitigate prompt-independent bias and enhance text diversity (Schiff et al., 2025; Nie et al., 2025a), though excessive guidance may degrade quality (Nisonoff et al., 2025; Rojas et al., 2025). In contrast, *classifier guidance* explicitly integrates class-conditional signals via an auxiliary classifier, enabling controllable generation across diffusion blocks (Schiff et al., 2025; Huang & Tang, 2025). Beyond classifier-based methods, *reward guidance* (Tae et al., 2025) leverages a reward model to adjust logits at inference by gradient ascent, thus steering outputs toward high-quality responses. Finally, *energy-based diffusion* augments the denoising distribution with an energy function $E_\phi$ and reweights candidate samples via importance sampling, providing a theoretically grounded correction mechanism (Xu et al., 2025). Together, these techniques highlight the trade-offs between diversity, controllability, quality, and theoretical interpretability in guided discrete diffusion.

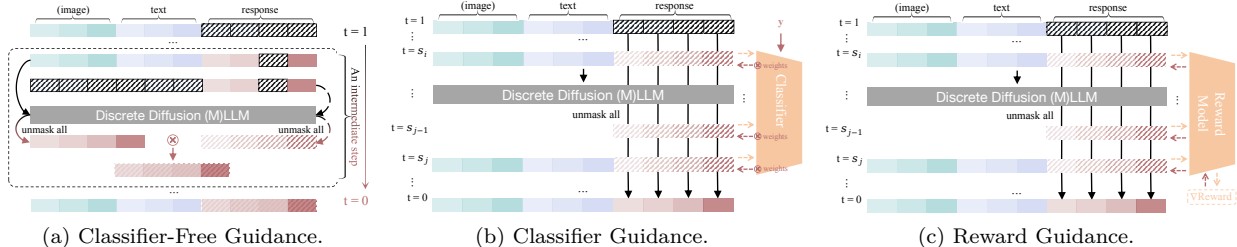

(a) Classifier-Free Guidance.    (b) Classifier Guidance.    (c) Reward Guidance.

Figure 7: **Guidance Techniques.** We divide the guidance techniques (i.e., the post-processing on the predicted logits or sampling probabilities) into three categories: (a) Classifier-Free Guidance, (b) Classifier Guidance, (c) and Reward Guidance.

## 6.5 Sampling Techniques

The generation process of dLLMs relies on a multi-round iterative denoising procedure. Beyond strictly following the mathematical definition of the inverse transition in discrete diffusion, many techniques introduce additional control over the iterative denoising process to achieve better performance or efficiency, such as, majority voting based on decoding history and early stopping. In this subsection, we provide a summary of these techniques.

### 6.5.1 Temporal Self-Consistency

Wang et al. (2025d) identifies a phenomenon termed *temporal oscillation*, where correct intermediate predictions emerge during the iterative decoding steps but are later overwritten by incorrect outputs in subsequent steps.

To mitigate the performance degradation caused by temporal oscillation, rather than relying exclusively on the final denoising step, Wang et al. (2025d) utilizes the predictions at all timesteps and predicts through a weighted voting scheme:

$$\text{answer}^* = \arg\max_a \sum_{t=1}^{T} f(t) \cdot \mathbf{1}\big(\text{semantic meaning}(x_0^t) = a\big), \tag{49}$$

where $x_0^t$ denotes the prediction at time step $t$, and semantic meaning$(x_0^t)$ represents the extracted answer from $x_0^t$, $\mathbf{1}(\cdot)$ is the indicator function and $f(t)$ is a weighting function (e.g., exponential decay) assigning more importance to later steps. Wang et al. (2025d) also introduces Temporal Semantic Entropy (TSE), a measure of semantic uncertainty across intermediate predictions. The computation of TSE is based on the semantic clustering of historical answers. This metric can be incorporated into the reinforcement learning advantage function, thereby mitigating the issue of temporal oscillation in the model.

### 6.5.2 Particle Gibbs Sampling for Discrete Diffusion Models

Instead of sampling only once to obtain a single trajectory, Dang et al. (2025) introduces a reference trajectory that is iteratively refined across multiple rounds of sampling.

The method begins by drawing an initial reference trajectory $x'_{0:T}$ using Sequential Monte Carlo (SMC) with a single particle. At each subsequent iteration, a set of $k$ particles is generated, where one particle is deterministically fixed to the reference trajectory while the remaining $k-1$ particles are defined as the candidate particles and are initialized randomly. At each step $t$, candidate particles are propagated according to the model's denoising distribution, $\bar{x}_{t-1}^{(i)} \sim p_\theta(x_{t-1} \mid c, x_t^{(i)})$, with the reference particle $\bar{x}_{t-1}^{(k)}$ kept fixed. The importance weight are for each candidate particles at time $t$ is defined based on a reward function $r$:

$$w_{t-1}^{(i)} = \exp\left(\frac{r(c, \bar{x}_{t-1}^{(i)}) - r(c, x_t^{(i)})}{\beta}\right), \tag{50}$$

and normalized before resampling. After resampling, the reference trajectory is updated by selecting one trajectory from the set with probability proportional to its importance weight. Iterating this procedure $m$ times yields a refined trajectory distribution that converges to the reward-weighted posterior.

### 6.5.3 Early Stopping

Li et al. (2025c) finds that diffusion models can often generate the correct answer before completing the entire decoding process. Therefore, discrete diffusion can leverage early stopping to improve decoding efficiency. Both Li et al. (2025c) and Jin et al. (2025) employ either predefined response templates or in-place prompts to divide the response space into two parts: the final answer region and the reasoning region. Once the confidence in the final answer region reaches a sufficient level, the subsequent decoding is terminated.

### 6.6 Context Length Extension

A notable phenomenon observed in dLLMs is that, with bidirectional attention, their robustness in handling extended contexts. Unlike autoregressive models that fail beyond the pretraining context window, diffusion LLMs can still retrieve information from the most recent segment of the input, even at depths far beyond the training context length.

To further extend the context window, LongLLaDA (Liu et al., 2025a) applies the NTK-aware scaling method to RoPE within diffusion LLMs. Following (u/emozilla (Reddit user), 2023), the critical dimension $d_{\text{extra}}$ is:

$$d_{\text{extra}} = 2 \cdot \left\lceil \frac{d}{2} \cdot \log_{\beta_0} \left( \frac{T_{\text{train}}}{2\pi} \right) \right\rceil, \tag{51}$$

where $d$ is the hidden dimension, $\beta_0$ is the rotary base, and $T_{\text{train}}$ is the pretraining context length. Given a target context length $t$, the NTK scaling factor $\lambda$ is chosen as:

$$\lambda = \beta_0^{-1} \cdot \left( \frac{t}{2\pi} \right)^{d/d_{\text{extra}}}. \tag{52}$$

### 6.7 Sparse Computation

The forward computation of diffusion-based large language models exhibits inherent sparsity, allowing inference efficiency to be improved by eliminating part of the computations.

Under the block-wise decoding setup, many techniques have been explored. Sparse-dLLM (Song et al., 2025a) indicates that it is unnecessary to cache and reuse all key–value pairs of tokens outside the current block. By applying an average attention score between non-current tokens and tokens in the current block, a subset of tokens can be filtered out, thereby reducing the memory consumption of the KV cache and improving inference throughput. DPad (Chen et al., 2025) demonstrates that tokens in subsequent blocks do not need to attend to all tokens in the current block. Instead, a random dropout can be applied, with a higher dropout rate assigned to tokens that are farther from the current block. Pipelined Parallel decoding (Wang et al., 2025e) adopts a "soft" block-wise decoding strategy. Instead of waiting for the current block to be fully decoded before moving to the next, it begins decoding tokens in the subsequent block once the proportion of decoded tokens in the last active block exceeds a predefined threshold. This gradual inclusion of additional blocks reduces computation on the right-side tokens and thereby improves efficiency.

### 6.8 Response Length Control

Adaptive control of response length is an essential capability in discrete diffusion models, and it can be achieved through either training-based or training-free approaches. In training-based methods, DreamOn (Wu et al., 2025d) enables dynamic adjustment of token counts by introducing two special tokens, expand and delete. Similarly, FlexMDM (Kim et al., 2025a) employs an auxiliary network to predict how many tokens should be inserted before each existing token, thereby allowing the response length to grow dynamically. In contrast, the following section introduces training-free techniques for adaptive response length control.

DAEDAL (Li et al., 2025a) introduces a two-stage adaptive mechanism for response length control. It uses the confidence in predicting the End-of-Sequence (EOS) token as a signal of length sufficiency. First, it starts with a short sequence of length $L_{\text{init}}$ and repeatedly appends the response length until the EOS confidence exceeds $\tau_{\text{eos}}$ or the maximum length $L_{\text{max}}$ is reached. Second, while denoising, low-confidence tokens ($p_\theta(x_t[i]) < \tau_{\text{low}}$) trigger expansion: the least confident position is replaced by $E_{\text{factor}}$ mask tokens, allowing the sequence to grow adaptively.

Revised Part

## 7 Efficiency and Trustworthiness

### 7.1 Quantization

Lin et al. (2025a) presents the first systematic study of post-training quantization (PTQ) for diffusion-based large language models (dLLMs), benchmarking mainstream quantization methods across multiple architectures. Their analysis reveals severe activation outliers in dLLMs, task-dependent sensitivity variations, and the comparatively higher robustness of instruction-tuned models over base models.

Quant-dLLM Zhang et al. (2025b) introduces a 2-bit post-training quantization framework incorporating masked calibration simulation, a data-aware arbitrary-order quantizer, and an adaptive block-wise mixed-precision scheduling strategy. These techniques significantly improve the accuracy of 2-bit quantization relative to autoregressive baselines.

Xu & Yang (2025) further investigates the failure modes of conventional PTQ methods on dLLMs, identifying three primary challenges: (1) the accumulation of quantization errors across iterative steps, (2) heterogeneous token distributions across decoding stages, and (3) substantial disparities in feature distributions along both token and channel dimensions. To address these issues, they propose **DLLMQuant**, a framework comprising Temporal-Mask Adaptive Sampling (TMAS), Interaction-Aware Activation Quantization (IA-AQ), and Certainty-Guided Quantization (CGQ). Detailed descriptions of these components are provided in Appendix E.1.

### 7.2 Privacy and Safety

Diffusion-based large language models (dLLMs) introduce distinct safety vulnerabilities arising from their *bidirectional context modeling* and *parallel decoding* mechanisms. While these properties enable efficient infilling and interactive generation, they also weaken safeguards that are effective in autoregressive (AR) LLMs.

Diffusion-based LLM Jailbreak Attack (DIJA) (Wen et al., 2025) and Parallel Decoding Attack (PAD) (Zhang et al., 2025c) reformulate conventional jailbreak prompts into an interleaved mask-text representation, inducing the model to generate unsafe outputs while preserving contextual coherence. Formally, let $a = (a_1, \ldots, a_R)$ denote a standard jailbreak prompt, and let $m = ([\text{MASK}], \ldots, [\text{MASK}])_Q$ represent $Q$ consecutive mask tokens. DIJA and PAD construct an interleaved prompt

$$p_i = a \oplus (m \otimes w), \tag{53}$$

where $\oplus$ denotes concatenation, $\otimes$ denotes interleaving, and $w$ represents benign separator text (e.g., 'Step 1" and 'Step 2"). Crucially, the harmful intent embedded in $a$ is preserved while key instructions are forced into masked positions.

Xie et al. (2025b) demonstrates that dLLMs are more susceptible to manipulation in the middle of a generated sequence than at its beginning. Existing optimization-based jailbreak methods effectively manipulate initial tokens but largely fail to control intermediate tokens. To mitigate this vulnerability, Xie et al. (2025b) proposes Middle Token Safety Alignment (MOSA), a reinforcement learning-based alignment approach that constrains generated middle tokens to align with a predefined set of safe tokens.

Empirical results indicate that enforcing alignment in the middle of the sequence is more effective than focusing solely on the initial tokens.

Watermarking Diffusion Language Models applies watermarking in expectation and biases token sampling to achieve over 99% detection accuracy with minimal impact on generation quality Gloaguen et al. (2026). DMark introduces predictive and bidirectional watermarking, achieving detection rates between 92% and 99.5% at a 1% false-positive rate without degrading text quality Wu et al. (2025c). Watermarking Discrete Diffusion Language Models employs a distribution-preserving Gumbel-Max trick with position-seeded randomness to enable distortion-free watermarking Bagchi et al. (2026). DiffuGuard identifies denoising-path dependence and harmful biases induced by greedy remasking, and proposes a dual-stage defense based on stochastic annealing and block-level auditing, reducing jailbreak success rates from 47.9% to 14.7% Li et al. (2026b). A2D aligns dLLMs to emit a refusal token upon detecting harmful content, reducing attack success rates to near zero while enabling early termination to accelerate inference Jeung et al. (2026). A study on priming vulnerability shows that inserting an affirmative token early in the denoising trajectory can steer outputs toward harmful content, and proposes training strategies that enforce safe responses even from contaminated intermediate states Yamabe & Sakuma (2026).

## 8 Applications

*Revised Part*

### 8.1 Language Related Applications

Recent work applies diffusion language models to language tasks. Lyu et al. (2023) introduce a diffusion-based model for fine-grained style transfer. Zhang et al. (2025a) leverage the bidirectional nature of diffusion for text embeddings, achieving strong reasoning performance. To enable controllable generation, Zhu et al. (2024b) propose Segment-Level Diffusion, decoding segments sequentially. In applications, Cheng & Li (2024) use diffusion with task-specific masking and attribute tags to generate poll options; Hu et al. (2024) enforce both semantics and meter in poetry; and Padole et al. (2025) apply masked diffusion with verifier guidance for style control. EdiText (Lee et al., 2025a) extends controllable generation to text editing, operating at both coarse and fine levels to achieve target attributes. Moving from editing to long-form generation, Do et al. (2025) design a discrete diffusion model for abstractive summarization. Complementing this direction, DiffETM (Shao et al., 2025) inject diffusion into the embedded topic model, enabling document–topic distributions to be sampled through a more realistic stochastic process. The $CDA^2$ framework (Xin et al., 2025) uses counterfactual diffusion augmentation to improve cross-domain sentiment adaptation. DiffusionCLS (Chen et al., 2024c) enhances low-resource classification by generating label-consistent pseudo-samples via sentiment-relevant token reconstruction. For aspect sentiment quad prediction, GDP (Zhu et al., 2024a) employs a template-guided diffusion strategy. In layout generation, Iwai et al. (2024) mitigate layout sticking in discrete diffusion models by introducing Layout-Corrector, which scores and resets misplaced tokens for improved positioning. TermDiffuSum (Dong et al., 2025) integrates term-aware attention and a re-ranking loss during diffusion, effectively emphasizing legally legally salient sentences. Diffutron Şuayp Talha Kocabay & Akkuş (2026) introduces a masked diffusion language model tailored for Turkish, a morphologically rich language. Leveraging LoRA-based continual pretraining and progressive instruction tuning, the model achieves competitive performance despite its compact size, demonstrating the effectiveness of diffusion modeling in low-resource settings.

### 8.2 Reasoning

Diffusion-of-Thought (DoT) (Ye et al., 2024) firstly integrates chain-of-thought reasoning into dLLMs to enhance reasoning capabilities. DiffuCOMET (Gao et al., 2024) develops a series of models that leverage diffusion to infer contextually relevant commonsense knowledge from narratives. DPCL-Diff (Cao et al., 2025) combines graph node diffusion with dual-domain periodic contrastive learning for temporal knowledge graph reasoning. The d1 framework (Zhao et al., 2025a) adapts pretrained dLLMs into reasoning models via

a combination of supervised fine-tuning and reinforcement learning. It introduces a novel critic-free policy-gradient algorithm called diffu-GRPO and employs masked SFT to distill reasoning knowledge directly from existing datasets. Ye et al. (2025a) provides an insight into dLLM reasoning: the difficulty of decoding individual tokens in a response varies, and it is not necessarily the case that tokens on the left are easier to decode than those on the right. NeSyDM (van Krieken et al., 2025) introduces a discrete diffusion process in the symbolic space. The framework first extracts symbolic representations from perceptual inputs and then performs a multi-step denoising diffusion process over these symbols.

**Revised Part**

### 8.3 Vision and Multimodal

UDAN-CLIP (Shaahid & Behzad, 2025) proposes an image-to-image diffusion framework for underwater enhancement. Turning to motion synthesis, M2D2M (Chi et al., 2024) employs a discrete diffusion model to generate continuous human-motion sequences from textual descriptions of multiple actions. AR-Diffusion (Sun et al., 2025b) introduces a novel architecture that blends autoregressive and diffusion techniques for flexible, asynchronous video generation. Kumbhar et al. (2026) explores the application of discrete vision–language diffusion models (DVLMs) to GUI grounding tasks. By adapting LLaDA-V for action and bounding-box prediction and introducing a hybrid masking schedule, the method improves grounding accuracy and demonstrates competitive performance with autoregressive models, highlighting the potential of diffusion-based GUI agents. Besides the understanding tasks, discrete diffusion is also largely used in the vision generation tasks (Chang et al., 2022; Bai et al., 2024). DiMo Zhang et al. (2026c) extends masked diffusion modeling to unified text–motion understanding and generation. Through iterative token refinement and enhanced token representations, it supports multiple tasks within a single framework and achieves strong performance across motion benchmarks. D3PIA Choi et al. (2026) addresses piano accompaniment generation using discrete diffusion with neighborhood attention. It better preserves harmonic structure than transformer and continuous diffusion baselines and produces more coherent musical outputs. DIFFA-2 Zhou et al. (2026) develops a practical diffusion-based large audio language model with improved semantic–acoustic alignment and multi-stage training. It achieves competitive performance with autoregressive models while maintaining efficiency under limited training resources. MaDiS Zuo et al. (2026) proposes a masked diffusion model for sign language generation, enabling bidirectional context modeling and parallel token generation. With multi-level cross-modal pretraining and efficient decoding strategies, it achieves state-of-the-art performance while reducing inference latency.

**Revised Part**

### 8.4 Robotics and Autonomous Driving

DiffVLA (Jiang et al., 2025) introduces a vision-language-guided diffusion policy for autonomous-driving planning. ViLaD (Cui et al., 2025) introduces a large vision–language diffusion framework for end-to-end autonomous driving, addressing the latency and unidirectional limitations of autoregressive VLM-based decision models. Meanwhile, Discrete Diffusion VLA (Liang et al., 2025b) unifies vision, language, and action decoding in a single transformer. It discretizes actions into tokens and employs iterative re-masking for adaptive, parallel action generation. Extending diffusion policies to robotics, VPDD (He et al., 2024) first pre-trains on large-scale actionless human videos and then transfers the learned discrete diffusion policy to robot-control tasks. Discrete-Guided Diffusion (DGD) (Liang et al., 2025a) integrates discrete multi-agent path finding (MAPF) with diffusion models to address scalable and safe multi-robot motion planning. Fast-dVLA proposes a parameter-space decoupling strategy to enhance pretrained VLA models without incurring the computational overhead of auxiliary objectives. By learning and merging capability vectors, the method achieves comparable performance to advanced fine-tuning approaches while significantly reducing training cost Song et al. (2026). MVLAD-AD Zhang et al. (2026a) proposes a masked vision–language–action diffusion framework for end-to-end driving. By introducing discrete action tokenization and geometry-aware embeddings, it improves planning precision and efficiency while providing interpretable decision-making.

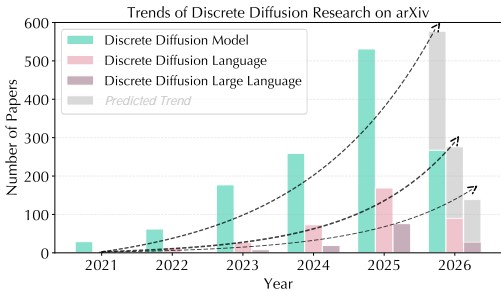

Figure 8: Number of arXiv publications retrieved via keyword-based search (*Discrete Diffusion Model*, *Discrete Diffusion Language*, and *Discrete Diffusion Large Language*) under the *Computer Science (cs)* category across titles and abstracts. The results show a consistent year-over-year increase, reflecting the growing research interest in this area.

## 8.5 Graph and Structured Predictions

LO-ARM (Wang et al., 2025f) introduces a learning-order autoregressive model that dynamically adapts generation order for molecular graphs, improving flexibility and validity. ReDiSC (Li et al., 2025g) proposes a reparameterized discrete masked diffusion model for node classification and achieves scalable and interpretable predictions on large graphs. Scaffold Diffusion (Jung, 2025) formulates sparse multi-category voxels as discrete token sequences and applies a dLLM for 3D sparse structure generation. Ghiglino et al. (2026) presents a hierarchical diffusion framework for eVTOL aircraft design. By combining discrete topology generation with continuous parameter modeling, the approach captures physical design principles while substantially accelerating simulation-based inference.

Revised Part

## 8.6 Biological and Drug Discovery

A molecular-editing framework named MolEditRL (Zhuang et al., 2025) combines a discrete graph-diffusion model with reinforcement learning to optimize molecular properties while preserving structural similarity. CFP-Gen (Yin et al., 2025) adopts a diffusion language model for combinatorial functional protein generation. TransDLM (Xiong et al., 2024) proposes a text-guided, multi-property molecular-optimization method that leverages a diffusion language model. GenMol (Lee et al., 2025c) presents a single, discrete diffusion model that serves as a versatile generator across diverse pharmaceutical tasks. DPLM-2 (Wang et al., 2024) is a multimodal protein language model capable of understanding and generating both protein sequences and their 3D structures. PepTune (Tang et al., 2025a) targets therapeutic-peptide design with a multi-objective, discrete diffusion framework built on a masked language model backbone. (Zhang et al., 2025d) propose CMCM-DLM, which integrates structure-control and property-control modules into a pretrained dLLM for molecules. D3LM reformulates DNA modeling as a masked diffusion process, enabling unified bidirectional understanding and generation. It outperforms comparable transformer-based baselines and achieves strong results in regulatory sequence generation, suggesting diffusion as a promising paradigm for biological foundation models Yang et al. (2026). MolHIT Jung et al. (2026) introduces a hierarchical discrete diffusion framework for molecular graph generation. By incorporating chemical priors and structured atom encoding, it achieves near-perfect chemical validity and state-of-the-art performance on the MOSES benchmark.

# 9 Conclusion

In summary, this survey provides a comprehensive overview of Discrete Diffusion Large Language Models (dLLMs) and Discrete Diffusion Large Multimodal Models (dMLLMs). We present a detailed exposition of their mathematical foundations and landmark developments. We further detail the training and inference

strategies behind them, and summarize the current application domains and potential future directions of them. As a promising alternative to autoregressive LLMs, dLLMs have attracted growing attention (see Figure 8) and show great potential in a variety of real-world scenarios. We hope this survey will serve as a valuable foundation for future research and development in this fast-evolving and important field. At the end of the Appendix, we also discuss some future directions of dLLM and dMLLM.

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

# - *Appendix* -

In this appendix, we provide additional details on the works and techniques introduced in the main text. Specifically, we present more comprehensive mathematical formulations for certain methods and include several related techniques not covered earlier. The organization of the appendix largely follows the structure of the main-text. Moreover, at the end of the appendix, we also discuss potential future research directions in dLLM and dMLLM.

## A Mathematical Formulations

### A.1 More Transition Matrices

- **Hybrid**: Hybrid transition was initially discussed in (Austin et al., 2021), where multiple types of transition are combined to create more expressive diffusion processes. For example, the Routlette Diffusion in (Haxholli et al., 2025) and the Generalized Interpolating Discrete Diffusion (GIDD) (von Rütte et al., 2025) both study the linear combination of the absorbing transition and the uniform transition. A key motivation for such "Linear + Absorbing" transitions is to overcome a fundamental limitation of absorbing diffusion: once a token is denoised, it cannot be revised. Such "Linear + Absorbing" transitions address this by allowing previously denoised tokens to re-enter the diffusion process via the uniform component. This grants the model the ability to correct earlier generation errors during inference.

- **Discretized Gaussian**:

$$[Q_t]_{ij} = \begin{cases} \dfrac{\exp\left(-\frac{4|i-j|^2}{(K-1)^2\beta_t}\right)}{\sum\limits_{n=-(K-1)}^{K-1} \exp\left(-\frac{4n^2}{(K-1)^2\beta_t}\right)} & \text{if } i \neq j \\ 1 - \sum\limits_{l \neq i}[Q_t]_{il} & \text{if } i = j \end{cases}$$

This matrix simulates Gaussian diffusion, suitable for ordinal data. Each state $i$ is most likely to transition to nearby states $j$ with probabilities resembling a Gaussian kernel. Closer states receive higher probabilities, while distant states receive lower ones. Diagonal values are chosen to ensure that each row sums to 1, yielding a uniform stationary distribution.

- **Band-diagonal**:

$$[Q_t]_{ij} = \begin{cases} \frac{1}{K\beta_t} & 0 < |i-j| \leq v \\ 0 & |i-j| > v \\ 1 - \sum\limits_{l \neq i}[Q_t]_{il} & i = j \end{cases}$$

Band-diagonal imposes local transitions only: state $i$ can only transition to its $v$-nearest neighbors on the ordinal scale. It biases the forward process toward small, local perturbations.

- **Embedding-based**: Let $A$ be an adjacent matrix build from based on the similarity between tokens in the embedding space, in (Austin et al., 2021) the Embedding-based transition matrix is defined as

$$Q_t = \exp(\alpha_t R) = \sum_{n=0}^{\infty} \frac{\alpha_t^n}{n!} R^n, \tag{54}$$

$$R_{ij} = \begin{cases} -\sum_{l \neq i} A_{il} & i = j \\ A_{ij} & i \neq j \end{cases}. \tag{55}$$

This transition matrix promotes a diffusion process where tokens are more likely to transition to others that are semantically or syntactically similar in the embedding space.

## A.2 Reparameterized Discrete Diffusion Model

Reparameterized Discrete Diffusion Models (RDMs) (Zheng et al., 2023) reformulate the backward process of discrete diffusion in D3PM into a two-stage sampling procedure. The core idea is to introduce a latent routing variable $v_t$ that determines decoding behavior for each token at every step.

Given a forward transition of the form:

$$q(x_t \mid x_{t-1}) = \beta_t x_{t-1} + (1 - \beta_t) q_{\mathrm{noise}}, \tag{56}$$

$$q(x_t \mid x_0) = \alpha_t x_{t-1} + (1 - \alpha_t) q_{\mathrm{noise}}, \tag{57}$$

where $q_{\mathrm{noise}}$ is the noise distribution, and $\alpha_t := \prod_{i=1}^{t} \beta_i$. The backward posterior $q(x_{t-1} \mid x_t, x_0)$ can be expressed as a mixture of two cases, depending on whether $x_t = x_0$:

$$q(x_{t-1} \mid x_t, x_0) = \begin{cases} \lambda_{t-1}^{(1)} x_t + (1 - \lambda_{t-1}^{(1)}) q_{\mathrm{noise}}, & x_t = x_0, \\ \lambda_{t-1}^{(2)} x_0 + (1 - \lambda_{t-1}^{(2)}) q_{\mathrm{noise}}(x_t), & x_t \neq x_0, \end{cases} \tag{58}$$

where $q_{\mathrm{noise}}(x_t)$ is the interpolation between $x_t$ and $q_{\mathrm{noise}}$, and $\lambda_{t-1}^{(1)}$, $\lambda_{t-1}^{(2)}$ are scalar coefficients derived from $\beta_t$, $\alpha_t$ and $q_{\mathrm{noise}}(x_t)$.

To reparameterize the backward transition, RDM introduces Bernoulli latent variables:

$$v_{t-1}^{(1)} \sim \mathrm{Bernoulli}(\lambda_{t-1}^{(1)}), \quad u_t^{(1)} \sim q_{\mathrm{noise}}, \tag{59}$$

$$v_{t-1}^{(2)} \sim \mathrm{Bernoulli}(\lambda_{t-1}^{(2)}), \quad u_t^{(2)} \sim q_{\mathrm{noise}}(x_t). \tag{60}$$

Let $b_t = \mathbf{1}[x_t = x_0]$, the reparameterized sample is computed as:

$$\begin{aligned} x_{t-1} = b_t &\left[ v_{t-1}^{(1)} x_t + (1 - v_{t-1}^{(1)}) u_t^{(1)} \right] \\ &+ (1 - b_t) \left[ v_{t-1}^{(2)} x_0 + (1 - v_{t-1}^{(2)}) u_t^{(2)} \right]. \end{aligned} \tag{61}$$

This formulation allows the model to explicitly route tokens through different behaviors: either retaining the current token, resetting to noise, or denoising back to the ground truth.

Based on Eq. (61), RDMs define a joint generative model:

$$p_\theta(x_0, x_{1:T}, v_{1:T}) = p_\theta(x_T) \prod_{t=1}^{T} p_\theta(x_{t-1}, v_{t-1} \mid x_t), \tag{62}$$

and the evidence lower bound (ELBO) becomes:

$$\log p_\theta(x_0) \geq \mathcal{L}_1 - \underbrace{\sum_{t=2}^{T} \mathcal{L}_t}_{\mathcal{L}_T} + C, \tag{63}$$

where $C$ is a constant.

For $t > 1$, the loss term decomposes as:

$$\begin{aligned} \mathcal{L}_t = \mathbb{E}_{x_{1:T}, v_{1:T} \mid x_0} &\Big[ \mathrm{KL}(q(v_{t-1}) \parallel p_\theta(v_{t-1} \mid x_t)) + \\ &\mathrm{KL}(q(x_{t-1} \mid v_{t-1}, x_t, x_0) \parallel p_\theta(x_{t-1} \mid v_{t-1}, x_t)) \Big]. \end{aligned} \tag{64}$$

By aligning $p_\theta(v_{t-1} \mid x_t)$ with $q(v_{t-1})$ and using the $x_0$-parameterization, the loss can be simplified into

$$\mathcal{L}_t = \mathbb{E}_{x_0, x_{1:T}} \left[ - \lambda_{t-1}^{(2)} \sum_{n=1}^{N} (1 - b_{t,n}) x_{0,n} \log[f_\theta(x_t)]_n \right], \tag{65}$$

where $\lambda_{t-1}^{(2)} = \frac{\alpha_{t-1} - \alpha_t}{1 - \alpha_t}$.

A central issue of RDM lies in its dependence on the ground truth $x_0$ to compute the backward transition probabilities Eq. (58). However, in the inference stage, $x_0$ is unknown, making it infeasible to directly evaluate the indicator function $b_t$ required for determining the appropriate transition path. To overcome this limitation, the authors propose a recursive approximation for computing $b_t$ by utilizing the Bernoulli routing variables $v$. Beginning with $b_T = 0$, which assumes the initial sequence is fully noisy, the clean token set is recursively updated via:

$$b_{t-1,n} = \left( b_{t,n} \wedge v_{t-1,n}^{(1)} \right) \vee v_{t-1,n}^{(2)}, \tag{66}$$

where $\wedge$ and $\vee$ denote logical conjunction and disjunction, respectively.

### A.3 Concrete Score

### A.3.1 Training Loss

Training is typically done by minimizing divergence-based losses that compare the predicted ratio to the true ratio.

**Concrete Score Matching (CSM) (Meng et al., 2022)** The most direct approach is Concrete Score Matching (CSM) (Meng et al., 2022), which uses a squared-error loss to match the predicted and true ratios:

$$\mathcal{L}_{\text{CSM}}(t) = \frac{1}{2} \mathbb{E}_{x \sim p_t} \left[ \sum_{x' \neq x} \left( s_\theta(x, t)_{x'} - \frac{p_t(x')}{p_t(x)} \right)^2 \right] \tag{67}$$

While theoretically sound, this $\ell_2$ loss does not penalize invalid (e.g., negative or zero) predictions sufficiently, potentially leading to instability.

**Diffusion-Weighted Denoising Score Entropy (Lou et al., 2023)** Leveraging Bregman divergence, score entropy is formulated as another score matching loss. Score entropy is non-negative, symmetric, and convex. It is also an extension of the conventional cross-entropy to general positive-valued functions beyond the probability simplex. Score entropy enables the construction of an ELBO for discrete diffusion models, resulting in the Diffusion-Weighted Denoising Score Entropy (DWDSE) loss (Lou et al., 2023)

$$\mathcal{L}_{\text{DWDSE}}(x_0) = \int_0^T \mathbb{E}_{x_t \sim p_{t|0}(\cdot|x_0)} \sum_{x' \neq x_t} R_t(x_t, x') \Bigg($$

$$s_\theta(x_t, t)_{x'} - \frac{p_{t|0}(x'|x_0)}{p_{t|0}(x_t|x_0)} \log s_\theta(x_t, t)_{x'}$$

$$+ K\left( \frac{p_{t|0}(x'|x_0)}{p_{t|0}(x_t|x_0)} \right) \Bigg) dt, \tag{68}$$

where $K$ is a normalizing constant function.

**Target Concrete Score Matching (ZHANG et al., 2025)** Target Concrete Score Matching (TCSM) introduces two score matching losses: the *score-based* and *distribution-based* losses. The score-based objective operates directly on the concrete score vectors:

$$\mathcal{L}_{\text{score}}(\theta) = \mathbb{E}_{\omega(t)p(x_t)h(x_1|x_t)} \sum_{i=1}^L \ell_{\text{score}}^i, \tag{69}$$

$$\ell_{\text{score}}^i = \mathcal{D}\left( \left[ \frac{p_{1|t}([y^i, x_1^{\backslash i}]|x_t)}{p_{1|t}(x_1|x_t)} \right]_{y^i=1}^V, \left[ \frac{p_{1|t}^\theta([y^i, x_1^{\backslash i}]|x_t)}{p_\theta^{1|t}(x_1|x_t)} \right]_{y^i=1}^V \right), \tag{70}$$

where $V$ is the vocabulary size, $\mathcal{D}(\cdot, \cdot)$ denotes a divergence measure, $\omega(t)$ is the distribution used to sample time index $t$, and $h(x_1|x_t)$ is a proposal distribution such as the ground-truth conditional distribution $p_{1|t}(x_1|x_t)$. In the equation, $[y^i, x^{\backslash i}] := [x^1, \ldots, x^{i-1}, y^i, x^{i+1}, \ldots, x^L]$ is used to define a new sequence of the same length as $x_1$, where the token at position $i$ is replaced by $y^i$ and all other tokens remain identical to those in $x_1$. $x_1^{\backslash i}$ is used to indicate all tokens in $x_1$ except for the token at position $i$.

The distribution-based objective aligns the model and true conditional distributions:

$$\mathcal{L}_{\text{distrib}}(\theta) = \mathbb{E}_{\omega(t)p(x_t)} \sum_{i=1}^{L} \mathbb{E}_{h(x_1^{\backslash i}|x_t)} \ell_{\text{distrib}}^i, \tag{71}$$

$$\ell_{\text{distrib}}^i = \mathbb{D}\left( p_{1|t}(x_1^i|x_1^{\backslash i}, x_t) \,\|\, p_{1|t}^\theta(x_1^i|x_1^{\backslash i}, x_t) \right), \tag{72}$$

where $\mathbb{D}(\cdot, \cdot)$ is a statistical divergence measures the differences between probability distributions. Shown in Proposition 3 of (ZHANG et al., 2025), with $h(x_1|x_t) = p^{1|t}(x_1|x_t)$, the two objectives are equivalent:

$$\mathcal{L}_{\text{score}}(\theta; \mathcal{D} = \mathcal{D}_{\text{GKL}}) \equiv \mathcal{L}_{\text{distrib}}(\theta; \mathbb{D} = V\mathbb{D}_{\text{KL}} + \mathbb{D}_{\text{IS}}), \tag{73}$$

where $\mathcal{D}_{\text{GKL}}$, $\mathbb{D}_{\text{KL}}$ and $\mathbb{D}_{\text{IS}}$ refer to the generalized Kullback–Leibler divergence, the Kullback–Leibler divergence and Itakura–Saito divergence, respectively.

By selecting different discrepancy measures and proposal distributions, there are different instantiations of the TCSM loss. For instance, choosing the generalized KL divergence as the discrepancy $\mathcal{D}$ and the true conditional distribution $p_{1|t}(x_1|x_t)$ as the proposal $h(x_1|x_t)$, the score-based TCSM becomes

$$\ell_{\text{score}}^i = \left( -\log p_{1|t}^\theta(x_1^i|x_t) + \frac{1}{V p_{1|t}^\theta(x_1^i|x_t)} \right) + \frac{1}{V} \sum_y \log p_{1|t}^\theta(y|x_t); \tag{74}$$

choosing the KL divergence as the discrepancy $\mathbb{D}$ and the true conditional distribution $p_{1|t}(x_1|x_t)$ as the proposal $h(x_1|x_t)$, the distribution-based TCSM becomes

$$\ell_{\text{distrib}}^i = -\mathbb{E}_{p_{1|t}} \log p_{1|t}^\theta(x_1^i|x_t) + C, \tag{75}$$

where $C$ is a constant. The distribution-based objective reduces to a cross-entropy loss, maximizing the pseudo-likelihood of $p_{1|t}^\theta$ under the true denoising distribution.

### A.3.2   Connection with CE Loss

Sun et al. (2023) leverages the theorem stating that two probability distributions are identical if and only if their conditional distributions are equal for all values of the condition. Based on this, the original marginal probability distributions in the concrete score are transformed into conditional probability distributions. Since both the numerator and denominator of the concrete score share the same functional form, they can be represented by a single neural network. That is, the concrete score can be expressed as:

$$p_t(X^d \mid x^{\backslash d}; \theta) \approx q_t(X^d \mid x^{\backslash d}) \Rightarrow$$
$$\frac{q_t(y^d, x^{\backslash d})}{q_t(x^d, x^{\backslash d})} = \frac{q_t(X_t^d = y^d \mid x^{\backslash d})}{q_t(X_t^d = x^d \mid x^{\backslash d})} \approx \frac{p_t(X_t^d = y^d \mid x^{\backslash d}; \theta)}{p_t(X_t^d = x^d \mid x^{\backslash d}; \theta)}. \tag{76}$$

From this perspective, Sun et al. (2023) propose the categorical ratio matching loss, which is a cross-entropy loss to fit conditional probabilities. Thus, Sun et al. (2023) shows that training a neural network with a cross-entropy loss is also able to fit the concrete score.

Haxholli et al. (2025) also connects the concrete score matching with the CE loss. Consider two discrete sequences $x$ and $y$ that only differ at position $i$. By Bayesian Theorem, the probability ratio $\frac{p_t(y)}{p_t(x)}$ can be

rewritten as

$$\frac{p_t(y)}{p_t(x)} = \sum_{h \in \mathcal{V}} \frac{p_{t|0}(y^i \mid h)}{p_{t|0}(x^i \mid h)} \cdot p_{0|t}^i(h \mid x_t), \tag{77}$$

where $p_{t|0}(\cdot \mid h)$ is the known conditional transition probability in the forward process, and $p_{0|t}^i(h \mid x_t)$ is the posterior of the clean token $h$ at position $i$ given the noised sequence $x_t$. Thus, the concrete score can be parameterized as

$$s_\theta^i(x_t, t) = \sum_{h \in \mathcal{V}} \frac{p_{t|0}(y^i \mid h)}{p_{t|0}(x^i \mid h)} \cdot f_\theta^i(x_t, t)[h], \tag{78}$$

where $f_\theta^i(x_t, t)[h]$ models the likelihood that token $h$ was the original clean token at position $i$. This allows training to proceed via a simple cross-entropy loss over the posterior $p_{0|t}$, rather than requiring explicit score estimation:

$$\mathcal{L}_{\text{CEDD}} = \mathbb{E}_t \mathbb{E}_{x_0, x_t} \sum_{i=1}^{L} w(t) \cdot \log f_\theta^i(x_t, t)[x_0^i], \tag{79}$$

where $w(t)$ is a timestep-dependent weighting function.

### A.3.3 Time Independency

An issue with the Concrete Score is its dependency on time, which prevents the use of caching techniques for inference and results in lower efficiency. Ou et al. (2025) shows that the concrete score in absorbing discrete diffusion can be reparameterized as a product of a time-dependent scalar and a conditional distribution over clean data. Specifically, in practice, $R_t$ can be parameterized as the multiplication between a scalar function and a constant rate, *i.e.*, $R_t = \sigma(t)R$. let $x_i = [\text{M}]$ denote a masked token at position $i$. Then, the concrete score for replacing $x_i$ with a token $x_i' \neq [\text{M}]$ is defined as:

$$\frac{p_t(x')}{p_t(x)} = \frac{p_t(x_1, \ldots, x_i', \ldots)}{p_t(x_1, \ldots, x_i, \ldots)} = \frac{e^{-\bar{\sigma}(t)}}{1 - e^{-\bar{\sigma}(t)}} \cdot p_0(x_i' \mid x^{\text{UM}}), \tag{80}$$

where:

- $\bar{\sigma}(t) = \int_0^t \sigma(s)ds$ is the cumulative noise schedule,

- $x^{\text{UM}}$ consists of all unmasked tokens of $x$,

- $p_0(x_i' \mid x^{\text{UM}})$ is the conditional distribution of the clean data.

This reparameterization removes the dependence on $t$ from the model output, enabling the training of a time-independent network $c_\theta$:

$$c_\theta(x)[i, \hat{x}^{(i)}] \approx p_0(\hat{x}^{(i)} \mid x^{\text{UM}}). \tag{81}$$

Such a model, termed RADD (Reparameterized Absorbing Discrete Diffusion), permits caching of the network outputs across steps when tokens remain unchanged, reducing the number of function evaluations (NFEs) during inference.

## B Modeling Language Diffusion

### B.1 Block Diffusion Models

Given a sequence $x = (x_1, \ldots, x_L)$, BD3-LMs partition it into $B$ blocks of length $L'$, denoted $x = (x^{(1)}, \ldots, x^{(B)})$. The joint likelihood under a Block Diffusion model is factorized autoregressively across blocks:

$$\log p_\theta(x) = \sum_{b=1}^{B} \log p_\theta(x^{(b)} \mid x^{(<b)}), \tag{82}$$

where each conditional distribution $p_\theta(x^{(b)} \mid x^{(<b)})$ is modeled by a discrete diffusion process applied within block $b$:

$$
\begin{aligned}
&p_\theta(x_s^{(b)} \mid x_t^{(b)}, x^{(<b)}) \\
&= \sum_{x^{(b)}} q(x_s^{(b)} \mid x_t^{(b)}, x^{(b)}) \, p_\theta(x^{(b)} \mid x_t^{(b)}, x^{(<b)}),
\end{aligned}
\tag{83}
$$

where $q(\cdot \mid \cdot)$ is the forward noise process, and $p_\theta(x^{(b)} \mid x_t^{(b)}, x^{(<b)})$ is the learned denoising model.

The model is parameterized by a transformer $f_\theta$ with a block-causal attention mask. For each block $x^{(b)}$, the model predicts:

$$
f_\theta\left(x_t^{(b)}, x^{(<b)}\right) \to \hat{x}_0^{(b)}.
\tag{84}
$$

During inference, block sampling proceeds sequentially over blocks, but parallel sampling is used within blocks. Block-wise KV caching can be used to avoid recomputing transformer states for previously generated blocks.

The training loss for the full sequence is obtained by summing the variational lower bound over all blocks:

$$
-\log p_\theta(x) \le \mathcal{L}_{\mathrm{BD}}(x; \theta) := \sum_{b=1}^{B} \mathcal{L}(x^{(b)}, x^{(<b)}; \theta),
\tag{85}
$$

where each $\mathcal{L}(x^{(b)}, x^{(<b)}; \theta)$ follows the discrete diffusion loss, optionally adapted to continuous time or masking processes.

## B.2   Flexible-Length Masked Diffusion

In a standard Masked Diffusion Models, each token can only exist in one of two states: *masked* or *unmasked*. In FlexMDM, each token may reside in one of three states: *empty*, *masked*, or *ground truth (gt)*. In the forward process, a token first transitions from the *ground truth* state to the *masked* state, and finally to the *empty* state. Let $x_1 = (x_1^1, \ldots, x_1^n) \sim p_1$ be a target sequence of length $n$. FlexMDM defines two smooth monotone schedules $\alpha, \beta : [0, 1] \to [0, 1]$ with boundary conditions $(\alpha_0, \beta_0) = (0, 0)$ and $(\alpha_1, \beta_1) = (1, 1)$. For each token $i$, FlexMDM independently sample an *insertion time* $T_1^i$ and an *unmasking time* $T_2^i$:

$$
T_1^i \sim \dot{\alpha}_t \, dt, \quad T_2^i \sim \mathbf{1}\{t \ge T_1^i\} \, \frac{\dot{\beta}_t}{1 - \beta_{T_1^i}} \, dt,
$$

where $\dot{\alpha}_t$ and $\dot{\beta}_t$ are the derivatives of $\alpha_t$ and $\beta_t$. The token state $x_t^i$ at time $t \in [0, 1]$ evolves as:

$$
x_t^i = \begin{cases}
\emptyset, & 0 < t < T_1^i, \\
m, & T_1^i \le t < T_2^i, \\
x_1^i, & T_2^i \le t \le 1,
\end{cases}
$$

where $m$ denotes the special mask symbol and $\emptyset$ represents token removal. The sequence $x_t$ is obtained by concatenating all non-empty $x_t^i$.

Thus, in FlexMDM, there are two complementary tasks:

1. **Unmasking Task.** Given a partially masked sequence $x_t$, the model predicts the ground-truth token for each masked position. This is identical to the original MDM formulation. Specifically, a diffusion language model $f_\theta$ is trained to approximate the posterior distribution over clean tokens.

2. **Insertion Task.** Beyond unmasking, FlexMDM introduces the additional task of inserting tokens into the sequence. For this purpose, an auxiliary network $g_\theta$ is trained to predict the expected

number of mask tokens that should be inserted before each existing token. During inference, this prediction is combined with a Poisson distribution to determine the actual number of insertions:

$$k_i \sim \text{Poisson}(r \cdot g_\theta(x_t, t)[i]),$$

where $r$ is a scaling factor. The sampled $k$ tokens are inserted as mask symbols $m$, thereby extending the sequence length.

At inference time, sequence generation alternates between the two tasks:

1. **Insertion:** For each position $i$, sample $k_i \sim \text{Poisson}(r \cdot g_\theta(x_t, t)[i])$ and insert $k_i$ mask tokens into the sequence.

2. **Unmasking:** Apply $f_\theta$ to predict clean tokens at masked positions and replace them accordingly.

This insertion–unmasking cycle is repeated iteratively until the sequence converges to a fully unmasked form, thereby producing variable-length outputs while preserving the any-order property of diffusion-based inference.

### B.3  Diffusion with Optimal Transport Position Coupling

At each timestep $t$, DDOT can output both (i) the predicted token value distribution and (ii) the velocity of token positions:

$$s_\theta(x_t, t) \quad \text{and} \quad v_\theta(z_t, t), \tag{86}$$

where $x_t$ are token values and $z_t \in [-1, 1]^L$ are continuous position variables. The extra velocity of token position is predicted by and additional linear head. The token positions are diffused in continuous space from an initial distribution $z_T \sim U(-1, 1)^L$ to the ground-truth positions $z_0$. The token denoising objective follows the score entropy loss from SEDD (Lou et al., 2023), while the position denoising is learned via a weighted mean-squared error loss:

$$\mathcal{L}_{\text{pos}}(\theta) = \mathbb{E}_{(z,t)} \left[ Q_t(x_t, y) \left\| v_\theta(z_t, t) - (z_0 - z_T) \right\|^2 \right]. \tag{87}$$

## C  Training Techniques

### C.1  Masking Scheduling Technique

#### C.1.1  Additional Uniform Masking Scheduling strategies

- **Geometric Schedule (Lou et al., 2023; Shi et al., 2024):**

$$\alpha_t = \exp\left(-\bar{\beta}_{\min}^{1-t} \bar{\beta}_{\max}^t\right), \tag{88}$$

$$w_t = \left( \frac{\exp\left(-\bar{\beta}_{\min}^{1-t} \bar{\beta}_{\max}^t\right)}{1 - \exp\left(-\bar{\beta}_{\min}^{1-t} \bar{\beta}_{\max}^t\right)} \right) \bar{\beta}_{\min}^{1-t} \bar{\beta}_{\max}^t \log\left( \frac{\bar{\beta}_{\min}}{\bar{\beta}_{\max}} \right). \tag{89}$$

- **Polynomial Schedule (Shi et al., 2024):**

$$\alpha_t = 1 - t^r, \tag{90}$$

$$w_t = -\frac{r}{t}. \tag{91}$$

### C.1.2 Token-wise Masking Scheduling

The Spindle-shaped Noise Schedule defines a custom corruption probability $\alpha_t^i$ for each token position $i$ at timestep $t$, determined by:

$$\alpha_t^i = 1 - \frac{t}{T} - S(t) \cdot \tilde{H}(x_0^i), \tag{92}$$

$$S(t) = \lambda \sin\left(\frac{t\pi}{T}\right), \tag{93}$$

$$\tilde{H}(x_0^i) = 1 - \frac{\sum_{j=1}^n H(x_0^j)}{n \cdot H(x_0^i)}, \tag{94}$$

where $H(x_0^i)$ denotes the entropy of the $i$-th token, measuring its information content, $S(t)$ is a sinusoidal scaling function that ensures zero informativeness contribution at $t = 0$ and $t = T$, $\lambda$ is a hyperparameter controlling the strength of entropy influence.

### C.2 Distillation through Dimensional Correlations

The distillation loss conveys the probabilistic knowledge of a teacher model that performs multi-step denoising to a student model designed for fewer-step generation. This is achieved by aligning the student's predicted posterior with that of the teacher at an intermediate noise level $\delta$. Formally, the loss is defined as:

$$\mathcal{L}_{\text{distil}}(\theta) = \mathbb{E}_{x_\delta \sim r_\delta}\left[D_{\text{KL}}\left(p_{0|\delta}^\psi(\cdot \mid x_\delta) p_{0|\delta}^\theta(\cdot \mid x_\delta)\right)\right] \tag{95}$$

where $p_{0|\delta}^\psi(\cdot|x_\delta)$ is the posterior distribution over clean data $x_0$ given intermediate noisy input $x_\delta$ under the teacher model; $p_{0|\delta}^\theta(\cdot|x_\delta)$ is the student model's predicted posterior; $r_\delta$ is a reference distribution over noisy states (typically chosen to match the forward diffusion at timestep $\delta$). This loss encourages to transfer the full-step generative knowledge from teacher to student by matching posteriors.

The consistency loss enforces that the student model produces stable predictions across varying intermediate noise levels. Specifically, if $x_t$ is a noisy sample at step $t$, there should be agreement between: 1). First denoising $x_t \rightarrow x_u$ via the teacher, then $x_u \rightarrow x_s$ via the student; 2). Directly predicting $x_t \rightarrow x_s$ via the student. This intuition is captured by the following KL divergence:

$$\mathcal{L}_{\text{consis}}(\theta) = \mathbb{E}_{x_t \sim r_t}\left[D_{\text{KL}}\left((p_{s|u}^\theta \circ p_{u|t}^\psi)(\cdot \mid x_t)\| p_{s|t}^\theta(\cdot \mid x_t)\right)\right] \tag{96}$$

where $p_{u|t}^\psi(x_u|x_t)$ is the teacher's distribution from timestep $t$ to $u$; $p_{s|u}^\theta(x_s|x_u)$ is the student's distribution from $u$ to $s$; $p_{s|t}^\theta(x_s|x_t)$ is the direct prediction by the student; $(p_{s|u}^\theta \circ p_{u|t}^\psi)(\cdot|x_t)$ denotes the composite distribution over $x_s$ via intermediate $x_u$. This loss enforces functional coherence across generation paths, capturing multi-dimensional correlations without assuming independence.

### C.3 Input Discrepancy Between the Training and Testing

Asada & Miwa (2025) mentioned a discrepancy between training and inference of dLLM. During training, the model receives ground-truth noisy tokens as input, while at inference time, the inputs of the model are the previously predicted tokens. To address this, Asada & Miwa (2025) propose the two-step loss.

Let $x_0$ denote the ground truth sequence. During training, a time step $t$ is randomly selected, and a noised version $x_t$ is generated from $x_0$ via a diffusion process. The model then predicts the original sequence $\hat{x}_0 = f_\theta(x_t)$. Subsequently, a second input $\hat{x}_{t-1}$ is generated by applying the forward diffusion transition matrixto the predicted sequence $\hat{x}_0$ and the model again attempts to recover the ground truth $\hat{\hat{x}}_0 = f_\theta(\hat{x}_{t-1})$. The two-step loss is calculated between the ground truth $x_0$ and the twice-denoised output $\hat{\hat{x}}_0$.

To ease training in early stages, the model does not always use this two-step strategy. Instead, a mixed strategy is adopted. With probability $1 - p_k$, the two-step strategy is used and the loss is evaluated between $\hat{\hat{x}}_0$ and $x_0$. With probability $p_k$, the conventional one-step strategy is used and the loss is evaluated between $\hat{x}_0$ and $x_0$. $p_k$ is set to be linearly increasing along the training step $k$.

### C.4 Reinforcement Learning

### C.4.1 Diffusion-based GRPO

UniGRPO in (Yang et al., 2025b) adapts policy optimization to the diffusion setting by combining structured noising, likelihood approximation, and clipped policy gradients. Let $q$ and $\{o_i\}_{i=1}^{G}$ denote a query and a batch of responses, respectively. For each response $o_i$, UniGRPO samples a masking ratio $p_i \sim U[0,1]$ and create a perturbed version $\tilde{o}_{i,p}$ by masking tokens. The token-level likelihood and the sequence-level likelihood are approximated as

$$\pi'_\theta(o_{i,t} \mid q, \tilde{o}_{i,p}, p_i) = \mathbb{E}_{p_i}\big[\mathbf{1}[\tilde{o}_{i,t,p} = \texttt{[M]}] \log p_\theta(o_{i,t,p} \mid q)\big], \tag{97}$$

$$\pi'_i = \frac{1}{M} \sum_{o_{i,t} \in M} \log p_\theta(o_{i,t} \mid q). \tag{98}$$

Following GRPO, UniGRPO loss is an integration of the clipped surrogate rewards $R_{i,t}$ with the KL regularization:

$$\mathcal{J}_{\text{UniGRPO}}(\theta) = \mathbb{E}\left[\frac{1}{G}\sum_{i=1}^{G}\frac{1}{|o_i|}\sum_{t=1}^{|o_i|} R_{i,t} - \beta D_{\text{KL}}(\pi'_\theta \| \pi'_{\text{ref}})\right]. \tag{99}$$

### C.4.2 Variance-Reduced Preference Optimization

LLaDA 1.5 (Zhu et al., 2025a) introduces the Variance-Reduced Preference Optimization (VRPO) which replaces the intractable log-likelihoods in DPO with Evidence Lower Bounds (ELBOs)

$$\mathcal{B}_\pi(y \mid x) \triangleq \mathbb{E}_{t\sim\mathcal{U}[0,1]}\mathbb{E}_{y_t\sim q(y_t\mid t,y,x)}\ell_\pi(y_t, t, y \mid x) \tag{100}$$

$$\leq \log \pi(y \mid x), \tag{101}$$

where $\ell_\pi$ is the mask prediction loss. To reduce estimator variance, VRPO introduces (i) increased sampling budgets, (ii) full sampling budget over timesteps, and (iii) antithetic sampling that shares noise between $\pi_\theta$ and $\pi_{\text{ref}}$.

### C.4.3 Stepwise Decomposition Preference Optimization

A central challenge in reinforcement learning alignment for discrete diffusion models is how to propagate reward information through the entire denoising trajectory. Stepwise Decomposition Preference Optimization (SDPO) (Han et al., 2025) reformulates trajectory alignment into a collection of tractable per-step alignment objectives.

The standard KL-regularized reward optimization objective for diffusion trajectories is:

$$\max_{p_\theta} \mathbb{E}_{p_\theta(x_{0:T}\mid c)}\big[r(x_0, c)\big] - \beta D_{\text{KL}}\big[p_\theta(x_{0:T} \mid c) \,\|\, p_{\text{ref}}(x_{0:T} \mid c)\big], \tag{102}$$

where $c$ is the conditioning context, $r(x_0, c)$ is a reward function on the clean sequence, and $p_{\text{ref}}$ is the reference model. However, this requires sampling and scoring the entire trajectory, which is computationally prohibitive. Instead of optimizing the whole trajectory, SDPO aligns the *per-step posterior*:

$$p_\theta(x_0 \mid x_t, c), \tag{103}$$

which admits exact likelihood evaluation under masked diffusion models. The stepwise alignment objective is:

$$\max_{p_\theta} \mathbb{E}_{p_\theta(x_0\mid x_t,c)}[r(x_0, c)] - \beta_t D_{\text{KL}}\big[p_\theta(x_0 \mid x_t, c)\|p_{\text{ref}}(x_0 \mid x_t, c)\big], \tag{104}$$

with step-dependent regularization $\beta_t = \beta/w(t)$. The work shows that this step-wise optimization is equivalent to a distribution matching problem, which can be used to further simplify the loss function.

### C.4.4 Weighted Policy Optimization

Weighted Policy Optimization (wd1) (Tang et al., 2025b) reformulates the reinforcement learning objective as a *weighted likelihood* objective.

Under the reverse-KL–regularized policy optimization, the target solution $\pi^*$ has the closed form. wd1 minimizes the KL divergence $D_{\mathrm{KL}}(\pi^* \| \pi_\theta)$, which reduces to a weighted negative log-likelihood:

$$L_{\mathrm{wd1}}(\theta) = -\mathbb{E}_{q \sim D, \{o_i\} \sim \pi^{\mathrm{ref\text{-}old}}} \left[ \sum_{i=1}^{G} w(q, o_i) \cdot \log \pi_\theta(o_i \mid q) \right]. \tag{105}$$

To address the issue where samples with small advantages receive disproportionately low weights, the weights are defined as:

$$w(q, o_i) = -w^+(q, o_i) + w^-(q, o_i), \tag{106}$$

$$w^+(q, o_i) = \frac{\exp(\psi A_i)}{\sum_{j=1}^{G} \exp(\psi A_j)}, \tag{107}$$

$$w^-(q, o_i) = \frac{\exp(-\psi A_i)}{\sum_{j=1}^{G} \exp(-\psi A_j)}, \tag{108}$$

where $A_i$ is the centered reward and $\psi$ is a scaling factor.

### C.4.5 Diffusion Chain of Lateral Thought

Diffusion Chain of Lateral Thought (DCoLT) (Huang et al., 2025b) introduces a Unmask Policy Module (UPM), which is trained via reinforcement learning to control the decoding order. The UPM learns a ranking-based policy over masked tokens:

$$h_{\theta,t}^i = \mathrm{UPM}(x_{t-1}, t, m_i), \tag{109}$$

where $h_{\theta,t}^i$ is the predicted score for token $i$ at step $t$, and $m_i$ is its mask indicator. A Plackett–Luce distribution is then used to sample a top-$K$ unmasking set $U_t$. Once $U_t$ is selected, the model predicts token values for those positions using the standard diffusion langauge model. The UPM is implemented as a lightweight transformer block attached to diffusion langauge model.

## D Decoding Techniques

### D.1 Unmasking and Remasking for Continuous-Time Models

### D.1.1 Continuous-Time Unmasking (Flow Matching)

In continuous-time inference under the discrete flow matching framework (e.g., FUDOKI (Wang et al., 2025c)), unmasking is modeled as a stochastic jump process along a probability path.

Let $\mathbf{x}_t$ denote the current sequence state at time $t \in [0, 1]$, and let $\mathbf{x}_1$ be the target sequence. For each token position $i$, the update from $\mathbf{x}_t$ to $\mathbf{x}_{t+h}$ is governed by:

1. Sample a predicted target $\hat{x}_1^{(i)} \sim p_\theta(x_1^{(i)} \mid \mathbf{x}_t)$;

2. Compute a total transition rate

$$\lambda^{(i)} = \sum_{x \neq x_t^{(i)}} u_t^{(i)}(x, x_t^{(i)} \mid \hat{x}_1^{(i)});$$

3. Draw a random variable $Z \sim \mathcal{U}(0, 1)$;

4. If $Z \leq 1 - e^{-h\lambda^{(i)}}$, update $x_t^{(i)}$ by sampling from:

$$x_{t+h}^{(i)} \sim \frac{u_t^{(i)}(x, x_t^{(i)} \mid \hat{x}_1^{(i)})}{\lambda^{(i)}}.$$

This process dynamically determines which tokens to update (i.e., unmask) based on local transition rates. The higher the rate $\lambda^{(i)}$, the more likely token $i$ will jump to a new value, allowing the model to continuously refine its predictions in a semantically meaningful way.

### D.1.2 Remasking under Discrete Flow Matching.

In the discrete flow matching framework (Gat et al., 2024), remasking is incorporated via a velocity field that interpolates between forward and backward update directions:

$$\bar{u}_t^i(x^i, z) = \alpha_t \hat{u}_t^i(x^i, z) - \beta_t \check{u}_t^i(x^i, z), \tag{110}$$

where $\hat{u}_t^i$ and $\check{u}_t^i$ denote the forward-time and backward-time velocity fields, respectively. This combined velocity $\bar{u}_t^i$ is valid as long as $\alpha_t, \beta_t > 0$ and satisfies the probability flow condition for $t \in (0, 1)$. When $\alpha_t - \beta_t = 1$, each step progresses forward in time with remasking (corrector sampling), enabling iterative refinement. When $\alpha_t - \beta_t = 0$, the model operates in a stationary regime (corrector iteration), reintroducing noise and adjusting tokens within a fixed diffusion step.

## D.2 Guidance Techniques

### D.2.1 Classifier-Free Guidance

Schiff et al. (2025); Nie et al. (2025a) introduce the unsupervised *classifier-free guidance* strategy for discrete diffusion generation. The method performs two forward predictions at each diffusion timestep $t$:

- A **conditional prediction**, conditioned on both the prompt $p_0$ and the noisy response $r_t$;

- An **unconditional prediction**, conditioned on a sequence of mask token $M$ and the same response $r_t$.

The unconditional prediction captures the model's inherent bias in the absence of instruction signals. The final guided prediction is adjusted as:

$$\tilde{p}_\theta(r_0 \mid p_0, r_t) \propto \frac{p_\theta(r_0 \mid p_0, r_t)^{1+w}}{p_\theta(r_0 \mid m, r_t)^w}, \tag{111}$$

where $w$ is a tunable hyperparameter controlling the strength of the guidance. This guidance promotes *text diversity* by reducing the dominance of generic, encouraging prompt-independent responses.

In the continuous-time setting, Nisonoff et al. (2025) provides the formulation of classifier-free guidance applied to the reverse transition matrix

$$\hat{R}_t(x, x') = R_t(x', x) \left( \frac{p_t(x)}{p_t(x')} \right)^w \left( \frac{q_t(x)}{q_t(x')} \right)^{1-w}, \tag{112}$$

where distribution $q_t$ is the reference distribution. Rojas et al. (2025) further demonstrate that applying strong guidance during early decoding can severely degrade sample quality. This degradation arises because high guidance accelerates the unmasking rate, leading to overly confident and premature token predictions. To mitigate this issue, the authors suggests applying a column-wise normalization of the guided transition matrix.

### D.2.2 Classifier Guidance

To improve controllability, for block diffusion model, Schiff et al. (2025); Huang & Tang (2025) introduce the *classifier guidance* framework that integrates class-conditional preferences into the sampling process.

At each diffusion step $t$ for block $b$, the guided reverse process modifies the original sampling distribution $p_\theta$ by incorporating the signal from a classifier $p_\xi$, yielding:

$$p_\gamma(x_b^s \mid x_b^t, x_{<b}, y)$$
$$\propto p_\theta(x_b^s \mid x_b^t, x_{<b}) \cdot p_\xi(y \mid x_b^s, x_b^t, x_{<b})^\gamma, \tag{113}$$

where $y$ is the desired class label and $\gamma$ controls the influence of the classifier. To reduce computational complexity, the method assumes intra-block independence and approximates the classifier as:

$$p_\xi(y \mid x_b^s, x_b^t, x_{<b}) \approx \prod_{\ell=1}^{L} p_\xi(y \mid \hat{x}_{b,t|s}^\ell, x_{<b}), \tag{114}$$

where $\hat{x}_{b,t|s}^\ell$ denotes the sequence with the $\ell$-th token in $x_b^t$ replaced by the candidate token $x_{b,\ell}^s$. This allows the guided probability to be reformulated as:

$$p_\gamma(x_b^s \mid x_b^t, x_{<b}, y) =$$
$$\prod_{\ell=1}^{L} \frac{p_\theta(x_{b,\ell}^s \mid x_b^t, x_{<b}) \cdot p_\xi(y \mid \hat{x}_{b,t|s}^\ell, x_{<b})^\gamma}{\sum_{x'} p_\theta(x' \mid x_b^t, x_{<b}) \cdot p_\xi(y \mid \hat{x}_{b,t|s}'^\ell, x_{<b})^\gamma}. \tag{115}$$

By integrating classifier predictions with the model's native probabilities, this approach enables fine-grained, attribute-conditioned generation across blocks while maintaining computational feasibility.

### D.2.3 Reward Guidance

TESS 2 (Tae et al., 2025) represents a unique approach under the extra-model guidance category by leveraging an external reward model to guide token prediction. The main purpose of this method is to improve the quality of the generated response. Specifically, at each diffusion step, the model output $\hat{S}_\theta$ is first transformed into a token probability distribution:

$$\mathbf{p}_t = \text{softmax}(\hat{S}_\theta), \tag{116}$$
$$\mathbf{c}_w = \mathbf{E}\mathbf{p}_t, \tag{117}$$

where $\mathbf{E}$ is the embedding matrix. The resulting continuous representation $\mathbf{c}_w$ is fed into the reward model, which outputs a scalar reward $R \in \mathbb{R}$.

To maximize this reward, TESS 2 performs gradient ascent on the logits by computing $\nabla_\theta R$ and updates the model output:
$$\hat{S}_\theta := \hat{S}_\theta + \eta \cdot \nabla_\theta R, \tag{118}$$

where $\eta$ is a tunable guidance coefficient. This process is performed during inference only, and no guidance is applied during training. By incorporating gradient signals from the reward model, TESS 2 can steer the generation towards more desirable outputs without modifying the base diffusion model.

### D.2.4 Energy-Based Diffusion

Xu et al. (2025) proposes the Energy-based Diffusion Language Model (EDLM), which augments a pretrained diffusion model $p_\theta(x_0 \mid x_t)$ with an unnormalized energy model $E_\phi(x_0, x_t, t)$, yielding a joint denoising distribution:
$$p_{\theta,\phi}(x_0 \mid x_t) = p_\theta(x_0 \mid x_t) \cdot \frac{\exp(-E_\phi(x_0, x_t, t))}{Z_\phi(x_t)}, \tag{119}$$

where $Z_\phi(x_t)$ is the intractable partition function required for normalization. This residual formulation corrects the denoising distribution by reweighting samples from the diffusion model using the energy function.

The energy function can be derived from either a pretrained autoregressive (AR) model or a finetuned diffusion model. To implement this energy function-based guidance, Xu et al. (2025) adopts an importance sampling strategy. The decoding process at each time step can be summarized as follows.

1. Generate $k$ candidate samples $\{x_0^{(i)}\}_{i=1}^k \sim p_\theta(x_0 \mid x_t)$ using the diffusion model.

2. Compute the unnormalized energy scores $e^{(i)} = E_\phi(x_0^{(i)}, x_t, t)$ for all samples in $\{x_0^{(i)}\}_{i=1}^k$.

3. Sample one $x_0$ from the candidate pool according to importance weights:

$$w^{(i)} = \frac{\exp(-e^{(i)})}{\sum_{j=1}^k \exp(-e^{(j)})}. \tag{120}$$

4. Use the sampled $x_0$ to perform one denoising step via the backward posterior:

$$x_{t-1} \sim q(x_{t-1} \mid x_t, x_0). \tag{121}$$

## E    Quantization

### E.1    DLLMQuant

Xu & Yang (2025) introduces **DLLMQuant**, a framework includes Temporal-Mask Adaptive Sampling (TMAS), Interaction-Aware Activation Quantization (IA-AQ), and Certainty-Guided Quantization (CGQ).

TMAS addresses the calibration challenge in dLLMs by accounting for variations across time steps and masking ratios. Specifically, it divides the iterative generation into blocks and selects calibration inputs at specific time intervals, ensuring coverage of diverse mask ratios across all timesteps.

Quantization errors $L(x_t)$ at time step $t$ in dLLMs accumulate geometrically across denoising steps. Formally, the error propagation can be expressed as:

$$\begin{aligned} L(x_t) &= x_t - \mathrm{Deq}(Q(x_t + L(x_{t+1}))) \\ &= Q_{\mathrm{Model}}(x_{t+1}) - Q_{\mathrm{Model}}(\mathrm{Deq}(Q(x_{t+1}))) \end{aligned} \tag{122}$$

where $Q(\cdot)$ is the quantization process, Deq is the dequantization operation, and $Q_{\mathrm{Model}}$ denotes the quantized model.

A key source of error is the matrix multiplication between softmax outputs and the value matrix in attention. IA-AQ redefines the quantization loss for value matrix $V$ as:

$$L(s) = \left\| \left( \lfloor \tfrac{V-z}{s} \rfloor - V \right) \cdot \mathrm{Deq}(O_{\mathrm{softmax}}) \right\|_F^2, \tag{123}$$

where $O_{\mathrm{softmax}}$ is the softmax output, $z$ is zero-point, and $s$ is the scale factor. The optimal scaling factor is chosen by:

$$s = \arg\min_{\alpha \in \{1.0, 0.8\}} L(\alpha \odot \hat{s}), \quad \hat{s} = \frac{V_{\max} - V_{\min}}{Q_{\max} - Q_{\min}}. \tag{124}$$

In addition, not all tokens equally affect subsequent iterations. Errors on unmasked or low-confidence tokens are small, while masked tokens with high confidence dominate the next step. To account for this, CGQ incorporates token certainty into the Hessian used for weight quantization:

$$H = \left( X \odot (1[X_t = M] + \sqrt{sc_t}) \right) \cdot \left( X \odot (1[X_t = M] + \sqrt{sc_t}) \right)^\top, \tag{125}$$

where $1[X_t = M]$ indicates a custom weighted indicator function, and $sc_t$ is the final confidence score to each token in model output. This certainty-weighted Hessian prioritizes minimizing quantization error on critical masked tokens.

# F    Discussion on the Benchmark Results in Tab. 1 and 2

These benchmark numbers should be interpreted as indicative evidence of recent progress rather than as direct apples-to-apples comparisons.

For LLaDA series, LLaDA 8B is trained from scratch as a diffusion language model on about 2.3 trillion tokens and then supervised fine-tuned on 4.5 million instruction-response pairs, without reinforcement learning. In contrast, LLaDA 1.5 keeps the same LLaDA backbone but adds a preference-optimization stage, training on 350K preference pairs with Variance-Reduced Preference Optimization (VRPO). Therefore, the improvement of LLaDA 1.5 over LLaDA should not be attributed only to the base diffusion architecture, but also to its additional preference-alignment procedure.

Dream 7B differs from LLaDA in both initialization and data scale. Dream is initialized from the autoregressive Qwen2.5 model and is further trained as a diffusion language model using approximately 580 billion tokens, followed by supervised fine-tuning on 1.8 million instruction-response pairs. Thus, compared with LLaDA, Dream uses much less diffusion pretraining data but benefits from autoregressive weight initialization and context-adaptive noise scheduling. This makes its benchmark numbers informative, but not directly comparable to those of a diffusion model trained from scratch.

For code generation models, DiffuCoder 7B is trained on 130 billion code tokens and further introduces coupled-GRPO, a diffusion-native reinforcement learning method designed to reduce the variance of likelihood estimates during policy optimization. Consequently, its code benchmark performance reflects not only the discrete diffusion generation paradigm but also domain-specific code training and additional RL-based optimization. This differs from general-purpose dLLMs such as LLaDA and Dream.

For multimodal models, Dimple adopts a hybrid autoregressive-then-diffusion training strategy, and its training data are comparable to the LLaVA-NEXT setting, with roughly 0.6M image-text pretraining samples and about 1.0M visual instruction-following samples. LaViDa, by contrast, builds on diffusion language backbones such as LLaDA or Dream and uses a two-stage multimodal training process with approximately 558K image-text pairs for visual alignment and about 1M visual instruction-following examples. LLaDA-V is built on LLaDA-8B and follows a multi-stage visual instruction tuning pipeline with datasets such as LLaVA-NeXT, MAmmoTH-VL, and VisualWebInstruct. Therefore, although Dimple, LaViDa, and LLaDA-V are all diffusion-based multimodal models, their results are influenced by different vision encoders, language backbones, data mixtures, and training schedules.

# G    Limitations and Future Directions

### G.1    Limitations

One limitation is computational inefficiency. Masked diffusion models recompute attention and feed-forward layers for every masked token at each denoising step, even though many tokens cannot be effectively decoded at that stage and thus do not contribute to meaningful progress. This inefficiency is exacerbated in block diffusion settings, where the model is initialized with a large number of masked tokens but can resolve only a subset per iteration, leading to redundant computation across steps. This limitation raises an open problem on adaptive computation for diffusion language models. Let $x_t$ denote the sequence at denoising step $t$, and let $L$ be the sequence length. An ideal sampler should update only a subset $S_t$ of uncertain or high-value positions, where $|S_t| \ll L$, instead of recomputing all positions at every step. Future work may therefore explore sparse denoising, token-level early stopping, dynamic step allocation, block-wise computation reuse, and cache-aware diffusion decoding. These directions are important for reducing redundant computation while preserving the global refinement capability of diffusion models.

A second limitation is that standard discrete diffusion training is more costly than autoregressive training. Efficient-DLM observes that existing diffusion language models either fail to outperform autoregressive models in speed or are constrained to small scales due to high training costs Fu et al. (2025b). A controlled comparison further shows that masked diffusion requires more steps to converge despite similar per-step throughput Vicentino (2026). This limitation motivates the open problem of data-efficient and compute-efficient diffusion training. Future research should investigate how to better reuse pretrained autoregressive models, design curriculum-based masking schedules, improve timestep sampling, and distill multi-step diffusion models into fewer-step samplers. Autoregressive–diffusion hybrid models are especially promising in this direction: autoregressive models provide strong pretrained likelihood modeling and efficient local token prediction, while diffusion models provide parallel refinement and global correction. Thus, a useful future direction is to use autoregressive components as initialization or local executors, and diffusion components for global planning and iterative revision.

A third limitation is the sensitivity of diffusion language models to decoding heuristics. Masked diffusion models rely on heuristics such as confidence or margin to determine the unmasking order, and early decisions can propagate errors. The Lookahead Unmasking study demonstrates that performance is highly dependent on the unmasking schedule and that heuristic choices can lead to error accumulation Kim et al. (2026). This limitation points to the open problem of principled decoding policy learning. Instead of selecting the unmasking set $S_t$ by a fixed rule such as confidence, margin, or entropy, future work may learn a policy $\pi_\phi(S_t \mid x_t, t)$ that decides which positions should be updated, kept, or remasked at each denoising step. Such policies could be optimized using validation likelihood, task rewards, uncertainty calibration, or downstream evaluation metrics. A learned decoding policy may reduce early commitment to incorrect tokens and improve robustness across tasks, sequence lengths, and decoding budgets.

A fourth limitation is that discrete diffusion requires the output length to be predetermined at sampling time. DAEDAL shows that short sequence lengths constrain reasoning and reduce performance, whereas long lengths introduce computational waste and may degrade quality; diffusion language models lack the dynamic length adaptation available to autoregressive models Lee et al. (2025b). This limitation leads to the open problem of dynamic length modeling. Future work should study flexible-length diffusion, insertion–deletion diffusion, explicit length-prediction modules, and hybrid decoding schemes that allow the generation canvas to expand or contract during inference. Hybrid discrete–continuous diffusion is also a promising direction, since continuous latent variables may help represent global generation attributes such as expected length, response layout, and reasoning structure.

A fifth limitation is the immaturity of training and serving pipelines. Standard reinforcement learning methods such as GRPO and VRPO, which effectively align autoregressive models, lead to instability and reward collapse when applied to diffusion language models due to intractable sequence probabilities. Although StableDRL proposes an alternative, it highlights the fragility of reinforcement learning in this setting Li et al. (2025b). This limitation corresponds to the open problem of diffusion-native post-training and deployment. In autoregressive models, the sequence log-likelihood is tractable and decomposes as

$$\log p_\theta(y \mid x) = \sum_i \log p_\theta(y_i \mid y_{<i}, x).$$

In diffusion models, this factorization is not directly available. Sequence likelihood is usually approximated by an evidence lower bound or Monte Carlo estimator, which may introduce high variance during reinforcement learning or preference optimization. Future work should therefore develop lower-variance likelihood estimators, sequence-level preference objectives, stable reward modeling, and diffusion-native policy optimization methods. On the serving side, deployment systems should jointly optimize decoding quality, latency, memory usage, and the number of denoising steps, rather than optimizing only a single efficiency metric.

A sixth limitation concerns scaling behavior at frontier model sizes. Prior work Zhong et al. (2026) indicates that diffusion LLMs exhibit scaling exponents comparable to autoregressive models but suffer from large constant-factor gaps; they require more computation to match autoregressive perplexity, and perplexity alone is insufficient for fair cross-paradigm comparison. This limitation raises the open

problem of establishing principled scaling laws and evaluation protocols for diffusion language models. Future research should jointly analyze accuracy, inference latency, number of denoising steps, training FLOPs, memory usage, likelihood quality, and controllability. A more complete evaluation objective may take the form

$$\text{Utility} = \text{Performance} - \lambda_1 \cdot \text{Latency} - \lambda_2 \cdot \text{Compute},$$

where $\lambda_1$ and $\lambda_2$ reflect deployment constraints. Such evaluation would better capture the trade-off between quality, efficiency, and controllability in diffusion-based generation.

Overall, these limitations suggest several concrete future research directions: efficient sparse and cache-aware decoding, data-efficient diffusion training, principled unmasking and remasking policies, dynamic length generation, diffusion-native reinforcement learning and preference alignment, and fair scaling-law evaluation. Addressing these open problems will be essential for making discrete diffusion language and multimodal language models practical at large scale.

### G.2 Other Future Directions

**Autoregressive–diffusion hybrids for reasoning and planning**. Autoregressive (AR) models excel at likelihood modeling and sequential reasoning, whereas discrete diffusion models enable parallel generation and intrinsic self-correction. Recent work explores hybrid architectures in which diffusion models perform high-level planning and AR models execute it. The Planner–Executor framework couples a discrete diffusion planner with an AR executor; by communicating in latent space, it achieves substantial accuracy gains while reducing reasoning token usage Berrayana et al. (2025). Efficient-DLM converts pretrained AR models into diffusion models using block-wise attention and a continuous pretraining schedule, preserving pretrained weights and achieving approximately $4.5\times$ higher throughput with improved accuracy Fu et al. (2025c). Block Diffusion interpolates between diffusion and autoregressive generation via block-wise diffusion with key–value caching and data-driven noise schedules, supporting flexible sequence lengths and achieving state-of-the-art performance among diffusion models Arriola et al. (2025b). These results suggest that diffusion can provide global planning while autoregressive decoding enables local refinement. Future work may investigate unified architectures that dynamically switch between diffusion and autoregression based on task complexity or context length.

**Hybrid discrete–continuous diffusion**. Another promising direction is to augment discrete diffusion with continuous latent channels or hybrid corruption processes. Latent-Augmented Discrete Diffusion Models combine masked token diffusion with continuous latent diffusion carrying cross-token information, enabling joint or sequential denoising and improving unconditional generation under limited sampling budgets Shariatian et al. (2026). Loopholing Discrete Diffusion introduces a deterministic latent pathway that preserves distributional information across steps, reducing perplexity by up to 61% and improving reasoning accuracy Jo et al. (2026). Residual Context Diffusion recycles representations of remasked tokens by converting them into residuals and reinjecting them into the denoising process, improving accuracy by 5–10 points and reducing denoising steps by $4–5\times$ Hu et al. (2026). CANDI decouples discrete identity corruption from continuous rank degradation, resolves temporal inconsistencies between them, and enables gradient-based guidance with off-the-shelf classifiers, outperforming masked diffusion under low noise-free evaluations Pynadath et al. (2025). Soft-Masked Diffusion Language Models replace binary masking with a soft mask that blends the mask embedding with top-$k$ predictions, preserving partial information and improving perplexity and MAUVE scores Hersche et al. (2026). Time-Annealed Perturbation Sampling is a training-free inference strategy that perturbs early denoising steps to encourage semantic exploration and gradually reduces perturbations, enhancing diversity without sacrificing quality Wu et al. (2026). Collectively, these approaches indicate that hybrid discrete–continuous diffusion can capture both global structure and fine-grained detail.

