# OpenReview forum: "Discrete Diffusion in Large Language and Multimodal Models: A Survey"
_TMLR — Under review for TMLR_

### Review · Reviewer_kKT6 · 2026-03-08

**Summary Of Contributions:**

This paper provides a comprehensive survey of discrete diffusion large language and multimodal models (dLLMs/dMLLMs). It systematically reviews their mathematical foundations, categorizes representative architectures, and benchmarks model performance against autoregressive baselines. Also, it analyzes crucial training strategies, advanced inference techniques, quantization challenges, unique safety vulnerabilities, and diverse real-world applications.

**Audience:**

Yes

**Audience Explanation:**

This survey highlights recent progress in the research community, providing a timely summary of the rapidly evolving discrete diffusion LLM and LLM landscape.

**Claims And Evidence:**

Yes

**Claims Explanation:**

This paper provides a survey of recent advancements in the research community.

**Requested Changes:**

1. Later sections (e.g., Secs. 7–9) are rather short compared with the longer ones at the beginning, which creates an imbalance in the distribution of text and reading load. Consider expanding these later sections to provide more depth or examples, or combine shorter subsections to balance the overall structure.

2. Distillation seems quite important for frontier competitor LLMs to catch up with one another. It would be interesting to see more extension on the role of distillation for discrete LLMs.

3. What future research directions can readers learn from this paper, beyond its organized listing of existing works?

---

> ### Author Response · Authors · 2026-04-07
> **Rebuttal to Review Part 1**
>
> ### 1. Response to comment 1: balancing Sections 7–9
>
> We appreciate the reviewer’s observation that the original Sections 7–9 were too brief compared with the earlier parts of the survey. In the revised manuscript, we will extend these sections to provide a detailed discussion of quantization, safety and applications of discrete diffusion language models (dLLMs).
>
> ##### 1.1 Quantization and safety
>
> We will add another paper on quantization and six papers on safety. Furthermore, because the corpus of safety and quantization papers is smaller than that of training or decoding, we have merged Sections 7 and 8 into a unified section titled "Quantization and safety."
>
> Below is a summary of the content we plan to add to the Quantization and safety section:
>
> **Quantization**   Quant‑dLLM [1] introduces a 2‑bit post‑training quantization scheme using masked calibration simulation, a data‑aware any‑order quantizer and an adaptive blockwise mixed‑precision schedule. These techniques improve 2‑bit quantization accuracy compared with autoregressive baselines.
>
> **Safety**  Watermarking Diffusion Language Models applies a watermark in expectation and biases token sampling to ensure more than 99 % detection while minimally affecting quality [2]. DMark introduces predictive and bidirectional watermarking that achieves 92–99.5 % detection at a 1 % false‑positive rate without degrading text quality [6], and Watermarking Discrete Diffusion Language Models employs a distribution‑preserving Gumbel‑max trick with position‑seeded randomness to enable distortion‑free watermarking [7]. DiffuGuard identifies denoising‑path dependence and harmful bias in greedy remasking and proposes a dual‑stage defence based on stochastic annealing and block‑level audit; this reduces jailbreak success rates from 47.9 % to 14.7 % [3]. A2D aligns dLLMs to emit a refusal token when harmful content appears, reducing attack success rates to near zero and allowing early termination to speed inference [4]. A study of priming vulnerability shows that inserting an affirmative token early in the denoising trajectory can steer outputs toward harmful content and proposes training models to generate safe responses from contaminated states [5].

---

> ### Author Response · Authors · 2026-04-07
> **Rebuttal to Review Part 2**
>
> ##### 1.2 Applications
>
> We will further expand the application section by incorporating an additional 11 papers, covering a wide range of domains, including GUI grounding, vision–language–action (VLA) modeling, low-resource language modeling, aircraft design, biological sequence modeling, autonomous driving, molecular generation, motion understanding, music generation, audio understanding, and sign language generation. Below we summarize the added works, organized by category in a manner consistent with the original manuscript.
>
> GUI grounding Towards GUI Agents explores the application of discrete vision–language diffusion models (DVLMs) to GUI grounding tasks. By adapting LLaDA-V for action and bounding-box prediction and introducing a hybrid masking schedule, the method improves grounding accuracy and demonstrates competitive performance with autoregressive models, highlighting the potential of diffusion-based GUI agents [8].
>
> Vision–language–action (VLA) Fast-dVLA proposes a parameter-space decoupling strategy to enhance pretrained VLA models without incurring the computational overhead of auxiliary objectives. By learning and merging capability vectors, the method achieves comparable performance to advanced fine-tuning approaches while significantly reducing training cost [9].
>
> Low-resource language modeling Diffutron introduces a masked diffusion language model tailored for Turkish, a morphologically rich language. Leveraging LoRA-based continual pretraining and progressive instruction tuning, the model achieves competitive performance despite its compact size, demonstrating the effectiveness of diffusion modeling in low-resource settings [10].
>
> Engineering design Do Diffusion Models Dream of Electric Planes? presents a hierarchical diffusion framework for eVTOL aircraft design. By combining discrete topology generation with continuous parameter modeling, the approach captures physical design principles while substantially accelerating simulation-based inference [11].
>
> Biological sequence modeling D3LM reformulates DNA modeling as a masked diffusion process, enabling unified bidirectional understanding and generation. It outperforms comparable transformer-based baselines and achieves strong results in regulatory sequence generation, suggesting diffusion as a promising paradigm for biological foundation models [12].
>
> Autonomous driving MVLAD-AD proposes a masked vision–language–action diffusion framework for end-to-end driving. By introducing discrete action tokenization and geometry-aware embeddings, it improves planning precision and efficiency while providing interpretable decision-making [13].
>
> Molecular generation MolHIT introduces a hierarchical discrete diffusion framework for molecular graph generation. By incorporating chemical priors and structured atom encoding, it achieves near-perfect chemical validity and state-of-the-art performance on the MOSES benchmark [14].
>
> Motion generation and understanding DiMo extends masked diffusion modeling to unified text–motion understanding and generation. Through iterative token refinement and enhanced token representations, it supports multiple tasks within a single framework and achieves strong performance across motion benchmarks [15].
>
> Music generation D3PIA addresses piano accompaniment generation using discrete diffusion with neighborhood attention. It better preserves harmonic structure than transformer and continuous diffusion baselines and produces more coherent musical outputs [16].
>
> Audio understanding DIFFA-2 develops a practical diffusion-based large audio language model with improved semantic–acoustic alignment and multi-stage training. It achieves competitive performance with autoregressive models while maintaining efficiency under limited training resources [17].
>
> Sign language generation MaDiS proposes a masked diffusion model for sign language generation, enabling bidirectional context modeling and parallel token generation. With multi-level cross-modal pretraining and efficient decoding strategies, it achieves state-of-the-art performance while reducing inference latency [18].

---

> ### Author Response · Authors · 2026-04-07
> **Rebuttal to Review Part 3**
>
> ### Response to comment 2: extending the distillation section
>
> The reviewer notes that distillation plays a key role in making frontier dLLMs practical. We will expand Section 6 to survey a broad range of distillation methods. We will add an introduction to the following methods.
>
> Learnable Sampler Distillation trains a few‑step student sampler by aligning its intermediate score trajectory with that of a high‑quality teacher and extends to non‑uniform time schedules (LSD+) to further improve sampling quality [19]. Ultra‑Fast Language Generation via Discrete Diffusion Divergence Instruct minimises an integral KL divergence between student and teacher while matching intermediate states and using grouped reward normalisation; this method achieves up to 64× speed‑ups with negligible entropy loss [20]. Distillation of Discrete Diffusion by Exact Conditional Distribution Matching decomposes the reverse conditional distribution and matches conditional distributions to train one‑step and few‑step students [21]. CD4LM introduces discrete‑space consistency distillation and confidence‑adaptive decoding to produce trajectory‑invariant students and allocate computation based on token confidence, yielding a 5× speed‑up on GSM8K and a 3× average speed‑up across tasks [22]. dUltra formulates distillation as an on‑policy reinforcement learning problem and jointly trains the diffusion LLM with an unmasking planner to achieve superior accuracy–efficiency trade‑offs [23]. T3D uses trajectory self‑distillation with direct discriminative optimisation to narrow the gap between few‑step and full‑step decoding [24]. Discrete Moment Matching Distillation adapts moment matching to discrete spaces, maintaining high quality and diversity and sometimes enabling the student to outperform its teacher [25]. IDLM extends inverse distillation to discrete diffusion models, proves the uniqueness of the optimisation and introduces gradient‑stable relaxations; experiments show a 4–64× reduction in inference steps without sacrificing perplexity [26]. By covering trajectory alignment, conditional distribution matching, reinforcement learning, moment matching and inverse optimisation, the revised Section 6 provides a comprehensive taxonomy of distillation techniques and clarifies their respective advantages and limitations.

---

> ### Author Response · Authors · 2026-04-07
> **Rebuttal to Review Part 4**
>
> ### Response to comment 3: future research directions
>
> We thank the reviewer for the question. Section F of the supplementary material outlined several future directions. We will also update this section to incorporate a discussion of future research directions based on recent developments. We plan to add the following content:
>
> **Autoregressive–diffusion hybrids for reasoning and planning.**  Autoregressive (AR) models excel at likelihood modelling and sequential reasoning, whereas discrete diffusion models offer parallel generation and built‑in self‑correction. Recent work explores hybrid architectures in which diffusion models plan high‑level reasoning and AR models execute it. The Planner–Executor framework couples a discrete diffusion planner with an AR executor; by communicating in latent space it achieves significant accuracy improvements and reduces the number of tokens used for reasoning [27]. Efficient‑DLM converts pretrained AR models into diffusion models using block‑wise attention and a continuous pretraining schedule; this preserves pretrained weights and yields roughly 4.5× higher throughput with improved accuracy [28]. Block Diffusion interpolates between diffusion and autoregressive generation through block‑wise diffusion with key–value caching and data‑driven noise schedules, supporting flexible sequence lengths and achieving state‑of‑the‑art performance among diffusion models [29]. These studies suggest that diffusion can provide global planning while autoregressive decoders offer local refinement. Future work could explore unified architectures that dynamically switch between diffusion and autoregression based on task complexity or context length.
>
> **Hybrid discrete–continuous diffusion.**  Another promising direction is to augment discrete diffusion with continuous latent channels or hybrid corruption processes. Latent‑Augmented Discrete Diffusion Models couple masked token diffusion with a continuous latent diffusion carrying cross‑token information; they perform joint or sequential denoising and improve unconditional generation at low sampling budgets [30]. Loopholing Discrete Diffusion introduces a deterministic latent pathway that preserves distributional information across steps, reducing perplexity by up to 61 % and improving reasoning accuracy [31]. Residual Context Diffusion recycles representations of remasked tokens by converting them into residuals and injecting them back into the denoising process, boosting accuracy by 5–10 points and reducing denoising steps by 4–5× [32]. CANDI decouples discrete identity corruption from continuous rank degradation, resolves temporal dissonance between the two corruptions and enables gradient‑based guidance with off‑the‑shelf classifiers, outperforming masked diffusion at low noise‑free evaluations [33]. Soft‑Masked Diffusion Language Models replace binary masking with a soft mask blending the mask embedding with top‑k predictions, preserving partial information and improving perplexity and MAUVE scores [34]. Time‑Annealed Perturbation Sampling is a training‑free inference strategy that perturbs early denoising steps to encourage semantic branching and gradually reduces perturbations, enhancing diversity without sacrificing quality [35]. Collectively these works indicate that hybrid discrete–continuous diffusion can capture both global structure and fine‑grained details. Our updated future‑work section argues that combining diffusion with autoregressive or continuous components could overcome current limitations such as high sampling cost, sensitivity to denoising order and difficulty in modelling long‑range dependencies. Furthermore, continuous spaces are more conducive to modelling images; discrete–continuous combinations capable of generating both images and text represent another direction of research.

---

> ### Author Response · Authors · 2026-04-07
> **Rebuttal to Review Part 5**
>
> References
>
> [1] T. Zhang, Z. Li, X. Yan, H. Qin, Y. Guo and Y. Zhang, "Quant‑dLLM: Post‑Training Extreme Low‑Bit Quantization for Diffusion Large Language Models".
>
> [2] T. Gloaguen, R. Staab, N. Jovanć and M. Vechev, "Watermarking Diffusion Language Models".
>
> [3] Z. Li, Z. Nie, Z. Zhou, Y. Liu, Y. Zhang, Y. Cheng, Q. Wen, K. Wang, Y. Guo and J. Zhang, "DiffuGuard: On the Robustness of Diffusion Language Models".
>
> [4] W. Jeung, S. Yoon, Y. Cho, D. Jeon, S. Shin, H. Hong and A. No, "A2D: Any‑Order, Any‑Step Safety Alignment for Diffusion Language Models".
>
> [5] S. Yamabe and J. Sakuma, "Toward Safer Diffusion Language Models: Discovery and Mitigation of Priming Vulnerability".
>
> [6] L. Wu, L. Zhong, W. Qu, Y. Li, Y. Liu, S. Zhai, C. Shen and J. Zhang, "DMark: Order‑Agnostic Watermarking for Diffusion Large Language Models".
>
> [7] A. Bagchi, A. Bhimaraju, M. Choraria, D. Alabi and L. Varshney, "Watermarking Discrete Diffusion Language Models."
>
> [8] S. Kumbhar, H. Liao, S. Appalaraju and K. Y. Singh, "Towards GUI Agents: Vision‑Language Diffusion Models for GUI Grounding".
>
> [9] W. Song, J. Chen, S. Chen, J. Wang, P. Ding and H. Zhao, "Fast‑dVLA: Accelerating Discrete Diffusion VLA to Real‑Time Performance".
>
> [10] Ş. T. Kocabay and T. R. Akkuş, "Diffutron: A Masked Diffusion Language Model for Turkish Language".
>
> [11] A. Ghiglino, D. Elenius, A. Roy, R. Kaur, M. Acharya and C. Samplawski, "Do Diffusion Models Dream of Electric Planes? Discrete and Continuous Simulation‑Based Inference for Aircraft Design".
>
> [12] Z. Yang, H. Liu, C. Cao and B. Su, "D3LM: A Discrete DNA Diffusion Language Model for Bidirectional DNA Understanding and Generation".
>
> [13] J. Zhang, M. Gagvani, C. Cui, J. Peng, R. Zhang and Z. Wang, "Efficient and Explainable End‑to‑End Autonomous Driving via Masked Vision‑Language‑Action Diffusion".
>
> [14] H. Jung, R. Hormazabal, J. Jo, Y. Park, K. Roh and S.‑Y. Yun, "MolHIT: Advancing Molecular‑Graph Generation with Hierarchical Discrete Diffusion Models".
>
> [15] N. Zhang, Z. Li, K. W. Loh, M. Xu, Q. Wang and Z. Wen, "DiMo: Discrete Diffusion Modeling for Motion Generation and Understanding".
>
> [16] E. Choi, H. Kim, H. Bang, T. Kwon and J. Nam, "D3PIA: A Discrete Denoising Diffusion Model for Piano Accompaniment Generation from Lead Sheet".
>
> [17] J. Zhou, X. Cheng, S. Zhao, Y. Jia, C. Liu and K. Zeng, "DIFFA‑2: A Practical Diffusion Large Language Model for General Audio Understanding".
>
> [18] R. Zuo, R. A. Potamias, Q. Sun, E. Ververas, J. Deng and S. Zafeiriou, "MaDiS: Taming Masked Diffusion Language Models for Sign Language Generation".
>
> [19] F. Fu, T. Guo and Z. Liu, "Learnable Sampler Distillation for Discrete Diffusion Models".
>
> [20] H. Zheng, X. Liu, C. Kong, N. Jiang, Z. Hu, W. Luo, W. Deng and G. Lin, "Ultra‑Fast Language Generation via Discrete Diffusion Divergence Instruct".
>
> [21] Y. Gao and Y. Sun, "Distillation of Discrete Diffusion by Exact Conditional Distribution Matching".
>
> [22] Y. Liang, Z. Wang, H. Chen, X. Sun, J. Wu, X. Yu, J. Liu, E. Barsoum, Z. Liu and N. K. Jha, "CD4LM: Consistency Distillation and Adaptive Decoding for Diffusion Language Models".
>
> [23] S. Chen, J. Jiao, L. J. Ratliff and B. Zhu, "dUltra: Ultra‑Fast Diffusion Language Models via Reinforcement Learning".
>
> [24] T. Zhang, X. Zhang, L. Han, H. Shi, X. He, Z. Li, H. Wang, K. Xu, A. Srivastava, V. Pavlovic and D. N. Metaxas, "T3D: Few‑Step Diffusion Language Models via Trajectory Self‑Distillation with Direct Discriminative Optimization".
>
> [25] E. Hoogeboom, D. Ruhe, J. Heek, T. Mensink and T. Salimans, "Beyond Single Tokens: Distilling Discrete Diffusion Models via Discrete MMD".
>
> [26] D. Li, N. Gushchin, D. Abulkhanov, E. Moulines, I. Oseledets, M. Panov and A. Korotin, "IDLM: Inverse‑Distilled Diffusion Language Models".
>
> [27] L. Berrayana, A. Heakl, M. A. Sohail, T. Hofmann, S. Khan and W. Chen, "Planner and Executor: Collaboration between Discrete Diffusion and Autoregressive Models in Reasoning".
>
> [28] Y. Fu, L. Whalen, Z. Ye, X. Dong, S. Diao, J. Liu, C. Wu, H. Zhang, E. Xie, S. Han, M. Khadkevich, J. Kautz, Y. C. Lin and P. Molchanov, "Efficient‑DLM: From Autoregressive to Diffusion Language Models, and Beyond in Speed".
>
> [29] M. Arriola, A. Gokaslan, J. T. Chiu, Z. Yang, Z. Qi, J. Han, S. S. Sahoo and V. Kuleshov, "Block Diffusion: Interpolating Between Autoregressive and Diffusion Language Models".
>
> [30] D. Shariatian, A. Durmus, U. Simsekli and S. Peluchetti, "Latent‑Augmented Discrete Diffusion Models".
>
> [31] M. Jo, J. Yoon, J. Deschenaux, C. Gulcehre and S. Ahn, "Loopholing Discrete Diffusion: Deterministic Bypass of the Sampling Wall".
>
> [32] Y. Hu, , et al, "Residual Context Diffusion Language Models".
>
> [33] P. Pynadath, J. Shi and R. Zhang, "CANDI: Hybrid Discrete‑Continuous Diffusion Models".
>
> [34] M. Hersche, , et al, "Soft‑Masked Diffusion Language Models".
>
> [35] J. Wu, et al, "Time‑Annealed Perturbation Sampling: Diverse Generation for Diffusion Language Models".

---

> > ### Comment · Reviewer_kKT6 · 2026-04-07
> >
> > Thanks for your response. Would it also be possible to update the PDF at this stage to reflect the changes you've made?

---

> ### Author Response · Authors · 2026-04-09
> **PDF Update**
>
> Thanks for the prompt response. We have begun updating the PDF. As recommended by the TMLR guidelines, we initially planned to revise the PDF comprehensively after all three reviews had been submitted and to update all content at once. However, we will proceed with updating the PDF immediately based on the existing revision plan. After completing the update, we will submit an additional author comment to inform you of the changes.

---

> ### Author Response · Authors · 2026-04-28
> **PDF Update**
>
> We sincerely thank the reviewer again for the valuable feedback. We have revised and updated the manuscript accordingly. Should there be any remaining concerns or suggestions, we would be more than willing to address them thoroughly and to the best of our ability.

---

> > ### Comment · Reviewer_kKT6 · 2026-04-29
> >
> > Thanks for your revision. I don't have any other concerns at the moment.

---

### Review · Reviewer_FmsB · 2026-03-25

**Summary Of Contributions:**

This paper surveys discrete diffusion language models (dLLMs) and discrete diffusion multimodal language models (dMLLMs), with emphasis on masked/absorbing-state formulations, recent LLM-scale model families, training and inference techniques, quantization, safety, and downstream applications. The manuscript’s stated goal is to provide a systematic overview of mathematical foundations, modeling methods, representative models, and practical engineering considerations for discrete diffusion at LLM scale. A major strength is that it is not just a mathematical survey: it also attempts to synthesize the emerging systems stack around dLLMs, including initialization, decoding, caching, sparse computation, and multimodal extensions. At the same time, the paper currently overstates several claims, contains a number of technical imprecisions and editorial issues, and does not yet position itself sharply enough relative to other recent diffusion-language-model surveys.

**Audience:**

Yes

**Audience Explanation:**

I believe this work would be of interest to the community, due to the growing body of work on discrete diffusion models and especially their application to natural language applications. This work provides a good overview of various works and concepts involved, but some issues pointed out in my review limit the paper’s impact and usefulness.

**Broader Impact Concerns:**

The paper presents a a survey of existing discrete diffusion-based models for language and other applications. While generative AI systems can be used for harmful purposes, there are no specific concerns or ethical implications of this work that need to be highlighted.

**Claims And Evidence:**

No

**Claims Explanation:**

- **Positioning Relative to Other Surveys.** There are several similar survey papers on diffusion language models [1,2,3,4,5]. The manuscript must clearly differentiate itself from these existing works, in terms of scope, coverage of topics, any unifying insights, level of detail etc. Without such a discussion, it is hard to gauge the value added by this work.
- **Writing issues.** The manuscript needs substantial copy editing. There are many grammatical errors and awkward phrasings, especially in the early sections, for example: “These capabilities are previously difficult...”, “domains and etc..”, “D3PM framework support”, “an final loss”, “predictor-corrector steps is used”, “Standard discrete diffusion models relies”, and “posting-training.”
- **Insufficient discussion on limitations of diffusion LMs.** There is almost a complete absence of a discussion on the weaknesses of diffusion LMs, such as: high sensitivity to decoding heuristics, have awkward length control, have a less mature alignment/serving stack, and still have uncertain scaling behavior at frontier model sizes. The survey paper should provide an objective discussion of not only the strengths of the subject being discussed, but a well-rounded discussion on all aspects.
- **Several claims are too strong or misleading.** The paper overstates the contrast between diffusion and autoregressive models several times. Some specific points:
    - The claim that dLLMs have been “widely shown” to outperform AR models is too broad. Current evidence is mixed and highly dependent on scale, task, and specific training setup of the models.
    - The claim that proprietary models (Gemini Diffusion, Mercury) beat “their AR counterpart” while achieving large speedups is misleading. The cited proprietary results compare with AR models trained on different data and setups (often they compare with older models, for eg Gemini Diffusion was released after Gemini 2.5 but the comparison was made with Gemini 2.0 flash lite).
    - The “dynamic perception” claim is overstated. Autoregressive models also repeatedly condition on the full generated history at every step, so describing AR generation as “one-pass static perception” is misleading.
    - The discussion of control over length, format, and reasoning structure overstates diffusion’s advantage. These are not unique diffusion capabilities, since AR LLMs can also achieve them through prompting and instruction following.
    - Tables 1 and 2 contain mix results from models with training settings, and do not normalize for training corpus, tokenizer, decoding budget, post-training recipe. Because the numbers are “collected from the original papers,” the current presentation risks encouraging apples-to-oranges comparison without enough caveats.
- **Minor Issues**
    - The introduction’s section roadmap lists Sec. 8 before Sec. 7.
    - The CTMC section contains a literal placeholder: “tau-leaping algorithm [reference]”.
    - There are many grammatical and typographical issues, e.g. “domains and etc..”, “D3PM framework support”, “an final loss”, “predictor-corrector steps is used”, “posting-training”, and “Standard discrete diffusion models relies”.
    - The notation and naming are inconsistent in places, e.g. LLaDA1.5 vs LLaDA 1.5, Dream_Instruct, raw benchmark header names like MMBench_en_test.
    - Figure 8’s publication-trend plot is based on keyword search over “All fields” metadata, which is a weak bibliometric proxy and may not add much scientific value in its current form.

*[1] Li, Tianyi, et al. "A survey on diffusion language models." arXiv preprint arXiv:2508.10875 (2025).*

*[2] Yi, Qiuhua, et al. "Diffusion models in text generation: a survey." PeerJ Computer Science 10 (2024): e1905.*

*[3] Zhang, Lingzhe, et al. "A survey on parallel text generation: From parallel decoding to diffusion language models." arXiv preprint arXiv:2508.08712 (2025).*

*[4] de Groot, Lars, Ruurd Jan Anthonius Kuiper, and Ayoub Bagheri. "A Survey
 of Discrete Diffusion for Text and Genomic Sequence Generation." The 37th Benelux Conference on Artificial Intelligence and the 34th Belgian Dutch Conference on Machine Learning.*

*[5] Tseng, Chiung-Yi, et al. "Diffusion-based Large Language Models Survey." Authorea Preprints (2025).*

**Requested Changes:**

- The positioning of this work with respect to other similar survey papers on diffusion LMs should be explicitly clarified to clarify what new value is added in this work.
- Some claims with respect to the advantages of diffusion LMs and their relative performance to AR LLMs should be corrected and toned down, or provided with citations that provide fair comparisons.
- For a complete presentation on diffusion LMs, the limitations and weaknesses of these method particularly with respect to AR counterparts should also be highlighted and discussed in detail. Without this discussion, the paper provides an incomplete understanding.
- Please fix some writing issues, grammatical errors, notation inconsistencies pointed out in the above fields.

---

> ### Author Response · Authors · 2026-04-07
> **Rebuttal to Review Part 1**
>
> ### Positioning Relative to Other Surveys
>
> We appreciate the reviewer’s request for a clearer positioning of our survey relative to existing reviews on diffusion language models. We will add additional discussion to explicitly clarify this aspect.
>
> Earlier surveys, such as Li et al. (2025) [1], Yi et al. (2024) [2], Zhang et al. (2025) [3], de Groot et al. (2025) [4] and Tseng et al. (2025) [5], provide overviews of diffusion-based generative models. However, several of these works rely on relatively outdated literature and therefore do not reflect the most recent developments in the field.
>
> For instance, Yi et al. (2024) [2] divide text generation into conditional, unconstrained, and multi-mode categories, and their literature coverage primarily extends up to **late 2023**. De Groot et al. (2025) [4] survey discrete diffusion models across natural language and genomic sequences, focusing on foundational formulations and adaptations of pretrained models. Their chronological overview includes only seven dLLM papers up to **mid-2025**, and does not cover billion-scale diffusion language models or unified language–vision systems. Tseng et al. (2025) [5] propose a taxonomy of diffusion-based large language models based on sampling strategies, guidance types, noise schedules, and temporal conditioning. While informative, their survey covers work only up to **early 2025**, and includes only very limited discussion of subsequent developments (e.g., a single 2025 work).
>
> As a result, these surveys primarily cover early-stage diffusion models in either continuous or discrete settings, and do not capture the rapid evolution of masked-based diffusion language models (dLLMs) and multimodal diffusion systems emerging after mid-2025. Notably, during this period, dLLMs have scaled to the **billion-parameter regime**, and both training and inference paradigms have undergone substantial changes compared to earlier small-scale models.
>
> Our survey differs substantively from these works in both scope and organisation. We provide a comprehensive overview of discrete diffusion language and multimodal models through early 2026, cataloguing proprietary and open-source models exceeding one billion parameters—including LLaDA 1.5, Dream and DreamOn, Seed Diffusion, Fudoki, MMaDA, Muddit, and Mercury—and covering multimodal systems such as Dimple, LaViDa, and LLaDA-V. In contrast to earlier surveys that predate large-scale dLLMs, we synthesise recent industrial systems and emerging research architectures, enabling readers to understand the current state of the field. We also provide a rigorous mathematical exposition of discrete diffusion, tracing the development of masked and continuous-time formulations, and systematically analysing the evolution of forward and reverse processes across successive model generations.
>
> Our organisation provides finer granularity than the high-level taxonomies in Li et al. (2025) [1]. Li et al. categorise training strategies into pre-training and post-training, and group inference techniques into parallel decoding, unmasking/remasking, guidance, and efficiency improvements. While these categories capture broad trends, they do not fully reflect the detailed design space. In contrast, we extend this taxonomy by explicitly distinguishing initialization strategies, complementary masking schedules, importance reweighting mechanisms, loss formulations, and reinforcement learning adaptations. For inference, we further cover adaptive sampling, heuristic and planner-based unmasking strategies, step-reduction techniques, KV-cache optimisation, dynamic length control, and hybrid autoregressive–diffusion decoding. This fine-grained organisation enables a more precise understanding of methodological differences and open research challenges.
>
> Finally, Zhang et al. (2025) [3] survey parallel text generation and categorise methods into autoregressive (AR) and non-autoregressive paradigms, with diffusion models appearing only as a subsection (Section 4.2) within the non-AR category. Their primary focus lies on AR-based acceleration techniques such as speculative decoding and edit-based refinement, while diffusion-based approaches receive comparatively limited coverage. In contrast, our survey is fully dedicated to diffusion language models. We provide comprehensive discussions on training, inference, multimodal extensions, safety, and quantization, and construct a detailed timeline tracking the evolution from sub-billion to multi-billion parameter diffusion models. This holistic and up-to-date coverage, combined with in-depth theoretical analysis, positions our survey as a complementary and substantially extended reference to prior work.

---

> ### Author Response · Authors · 2026-04-07
> **Rebuttal to Review Part 2**
>
> ### Writing Issues
>
> We appreciate the reviewer’s suggestions. We will comprehensively revise the paper. Our revision plan includes manual copy-editing by all authors and a subsequent proof-read using professional tools such as Grammarly. Our revisions will cover the following aspects. First, incorrect or inconsistent articles and plurality will be corrected; for example, expressions such as "an final loss" will be changed to "a final loss," and "predictor-corrector steps is used" will become "predictor-corrector steps are used." Second, subject–verb agreement errors will be fixed; for example, "D3PM framework support various types of transition matrices" will be changed to "supports various types of transition matrices." Third, awkward phrasing and redundancy such as "domains and etc.." will be rewritten to "across language, vision–language and biological domains." Fourth, typographical errors and misspellings such as "posting-training" will be corrected to "post-training," and placeholders like "tau-leaping algorithm [reference]" will be replaced with the appropriate citation. Fifth, we will standardise notation and naming, ensuring consistency between notations, such as "LLaDA 1.5" and "LLaDA1.5", removing underscores in names like "Dream Instruct," and normalising benchmark names. The revised manuscript will incorporate these corrections and undergo a final grammar check.
>
> ---
>
> ### Discussion of Limitations of Diffusion LMs
>
> We agree with the reviewer that a balanced survey should discuss the limitations of discrete diffusion language models relative to autoregressive LLMs. We will add a subsection to the introduction section summarising the main weaknesses of discrete diffusion models, supported by recent literature. The following content will be added:
>
> One limitation is computational inefficiency. Masked diffusion models recompute attention and feed-forward layers for every masked token at each denoising step, even though many masked tokens cannot be effectively decoded at that step and thus do not contribute to meaningful progress. This inefficiency is further exacerbated in block diffusion settings, where the model is initialized with a large number of masked tokens but is only able to resolve a subset of them in each iteration, leading to redundant computation across steps.
>
> A second limitation is that standard discrete diffusion training is costly compared with autoregressive training. Efficient-DLM notes that existing diffusion LMs either fail to outperform AR models in speed or are limited to small scale because of high training costs [6], and a controlled comparison finds that masked diffusion requires more steps to converge despite similar per-step throughput [7].
>
> A third limitation is that diffusion LMs are sensitive to decoding heuristics. Masked diffusion models unmask tokens based on heuristics such as confidence or margin, and early decisions can propagate errors. The Lookahead Unmasking study shows that the performance of masked diffusion is highly dependent on the unmasking order and that decoding heuristics allow errors to cascade [8].
>
> A fourth limitation is that discrete diffusion requires the output length to be predetermined at sampling time. DAEDAL demonstrates that choosing a short sequence length restricts reasoning and reduces performance, whereas choosing a long length causes wasted computation and can degrade quality; diffusion LMs lack the dynamic length adaptation available to AR models [9].
>
> A fifth limitation is that training and serving stacks remain immature. Standard reinforcement learning policies such as GRPO and VRPO, which successfully align autoregressive models, cause instability and reward collapse when applied to diffusion LMs because sequence probabilities are intractable; StableDRL proposes an alternative but highlights that reinforcement learning for diffusion LMs is still fragile [10].
>
> A sixth limitation is that scaling behaviour at frontier model sizes remains uncertain. Prior work [11] shows that diffusion LLMs have scaling exponents comparable to AR models but suffer from large constant-factor gaps; they require more compute to match AR perplexity, and perplexity alone is insufficient for fair cross-paradigm comparison.
>
> We will integrate this discussion into the revised introduction to provide a balanced view of both strengths and weaknesses.

---

> ### Author Response · Authors · 2026-04-07
> **Rebuttal to Review Part 3**
>
> ### Clarification and Moderation of Claims
>
> We appreciate the reviewer’s feedback that several statements in the current draft overstate the advantages of diffusion LLMs. In the revision we will moderate these claims and provide nuance.
>
> First, we will clarify that evidence comparing diffusion and autoregressive models remains mixed. While certain diffusion LLMs report promising results—for instance, LLaDA 8B matches LLaMA3 on downstream benchmarks according to [5] and Efficient-DLM converts pretrained AR models into diffusion models with improved throughput and comparable accuracy [6]—overall performance depends strongly on model scale, domain and training regimen. A controlled comparison on TinyStories shows that masked diffusion has nearly identical per-step training throughput to AR models but requires more training steps to converge [7]. Thus diffusion is a promising alternative rather than a universally superior paradigm. We will revise the statement that diffusion LLMs have been "widely shown" to outperform AR models to emphasise that diffusion LLMs match or modestly exceed AR models only in certain tasks and scales, and that more controlled comparisons are needed.
>
> Second, we will revise the description of proprietary models. Our original draft claimed that Mercury and Gemini Diffusion "beat their AR counterparts" while providing large speed-ups. As the reviewer notes, available reports compare diffusion models with earlier AR baselines using different training data. We will instead report the published results without qualitative judgement: Gemini Diffusion, released in May 2025, achieves a latency advantage but trails the February 2025 Gemini 2.5 AR model by roughly three percent on average; Mercury Coder 2, released in February 2026, shows an average two percent improvement over the August 2025 GPT-5 mini baseline on code tasks.
>
> Third, regarding "dynamic perception", our wording was ambiguous. We will avoid suggesting that AR models have "one-pass static perception". Autoregressive transformers compute the key–value (KV) representations of tokens at each generation step, but once generated, these representations remain fixed. Diffusion LMs, by contrast, employ bidirectional attention, allowing the representations of tokens to be updated conditioned on both left and right context. We will rewrite this description to clarify that diffusion models enable iterative refinement of token representations across all positions, whereas AR models update only newly generated tokens.
>
> Fourth, we will modify statements about control over length, format and reasoning structure. Prompting and instruction-following in autoregressive LLMs already allow users to specify desired lengths and formats. Diffusion models offer an alternative mechanism via mask scheduling and guidance, and may provide finer control when combined with template conditioning, but these capabilities are not unique. We will reflect this nuance in the revised text.
>
> Fifth, we will add explicit clarification to the tables. Tables summarising model performance will be prefaced with a disclaimer stating that results are collected from original papers with differing training setups, corpora, tokenizers, decoding budgets and post-training recipes, and therefore cannot be used for direct apples-to-apples comparisons. We will further emphasise that these results provide indicative evidence of current progress rather than definitive conclusions.
>
> ---
>
> ### Minor Issues
>
> We will address all minor issues raised by the reviewer. The section roadmap currently lists Section 8 before Section 7; we will correct the ordering. The placeholder "tau‑leaping algorithm [reference]" will be replaced with the appropriate citation to the tau‑leaping method in continuous‑time Markov chain simulation. All grammatical and typographical errors, including duplicated periods and inconsistent punctuation, will be corrected as noted above. Notation for models and benchmarks will be standardised, resolving discrepancies such as "LLaDA1.5" versus "LLaDA 1.5" and normalising names like "MMBench en test." We will redraw Figure 8 using a more rigorous bibliometric methodology: we will restrict the query to titles and abstracts within the computer science category on arXiv, deduplicate papers with identical author lists and similar abstracts, and present the resulting trend for publications on discrete diffusion LMs. The updated figure will appear in the revised PDF.

---

> ### Author Response · Authors · 2026-04-07
> **Rebuttal to Review Part 4**
>
> References
>
> [1] Tianyi Li, Mingda Chen, Bowei Guo, and Zhiqiang Shen. A survey on diffusion language models.
>
> [2] Qiuhua Yi, Xiangfan Chen, Chenwei Zhang, Zehai Zhou, Linan Zhu, and Xiangjie Kong. Diffusion models in text generation: a survey.
>
> [3] Lingzhe Zhang, Liancheng Fang, Chiming Duan, Minghua He, Leyi Pan, Pei Xiao, Shiyu Huang, Yunpeng Zhai, Xuming Hu, Philip S. Yu, and Aiwei Liu. A survey on parallel text generation: from parallel decoding to diffusion language models.
>
> [4] Lars de Groot, Ruurd Jan Anthonius Kuiper, and Ayoub Bagheri. A survey of discrete diffusion for text and genomic sequence generation.
>
> [5] Chiung‑Yi Tseng, Ching‑Min Chen, Kuan‑Heng Liu, Yen‑Chang Hsiao, and Hung‑Yi Lee. Diffusion‑based large language models survey.
>
> [6] Yonggan Fu, Lexington Whalen, Zhifan Ye, Xin Dong, Shizhe Diao, Jingyu Liu, Chengyue Wu, Hao Zhang, Enze Xie, Song Han, Maksim Khadkevich, Jan Kautz, Yingyan Celine Lin, and Pavlo Molchanov. Efficient‑DLM: from autoregressive to diffusion language models, and beyond in speed.
>
> [7] Caio Vicentino. Autoregressive vs. masked diffusion language models: a controlled comparison.
>
> [8] Minseo Kim, Chenfeng Xu, Coleman Richard Charles Hooper, Harman Singh, Ben Athiwaratkun, Ce Zhang, Kurt Keutzer, and Amir Gholami. Consistency diffusion language models: up to 14× faster inference without sacrificing quality.
>
> [9] Sanghyun Lee, Seungryong Kim, Jongho Park, and Dongmin Park. Lookahead unmasking elicits accurate decoding in diffusion language models.
>
> [10] Jinsong Li, Xiaoyi Dong, Yuhang Zang, Yuhang Cao, Jiaqi Wang, and Dahua Lin. Beyond fixed: training‑free variable‑length denoising for diffusion large language models.
>
> [11] Jianyuan Zhong, Kaibo Wang, Ding Ding, Zijin Feng, Haoli Bai, Yang Xiang, Jiacheng Sun, and Qiang Xu. Stabilizing reinforcement learning for diffusion language models.
>
> [12] Subham Sekhar Sahoo, Jean‑Marie Lemercier, Zhihan Yang, Justin Deschenaux, Jingyu Liu, John Thickstun, and Ante Jukic. Scaling beyond masked diffusion language models.
>
> [13] Kaiwen Zheng, Yongxin Chen, Hanzi Mao, Ming‑Yu Liu, Jun Zhu, and Qinsheng Zhang. Masked diffusion models are secretly time‑agnostic masked models and exploit inaccurate categorical sampling.

---

### Review · Reviewer_RBSM · 2026-05-05

**Summary Of Contributions:**

This paper surveys Discrete Diffusion Large Language Models (dLLMs) and their multimodal variants (dMLLMs), with coverage stated to extend through early 2026. Its main contributions are a multi-level taxonomy of training and inference techniques and a current catalogue of representative models including diffusion method for language, multi-modal, and unified model. The paper also covers various applications as well as a discussion of limitations and future directions.

**Strengths**
- Mathematical exposition in Sec. 2 is correct.
- The inference taxonomy is finer-grained than prior surveys.
- Appendix F provides a substantive critical synthesis of limitations and future directions.

**Weaknesses**:
- Sec. 4.3 and 4.4 describes dMLLMs and unified models without explaining how they actually work.
- The differentiation argument against Li et al. (2025f) has gaps on both sides: the paper misses several billion-parameter dLLMs that Li et al. (2025f) already covers, and Sec 5.7 catalogues fewer RL methods than Li et al. does.

**Audience:**

Yes

**Audience Explanation:**

Yes. dLLMs and dMLLMs are an active area, and the field would benefit from a current-state survey of this scope.

**Broader Impact Concerns:**

I have no significant broader-impact concerns since this is a survey paper.

**Claims And Evidence:**

No

**Claims Explanation:**

Most claims are well-supported. Two specific claims requires revision.
- Sec. 1.1 frame this survey as more fine-grained and more comprehensive. However, several billion-parameter or otherwise notable dLLMs covered by Li et al. (2025f, Aug 2025) are absent here: LLaDA-MoE[1], UltraLLaDA[2], D-DiT[3].
- The paper claims "formalize the underlying mathematical frameworks of dLLMs and dMLLMs". But the authors do not explain how multimodal dLLMs work.
	- Sec. 4.3 describes three dMLLMs without saying whether images are tokenized, whether image tokens are masked, or how continuous image features integrate with the discrete diffusion process.
	- Sec. 4.4 describes unified models with VQ-VAE-tokenized images jointly denoised with text, but never introduces formal framework.

[1] : Zhu, Fengqi, et al. "Llada-moe: A sparse moe diffusion language model." _arXiv preprint arXiv:2509.24389_ (2025).

[2] : He, Guangxin, et al. "Ultrallada: Scaling the context length to 128k for diffusion large language models." _arXiv preprint arXiv:2510.10481_ (2025).

[3] : Li, Zijie, et al. "Dual diffusion for unified image generation and understanding." _Proceedings of the Computer Vision and Pattern Recognition Conference_. 2025.

**Requested Changes:**

1. Calibrate the Sec. 1.1 differentiation argument against Li et al. (2025f).
	- The paper misses several billion-parameter dLLMs that Li et al. already covers (see weaknesses).
	- Sec 5.7 also misses several RL methods catalogued in Li et al.'s Table 2: SEPO, IGPO, etc.
2. Explain how dMLLMs work, formalize the mathematical framework, and distinguish them from unified models.
3. Eq. (41) gives the entropy, not the "negative entropy" as labeled.
4. Add a brief notation-conventions note flagging the two time-direction conventions used in the paper: Sec. 2.2 (0 = clean, T = noise) versus Sec. 2.5 and Sec. 3 (0=source, 1 = target).
5. Restructure Sec. 3 to be problem-centric: for each technique, explain why it is representative and which problem it addresses.

---

> ### Author Response · Authors · 2026-05-18
> **Rebuttal to Review Part 1**
>
> ### Inclusion of LLaDA-MoE, UltraLLaDA, and DDiT.
>
> We appreciate the reviewer’s suggestion. We will add descriptions of LLaDA‑MoE, UltraLLaDA and D‑DiT to make the review more comprehensive. Below are the paragraphs that we plan to include in the revised Section 4.2 and 4.3.
>
> LLaDA‑MoE[2] introduces a masked diffusion language model with a sparse Mixture‑of‑Experts (MoE) architecture.  The model maintains a total capacity of 7 billion parameters but activates only 1.4 billion parameters during inference.  Its training pipeline comprises two large pretraining stages on a mixed text corpus (10 trillion tokens each), followed by two annealing stages that refine the model on 500 billion tokens each and increase the rotary positional embedding base and context length from 4 k to 8 k tokens; finally, supervised fine‑tuning on high‑quality prompt–answer pairs produces the instruct model.  This sparse MoE design allows LLaDA‑MoE to surpass dense 8 billion‑parameter diffusion models such as LLaDA and Dream across knowledge, reasoning, mathematics and coding benchmarks while using far fewer active parameters.
>
> UltraLLaDA[3] studies how to extend the context window of diffusion LLMs without retraining from scratch.  The authors introduce a “diffusion‑aware NTK” extension of rotary positional embeddings and evaluate different long‑context masking strategies.  Post‑training the LLaDA‑8B base model with these modifications produces UltraLLaDA, a diffusion LLM capable of handling up to 128 k tokens.  The paper reports that this model significantly outperforms training‑free baselines on long‑context benchmarks, demonstrating stable perplexity and high task accuracy across extended context lengths.  In addition to the modified embeddings, the authors investigate adaptive attention masking and end‑of‑document concatenation to reduce cross‑document interference.  Their experiments show that UltraLLaDA consistently outperforms LongLLaDA and the original LLaDA base model on long‑context retrieval and language modeling tasks, highlighting the effectiveness of lightweight post‑training for context extension.
>
> Dual diffusion Transformer (D-DiT)[4] for unified image generation and understanding introduces a dual‑branch diffusion model that unifies continuous image diffusion and discrete masked text diffusion under a single transformer. The model employs a cross‑modal maximum‑likelihood framework that jointly trains the conditional likelihoods of images and text with a single loss, enabling end‑to‑end tasks such as text‑to‑image generation, image captioning and visual question answering. Experiments report improvements in image quality and multimodal tasks compared with baseline models, while noting that performance can degrade on complex scenes and that the model still requires a preset sequence length.

---

> ### Author Response · Authors · 2026-05-18
> **Rebuttal to Review Part 2**
>
> ### Inclusion of More RL Techniques
>
> We thank the reviewer for pointing out this important omission. Section 5.7 currently discusses diffusion-specific reinforcement learning methods such as diffu‑GRPO, Coupled‑GRPO and VRPO, but does not cover several new algorithms identified by [1]. We will expand this section to explain these methods and make clear which techniques fall under policy-gradient optimization versus preference‑optimization. Below are the descriptions we propose to add.
>
> Score Entropy Policy Optimization (SEPO)[5] is a policy‑gradient algorithm designed specifically for discrete diffusion models. It introduces a clipped‑ratio loss and self‑normalized importance sampling to handle non‑differentiable rewards while maintaining low variance, enabling fine‑tuning of diffusion LLMs for conditional or unconditional generation tasks with improved performance.
>
> Inpainting‑Guided Policy Optimization (IGPO)[6] addresses the unstable gradients encountered when training diffusion LLMs with reinforcement learning. It guides exploration by inserting partial ground‑truth reasoning traces during sampling and uses entropy‑based filtering to preserve self‑generated traces, thereby restoring meaningful gradients and substantially improving performance on math reasoning benchmarks.
>
> Sandwiched policy gradient (SPG)[7] for masked diffusion language models tackles the difficulty of applying reinforcement learning to diffusion LLMs by jointly optimizing tractable lower and upper bounds on the intractable log‑likelihood. SPG maximizes a lower bound on the likelihood for high‑reward sequences and minimizes an upper bound for low‑reward sequences, forming a ``sandwiched'' objective that yields more robust policy gradients. A block‑wise masking strategy is used to estimate these bounds via Monte Carlo, reducing gradient variance and improving optimization stability.
>
> Step‑Aware Policy Optimization (SAPO)[9] introduces a process‑based reward that encourages incremental progress through the denoising steps. By aligning the agent’s reward with a latent reasoning hierarchy, SAPO guides the model to learn structured reasoning paths and demonstrates strong performance on reasoning benchmarks.
>
> Boundary‑Guided Policy Optimization (BGPO)[9] tackles the high memory overhead of reinforcement learning in diffusion models. It maximizes a linear lower bound on the evidence lower bound objective, which is equivalent to the original objective for on‑policy training. This allows gradient accumulation over large Monte Carlo sample sizes while maintaining constant memory usage, enabling more accurate likelihood approximation and leading to improved results on math, code and planning tasks.
>
> We will revise Section 5.7 to incorporate these methods, distinguishing between policy‑gradient and preference‑optimization approaches. By adding the above descriptions, the survey will better reflect the current landscape of reinforcement learning techniques for discrete diffusion language models.

---

> ### Author Response · Authors · 2026-05-18
> **Rebuttal to Review Part 3**
>
> ### Formal Framework of dMLLM and Unified Model, and the Differences Between dMLLMs and Unified Models
>
> We thank the reviewer for these thoughtful comments. As suggested, we will extend Section 1 by adding one subsection that rigorously formalize discrete diffusion multimodal language models (dMLLMs) and unified models, explicitly distinguishing between these two paradigms. Furthermore, we will also clearly address key architectural details, including whether images are tokenized, whether image tokens are masked, and how continuous image features integrate with the discrete diffusion process. The following content is planned to be added.
>
> In the new subsection on dMLLMs, we will explain that these models extend dLLMs by conditioning on a continuous vision representation. Let $x=(x_1,\ldots,x_L)$ denote a sequence of discrete tokens (words) and let $v$ denote the image input. A vision encoder $E_{\mathcal{V}}$ produces continuous features $h_v=E_{\mathcal{V}}(v)$. The forward corruption process of a dMLLM is the same continuous-time Markov chain used in dLLMs: at time $t\in [0,T]$, each discrete token $x_i$ is replaced by a mask token with rate $\lambda(t)$; the image features $h_v$ remain fixed because the image is not discretized. The reverse process learns a conditional score function $s_\theta(\cdot)$ that predicts the distribution of the original tokens given their corrupted version and the vision features: $q_\theta(x,|,\tilde{x}_t,h_v)$. The training objective is the same negative evidence lower bound used in dLLMs, except that the model must marginalize over the corruption process conditional on $h_v$. Because the vision encoder output is continuous and fixed across all denoising steps, image tokens are not masked and are not sampled; only the discrete tokens are corrupted and denoised. The training of discrete diffusion multimodal language models (dMLLMs) generally follows a multi‑phase strategy that combines a pre‑trained visual backbone, vision‑language alignment, and post‑training. First, a strong vision encoder is adopted—often pre‑trained on large-scale image datasets—to produce continuous visual embeddings. These embeddings remain fixed throughout the diffusion process and serve as context for language generation. Second, an alignment phase is performed to connect visual and linguistic modalities. Third, post‑training refines the model using supervised fine‑tuning (SFT) on instruction‑following datasets and, when applicable, reinforcement learning (RL) based preference optimization.
>
> In the new subsection on unified models, we will formalize the approach taken by models such as Muddit, MMaDA, FUDOKI and D-DiT. Unified models seek to denoise text and images with a single discrete diffusion or flow process. Images are quantized into discrete tokens via a VQ-VAE or VQ-GAN; each token $y_i$ represents either a word from the text vocabulary or a code from the image codebook. The forward corruption is a continuous-time Markov chain that independently replaces each token by a mask token at rate $\lambda(t)$, regardless of modality. The reverse process is a masked token predictor that infers the original discrete tokens for both text and image modalities: $q_\theta(y,|,\tilde{y}*t)$. Because both modalities are discrete, there is no continuous vision encoder; all tokens, including image codes, are masked and denoised. For example, Muddit formulates a unified CTMC over a finite alphabet and defines the training objective as a continuous‑time negative ELBO shared across modalities. MMaDA generalizes this by combining a discrete diffusion loss $L_{\text{diff-disc}}$ for masked token prediction with a continuous diffusion loss $L_{\text{diff-cont}}$ for continuous latent variables (when included), and FUDOKI extends the framework using discrete flow matching instead of diffusion, introducing metric‑induced probability paths and kinetic optimal velocities.
> We will then contrast dMLLMs and unified models. In dMLLMs, images are encoded into continuous embeddings by a vision encoder and are not discretized or masked; only the text tokens participate in the discrete diffusion process. The model conditions on image features through cross‑attention and requires a predetermined output length. In unified models, images are quantized into discrete tokens via VQ-VAE or VQ-GAN, and the same diffusion or flow process corrupts and denoises both text and image tokens; there is no continuous vision encoder during sampling. Unified models can therefore generate images and text within a single transformer by masking and denoising a mixed sequence of discrete codes. These distinctions will be clearly articulated in the revised manuscript to address the reviewer’s questions about tokenization, masking and integration of continuous image features.

---

> ### Author Response · Authors · 2026-05-18
> **Rebuttal to Review Part 4**
>
> ### Restructuring of Sec. 3
>
> Thanks for the suggestion. We will revise Sec. 3 by organizing it in a more problem-centric manner. We also note that the techniques introduced in Sec.3 cannot be perfectly categorized by a one-to-one mapping between ``method'' and ``problem''. A single technique may address multiple limitations, and different techniques may provide overlapping solutions to the same issue. For example, both flexible-length masked diffusion and optimal transport position coupling address the fixed-length limitation, but the latter further considers position adjustment. Similarly, block diffusion primarily improves computational efficiency, but it also partially relaxes fixed-length generation Therefore, rather than strictly reorganizing Sec. 3 into disjoint problem categories, we will revise each subsection to begin with a clear statement of the problem addressed by the technique. This revision will make Sec. 3 more problem-centric. The following content will be added.
> Below is the proposed text to be added at the beginning of each subsection in Sec. 3.
>
> **Sec. 3.1 Block Diffusion Models**
>
> Block diffusion models address the computational inefficiency of applying diffusion-based denoising over the entire sequence with full bidirectional attention. In standard discrete diffusion language models, all tokens are jointly refined, which leads to expensive global attention computation and makes standard KV-cache reuse difficult. Block diffusion decomposes the sequence into multiple blocks: tokens within each block are generated through diffusion-based denoising, while previous blocks serve as autoregressive context. This design improves inference efficiency by reducing the effective denoising region and also partially relaxes the fixed-length limition in generation, since generation can proceed block by block rather than requiring the whole output sequence to be specified in advance.
>
> **Sec. 3.2 Flexible-Length Masked Diffusion**
>
> Flexible-length masked diffusion is designed to overcome the fixed-length generation constraint of standard masked diffusion models. Conventional masked diffusion typically initializes generation with a sequence of mask tokens of predetermined length, implying that the target output length must be specified before decoding. This assumption limits its applicability to open-ended generation, editing, and insertion-based generation. Flexible-length masked diffusion introduces additional sequence-editing states or operations, such as insertion and deletion, so that the model can dynamically expand or shrink the generation canvas during denoising. As a result, the model can generate variable-length outputs without relying on a fixed preset sequence length.
>
> **Sec. 3.3 Partial Masking**
>
> Partial masking addresses the inefficient use of intermediate denoising states in standard discrete diffusion. In conventional masked diffusion, a token may remain masked for several denoising steps, and intermediate model predictions before the final unmasking step are not explicitly preserved in the generated sequence. This creates a mismatch between the multi-step computation used during decoding and the discrete token-level update that only occurs when a token is finally unmasked. Partial masking mitigates this issue by decomposing each original token $x_i$ into multiple sub-tokens $(y_{i,1}, \ldots, y_{i,\ell})$. Instead of decoding the whole token in a single update, the model can progressively reveal sub-token information across multiple denoising steps. This enables a finer-grained denoising trajectory and allows intermediate computation to contribute more directly to token reconstruction.
>
> **Sec. 3.4 Diffusion with Optimal Transport Position Coupling**
>
> Diffusion with optimal transport position coupling addresses the lack of positional flexibility in standard discrete diffusion models. Conventional discrete diffusion assumes a fixed sequence canvas, where token positions are predetermined throughout the denoising process. This assumption is restrictive for tasks such as infilling, continuation, and text editing, where newly generated tokens may require position reallocation and existing position IDs may need to be adjusted after insertion or restructuring. This method jointly denoises token values and token positions by coupling discrete token generation with optimal-transport-based position refinement. Formally, generation is modeled as the joint evolution of content variables $x_t$ and position variables $z_t$, where token denoising updates $x_t$ and position refinement updates $z_t$. This joint formulation allows the model to modify both what tokens are generated and where they should be placed, thereby supporting more flexible infilling and variable-position generation.

---

> ### Author Response · Authors · 2026-05-18
> **Rebuttal to Review Part 5**
>
> ### Error in Eq. 41
>
> Thank you for pointing this out. We will correct Eq. 41 to ``negative entropy'' to ensure it is aligned with the corresponding description.
>
> ### Notation-Conventions Note in Sec. 2.2, Sec. 2.5 and Sec. 3
>
> We appreciate this valuable suggestion. We currently retained the notations in their original papers. To avoid any potential confusion, we will add a clarifying note regarding the notation in Sec. 2.2, Sec. 2.5, and Sec. 3 to explicitly define the directionality of time $t$ in each respective section.

---

> ### Author Response · Authors · 2026-05-18
> **Rebuttal to Review Part 6**
>
> ### Reference
>
> [1] Tianyi Li, Mingda Chen, Bowei Guo, and Zhiqiang Shen. A survey on diffusion language models.
>
> [2] Fengqi Zhu, Zebin You, Yipeng Xing, Zenan Huang, Lin Liu, Yihong Zhuang, Guoshan Lu, Kangyu Wang, Xudong Wang, Lanning Wei, Hongrui Guo, Jiaqi Hu, Wentao Ye, Tieyuan Chen, Chenchen Li, Chengfu Tang, Haibo Feng, Jun Hu, Jun Zhou, Xiaolu Zhang, Zhenzhong Lan, Junbo Zhao, Da Zheng, Chongxuan Li, Jianguo Li, and Ji‑Rong Wen. LLaDA‑MoE: A Sparse MoE Diffusion Language Model.
>
> [3] Guangxin He, Shen Nie, Fengqi Zhu, Yuankang Zhao, Tianyi Bai, Ran Yan, Jie Fu, Chongxuan Li, and Binhang Yuan. UltraLLaDA: Scaling the Context Length to 128K for Diffusion Large Language Models.
>
> [4] Zijie Li, Henry Li, Yichun Shi, Amir Barati Farimani, Yuval Kluger, Linjie Yang, and Peng Wang. Dual Diffusion for Unified Image Generation and Understanding.
>
> [5] Oussama Zekri and Nicolas Boullé. Fine‑Tuning Discrete Diffusion Models with Policy Gradient Methods.
>
> [6] Siyan Zhao, Mengchen Liu, Jing Huang, Miao Liu, Chenyu Wang, Bo Liu, Yuandong Tian, Guan Pang, Sean Bell, Aditya Grover, and Feiyu Chen. Inpainting‑Guided Policy Optimization for Diffusion Large Language Models.
>
> [7] Chenyu Wang, Paria Rashidinejad, DiJia Su, Song Jiang, Sid Wang, Siyan Zhao, Cai Zhou, Shannon Zejiang Shen, Feiyu Chen, Tommi Jaakkola, Yuandong Tian, and Bo Liu. SPG: Sandwiched policy gradient for masked diffusion language models
>
> [8] Shaoan Xie, Lingjing Kong, Xiangchen Song, Xinshuai Dong, Guangyi Chen, Eric P. Xing, and Kun Zhang. Step‑Aware Policy Optimization for Reasoning in Diffusion Large Language Models.
>
> [9] Nianyi Lin, Jiajie Zhang, Lei Hou, and Juanzi Li. Boundary‑Guided Policy Optimization for Memory‑efficient Reinforcement Learning of Diffusion Large Language Models.

---

### Review · Reviewer_tjFK · 2026-05-16

**Summary Of Contributions:**

This paper surveys discrete diffusion language and multimodal models, which generate text by repeatedly filling in or refining masked tokens instead of writing left to right. Its main contribution is organizing the field into clear sections on theory, model families, training methods, inference tricks, efficiency, safety, and applications. A major strength is that it covers a very broad and recent set of models and techniques, including large language, vision-language, and unified multimodal systems. Another strength is that it explains why diffusion models may be useful, especially for faster parallel generation and better control over outputs. Its main weakness is that it sometimes feels more like a long catalog than a deep comparison, and some claims or benchmark comparisons need more careful qualification.

**Audience:**

Yes

**Audience Explanation:**

I think some TMLR readers would likely be interested in this paper because discrete diffusion language models are an active and fast-growing alternative to autoregressive language models. The survey brings together recent work on theory, training, inference and applications, which would be useful for researchers trying to understand the current state of the field.

**Broader Impact Concerns:**

I do not have broader impact concern on this paper.

**Claims And Evidence:**

Yes

**Claims Explanation:**

The submission’s claims are generally supported by clear and relevant evidence.

1. The paper supports its main claim of being a broad survey by covering mathematical foundations, model architectures, training methods, inference techniques, efficiency, safety, applications, and future directions.
2. The paper provides examples of representative models across the field, including LLaDA, DREAM, DiffuCoder, Dimple, LaViDa, LLaDA-V, MMaDA, FUDOKI, and Muddit
3. The authors have included organizing figures and tables, such as the model timeline, taxonomy diagram, AR-vs-diffusion comparison, and benchmark tables, which make the provided evidence easier to understand and follow.
4. The authors are careful to note that benchmark results are not directly apples-to-apples because models differ in training data, decoding budgets, tokenizers, and post-training recipes.

**Requested Changes:**

1. It would be good to make the benchmark discussion more careful by clearly explaining that the reported numbers come from different papers and are not directly comparable.
2. It would be good to improve the discussion of limitations and open problems by making them more specific and clearly connected to future research directions.

---

> ### Author Response · Authors · 2026-05-17
> **Rebuttal to Review Part 1**
>
> ### More discussion on the benchmark results
>
> We thank the reviewer for this helpful suggestion. We agree that the benchmark discussion should be made more careful, since the reported results are collected from different papers and are affected by heterogeneous training data, initialization strategies, post-training recipes, and evaluation settings. We will add the following paragraphs in the revised manuscript.
>
> These benchmark numbers should be interpreted as indicative evidence of recent progress rather than as direct apples-to-apples comparisons.
>
> For LLaDA series, LLaDA 8B is trained from scratch as a diffusion language model on about 2.3 trillion tokens and then supervised fine-tuned on 4.5 million instruction-response pairs, without reinforcement learning. In contrast, LLaDA 1.5 keeps the same LLaDA backbone but adds a preference-optimization stage, training on 350K preference pairs with Variance-Reduced Preference Optimization (VRPO). Therefore, the improvement of LLaDA 1.5 over LLaDA should not be attributed only to the base diffusion architecture, but also to its additional preference-alignment procedure.
>
> Dream 7B differs from LLaDA in both initialization and data scale. Dream is initialized from the autoregressive Qwen2.5 model and is further trained as a diffusion language model using approximately 580 billion tokens, followed by supervised fine-tuning on 1.8 million instruction-response pairs. Thus, compared with LLaDA, Dream uses much less diffusion pretraining data but benefits from autoregressive weight initialization and context-adaptive noise scheduling. This makes its benchmark numbers informative, but not directly comparable to those of a diffusion model trained from scratch.
>
> For code generation models, DiffuCoder 7B is trained on 130 billion code tokens and further introduces coupled-GRPO, a diffusion-native reinforcement learning method designed to reduce the variance of likelihood estimates during policy optimization. Consequently, its code benchmark performance reflects not only the discrete diffusion generation paradigm but also domain-specific code training and additional RL-based optimization. This differs from general-purpose dLLMs such as LLaDA and Dream.
>
> For multimodal models, Dimple adopts a hybrid autoregressive-then-diffusion training strategy, and its training data are comparable to the LLaVA-NEXT setting, with roughly 0.6M image-text pretraining samples and about 1.0M visual instruction-following samples. LaViDa, by contrast, builds on diffusion language backbones such as LLaDA or Dream and uses a two-stage multimodal training process with approximately 558K image-text pairs for visual alignment and about 1M visual instruction-following examples. LLaDA-V is built on LLaDA-8B and follows a multi-stage visual instruction tuning pipeline with datasets such as LLaVA-NeXT, MAmmoTH-VL, and VisualWebInstruct. Therefore, although Dimple, LaViDa, and LLaDA-V are all diffusion-based multimodal models, their results are influenced by different vision encoders, language backbones, data mixtures, and training schedules.

---

> ### Author Response · Authors · 2026-05-17
> **Rebuttal to Review Part 2**
>
> ### Improved discussion of limitations and open problems
>
> We thank the reviewer for this valuable suggestion. We will revise the manuscript to make the limitations and open problems more specific and explicitly connected to concrete future research directions. We categorize future directions into two types. The first comprises future work that arises from addressing current limitations and open problems; these are discussed in the Limitations section. The second includes directions that do not necessarily stem from the limitations or deficiencies of current dLLMs, but rather reflect broader developmental trends that may shape future research; these are presented in the Other Future Directions section.
>
> The revisions to the Limitations section broadly include the following content.
>
> First, current dLLMs often suffer from computational inefficiency during iterative denoising. Unlike autoregressive models, which generate one token per forward step, diffusion models usually perform repeated full-sequence computation over multiple denoising steps. This is inefficient because, at many intermediate steps, only a small subset of uncertain tokens may actually need to be updated. This motivates future research on sparse denoising, token-level early stopping, dynamic step allocation, and cache-aware diffusion decoding. These directions may reduce redundant computation while preserving the global refinement ability of diffusion models.
>
> Second, diffusion language models can be expensive to train from scratch. Since the model must learn to denoise sequences under many masking ratios or timesteps, the training signal may be less direct than next-token prediction in autoregressive models. A promising direction is to develop more data-efficient and compute-efficient training recipes. This includes adapting pretrained autoregressive models into diffusion models, designing curriculum-based masking schedules, improving timestep sampling, and distilling multi-step diffusion samplers into fewer denoising steps.
>
> Third, many current dLLMs are sensitive to decoding heuristics. Existing samplers often rely on manually designed unmasking rules based on confidence, margin, or entropy. However, if an incorrect token is unmasked too early, the error may be fixed and propagated to later denoising steps. This limitation suggests the need for principled decoding policies. Instead of selecting the unmasking set by a fixed rule, future work may learn a policy that decides which tokens to update or remask at each denoising step. Such a policy could be optimized using validation likelihood, task reward, uncertainty calibration, or downstream evaluation metrics. This may reduce error accumulation and improve robustness across different tasks and decoding budgets.
>
> Fourth, fixed output length remains a practical limitation for many diffusion language models. Since the generation process often starts from a sequence of mask tokens with a predefined length, the model must know the response length before sampling. If the chosen length is too short, the response may be truncated; if it is too long, computation is wasted and generation quality may degrade. Future research should therefore study dynamic length modeling, including flexible-length diffusion, insertion–deletion diffusion, explicit length-prediction modules, and hybrid decoding schemes.
>
> Fifth, training and serving pipelines for dLLMs remain less mature than those for autoregressive LLMs, especially for reinforcement learning and preference alignment. In autoregressive models, the sequence log-likelihood is tractable and decomposes as $log\ p_{\theta}(y | x) = sum_i\ log\ p_{\theta}(y_i | y_{<i}, x)$. In diffusion models, this factorization is not directly available. Sequence likelihood is usually approximated by an evidence lower bound or Monte Carlo estimator, which may introduce high variance during preference optimization or reinforcement learning. Future work should develop diffusion-native post-training algorithms, including lower-variance likelihood estimators, sequence-level preference objectives, stable reward modeling, and efficient policy optimization methods.
>
> Sixth, the scaling behavior of dLLMs is still not fully understood. Existing results suggest that diffusion language models may exhibit scaling trends similar to autoregressive models, but with different compute constants, inference costs, and likelihood estimation procedures. Therefore, comparing models only by perplexity or benchmark accuracy is insufficient. Future research should establish principled scaling laws and evaluation protocols for dLLMs that jointly consider accuracy, inference latency, number of denoising steps, training FLOPs, memory usage, and controllability.
>
> Overall, these limitations indicate that future progress in dLLMs and dMLLMs will likely depend on several interconnected research directions.